# FEDERATED $Q$-LEARNING WITH REFERENCE-ADVANTAGE DECOMPOSITION: ALMOST OPTIMAL REGRET AND LOGARITHMIC COMMUNICATION COST

**Zhong Zheng, Haochen Zhang & Lingzhou Xue**[*]
Department of Statistics
The Pennsylvania State University
State College, PA, 16802, USA
`{zvz5337,hqz5340,lzxue}@psu.edu`

## ABSTRACT

In this paper, we consider model-free federated reinforcement learning for tabular episodic Markov decision processes. Under the coordination of a central server, multiple agents collaboratively explore the environment and learn an optimal policy without sharing their raw data. Despite recent advances in federated $Q$-learning algorithms achieving near-linear regret speedup with low communication cost, existing algorithms only attain suboptimal regrets compared to the information bound. We propose a novel model-free federated $Q$-learning algorithm, termed FedQ-Advantage. Our algorithm leverages reference-advantage decomposition for variance reduction and adopts three novel designs: separate event-triggered communication and policy switching, heterogeneous communication triggering conditions, and optional forced synchronization. We prove that our algorithm not only requires a lower logarithmic communication cost but also achieves an almost optimal regret, reaching the information bound up to a logarithmic factor and near-linear regret speedup compared to its single-agent counterpart when the time horizon is sufficiently large.

## 1 INTRODUCTION

Federated reinforcement learning (FRL) is a distributed learning framework that combines the principles of reinforcement learning (RL) (Sutton & Barto, 2018) and federated learning (FL) (McMahan et al., 2017). Focusing on sequential decision-making, FRL aims to learn an optimal policy through parallel explorations by multiple agents under the coordination of a central server. Often modeled as a Markov decision process (MDP), multiple agents independently interact with an initially unknown environment and collaboratively train their decision-making models with limited information exchange between the agents. This approach accelerates the learning process with low communication costs. Some model-based algorithms (e.g., Chen et al. (2023)) and policy-based algorithms (e.g., Fan et al. (2021)) have shown speedup with respect to the number of agents in terms of learning regret or convergence rate. Recent progress has been made in FRL algorithms based on model-free value-based approaches, which directly learn the value functions and the optimal policy without estimating the underlying model (e.g., Woo et al. (2023)). However, limiting our focus to tabular MDPs, most existing model-free federated algorithms do not actively update the exploration policies for local agents and fail to provide low regret. A comprehensive literature review is provided in Appendix B.

### 1.1 FEDERATED $Q$-LEARNING: PRIOR WORKS AND LIMITATIONS

In this paper, we focus on model-free FRL based on the classic $Q$-learning algorithm (Watkins, 1989), tailored for episodic tabular MDPs with inhomogeneous transition kernels. Specifically, we assume the presence of a central server and $M$ local agents in the system. Each agent interacts independently with an episodic MDP consisting of $S$ states, $A$ actions, and $H$ steps per episode.

---

[*]Z. Zheng and H. Zhang are co-first authors. L. Xue is the corresponding author.

Let $T$ denote the number of steps for each agent. Under the single-agent setting of the episodic MDP, Domingues et al. (2021) and Jin et al. (2018) established a lower bound for the expected total regret of $\Omega(\sqrt{H^2SAT})$. An algorithm is considered almost optimal when it achieves a regret upper bound of $\tilde{O}(\sqrt{H^2SAT})$[1] for large values of $T$. Multiple model-based algorithms (e.g., Zhang et al. (2023)) have been shown to be almost optimal. Research on provably efficient model-free algorithms began with Jin et al. (2018) and was further advanced by Bai et al. (2019); Zhang et al. (2020); Li et al. (2021). Specifically, Zhang et al. (2020); Li et al. (2021) proposed almost optimal algorithms that utilized reference-advantage decomposition for variance reduction.

For the federated setting, the information bound naturally translates to $\Omega(\sqrt{H^2SAMT})$, allowing us to define almost optimal federated algorithms similarly. However, the literature on federated model-free algorithms is quite limited. Bai et al. (2019) and Zhang et al. (2020) proposed concurrent algorithms where multiple agents generate episodes simultaneously and share their original data with the central server. These designs achieved low policy-switching costs $O(\sqrt{H^3SAT})$ and $O(\sqrt{H^2SAT})$ respectively but incurred a high communication cost of $O(MT)$. Zheng et al. (2024a) proposed federated algorithms with near-linear regret speedup compared to Jin et al. (2018) and Bai et al. (2019) and logarithmic communication cost, but they only achieved a suboptimal regret of $\tilde{O}(\sqrt{MH^3SAT})$. This raises the following question:

*Is it possible to design an almost optimal federated model-free RL algorithm that enjoys a logarithmic communication cost?*

## 1.2 SUMMARY OF OUR CONTRIBUTIONS

We answer this question affirmatively by proposing the FedQ-Advantage algorithm to achieve the almost optimal regret and the logarithmic communication cost. Our contributions are summarized below.

- **Algorithmic design.** In FedQ-Advantage, the server coordinates the agents by actively updating their policies, while the agents execute these policies, collect trajectories, and periodically share local aggregations with the server. We adopt upper confidence bounds (UCB) to promote exploration and use the reference-advantage decomposition when updating the $Q$-function. The algorithm design incorporates the following key elements that are crucial for achieving near-optimal regret.

  (1) *Aligned round-wise communication and unaligned stage-wise updates.* We design a new mechanism based on event-triggered communication and policy switching, both of which are triggered when specific conditions are satisfied. This structure divides the learning process into aligned communication rounds, which are grouped into stages. These stages are unaligned across different state-action-step tuples. Communication takes place after each round, while policy switching occurs only at the end of each stage. This mechanism employs the unaligned stage design from Zhang et al. (2020), which is not used in Zheng et al. (2024a). During communication, local agents share the round-wise aggregated sums of function values over visits to each tuple rather than entire trajectories. The central server then constructs global estimates within the reference-advantage decomposition framework, maintaining low communication costs.

  (2) *Heterogeneous event-triggered Communication.* An agent terminates its exploration and requests communication in a round when the number of visits to any state-action-step tuple reaches a threshold, which guarantees sufficient exploration under restrictions. We adopt a heterogeneous design for the threshold that encourages more visits in the early rounds of a stage and limits the visits in later rounds to form desired stage renewals. This differs from the condition in Zheng et al. (2024a) that always poses strict limits.

  (3) *An optional forced synchronization mechanism.* Under this mechanism offered by FedQ-Advantage, when one agent triggers the communication condition, the central server terminates the exploration for all agents and initiates a new round. This approach enhances robustness to heterogeneity in agents' exploration speeds and eliminates waiting time. In the absence of forced synchronization, the central server waits for each agent to individually meet the communication condition, thereby reducing the number of communication rounds required.

---

[1] $\tilde{O}$ hides logarithmic factors.

Table 1: Comparison of regrets and communication costs for multi-agent RL algorithms.

| Type | Algorithm (Reference) | Regret | Communication cost |
|---|---|---|---|
| Model-based | Multi-batch RL (Zhang et al., 2022) | $\tilde{O}(\sqrt{H^2SAMT})$ | - |
| | APEVE (Qiao et al., 2022) | $\tilde{O}(\sqrt{H^4S^2AMT})$ | - |
| | Byzan-UCBVI (Chen et al., 2023) | $\tilde{O}(\sqrt{H^3S^2AMT})$ | $O(M^2H^2S^2A^2\log T)$ |
| Model-free | Concurrent Q-UCB2H (Bai et al., 2019) | $\tilde{O}(\sqrt{H^4SAMT})$ | $O(MT)$ |
| | Concurrent Q-UCB2B (Bai et al., 2019) | $\tilde{O}(\sqrt{H^3SAMT})$ | $O(MT)$ |
| | Concurrent UCB-Advantage (Zhang et al., 2020) | $\tilde{O}(\sqrt{H^2SAMT})$ | $O(MT)$ |
| | FedQ-Hoeffding (Zheng et al., 2024a) | $\tilde{O}(\sqrt{H^4SAMT})$ | $O(M^2H^4S^2A\log T)$ |
| | FedQ-Bernstein (Zheng et al., 2024a) | $\tilde{O}(\sqrt{H^3SAMT})$ | $O(M^2H^4S^2A\log T)$ |
| | FedQ-Advantage (**this work**) | $\tilde{O}(\sqrt{H^2SAMT})$ | $O(f_MMH^3S^2A(\log H)\log T)$ |

$H$: number of steps per episode; $T$: total number of steps; $S$: number of states; $A$: number of actions; $M$: number of agents. -: not discussed. $f_M$ equals to $M$ if the forced synchronization design is used and equals to 1 else.

- **Performance guarantees.** FedQ-Advantage provably achieves an almost optimal regret and near-linear speedup in the number of agents compared with its single-agent counterparts (Zhang et al., 2020) when $T$ is sufficiently large. The regret bound holds regardless of whether forced synchronization is used. Its communication cost scales logarithmically with $T$, outperforming the federated algorithms in Zheng et al. (2024a) and matching the policy switching cost in Zhang et al. (2020), which is the best cost for $Q$-learning in the literature. To the best of our knowledge, it is the first model-free federated RL algorithm to achieve almost optimal regret with logarithmic communication cost. We compare the regret and communication costs under multi-agent tabular episodic MDPs in Table 1. Numerical experiments also demonstrate that FedQ-Advantage has better regret and communication cost compared to the federated algorithms in Zheng et al. (2024a).

- **Technical novelty.** We highlight two technical contributions here. (1) *Stage-wise approximations in non-martingale analysis.* The event-triggered stage renewal presents a non-trivial challenge involving the concentration of the sum of non-martingale difference sequences. The weight assigned to each visit of a given tuple $(s, a, h)$ depends on the total number of visits between two model aggregation points, which is not causally known during the visitation. This paper proves the concentration by relating the sequence to a martingale difference sequence and bounding their stage-wise gap, which is different from Woo et al. (2023) and Woo et al. (2024) that used static behavior policies and bound similar gaps element-wisely. Our approach does not rely on a stationary visiting probability or the estimation of visiting numbers. (2) *Heterogeneous triggering conditions for synchronization.* For different rounds (of synchronization) in a given stage (of policy update), we use different triggering conditions that allow more visits of a tuple $(s, a, h)$ in early rounds. This reduces the number of synchronizations within a stage to $O(f_M \log H)$ from $O(f_M H)$, which would occur under homogeneous triggering conditions in Zheng et al. (2024a). This is key to improving the communication cost of Zheng et al. (2024a) and matching the policy switching cost of Zhang et al. (2020).

The rest of this paper is organized as follows. Section 2 provides the background and problem formulation. Section 3 presents the algorithm design of FedQ-Advantage. Section 4 studies the performance guarantees in terms of regret and communication cost. Section 5 concludes the paper. Related works, proofs, numerical experiments, and more details are presented in the appendices.

## 2 BACKGROUND AND PROBLEM FORMULATION

### 2.1 PRELIMINARIES

We first introduce the mathematical model and background on Markov decision processes. Throughout this paper, we assume that $0/0 = 0$. For any $C \in \mathbb{N}$, we use $[C]$ to denote the set $\{1, 2, \ldots C\}$. We use $\mathbb{I}[x]$ to denote the indicator function, which equals 1 when the event $x$ is true and 0 otherwise.

**Tabular episodic Markov decision process (MDP).** A tabular episodic MDP is denoted as $\mathcal{M} := (\mathcal{S}, \mathcal{A}, H, \mathbb{P}, r)$, where $\mathcal{S}$ is the set of states with $|\mathcal{S}| = S$, $\mathcal{A}$ is the set of actions with $|\mathcal{A}| = A$, $H$ is the number of steps in each episode, $\mathbb{P} := \{\mathbb{P}_h\}_{h=1}^H$ is the transition kernel so that $\mathbb{P}_h(\cdot \mid s, a)$ characterizes the distribution over the next state given the state action pair $(s, a)$ at step $h$, and

$r := \{r_h\}_{h=1}^H$ is the collection of reward functions. We assume that $r_h(s,a) \in [0,1]$ is a deterministic function of $(s,a)$, while the results can be easily extended to the case when $r_h$ is random.

In each episode of $\mathcal{M}$, an initial state $s_1$ is selected arbitrarily by an adversary. Then, at each step $h \in [H]$, an agent observes a state $s_h \in \mathcal{S}$, picks an action $a_h \in \mathcal{A}$, receives the reward $r_h = r_h(s_h, a_h)$ and then transits to the next state $s_{h+1}$. The episode ends when an absorbing state $s_{H+1}$ is reached. Later on, for the ease of presentation, we use "for any $(\forall)$ $(s,a,h)$" to represent "for any $(\forall)$ $(s,a,h) \in \mathcal{S} \times \mathcal{A} \times [H]$" and denote $\mathbb{P}_{s,a,h}f = \mathbb{E}_{s_{h+1}\sim\mathbb{P}_h(\cdot|s,a)}(f(s_{h+1})|s_h = s, a_h = a)$ and $\mathbb{1}_s f = f(s), \forall(s,a,h)$ for any function $f : \mathcal{S} \to \mathbb{R}$.

**Policies, state value functions, and action value functions.** A policy $\pi$ is a collection of $H$ functions $\left\{\pi_h : \mathcal{S} \to \Delta^{\mathcal{A}}\right\}_{h\in[H]}$, where $\Delta^{\mathcal{A}}$ is the set of probability distributions over $\mathcal{A}$. A policy is deterministic if for any $s \in \mathcal{S}$, $\pi_h(s)$ concentrates all the probability mass on an action $a \in \mathcal{A}$. In this case, we denote $\pi_h(s) = a$.

Let $V_h^\pi : \mathcal{S} \to \mathbb{R}$ and $Q_h^\pi : \mathcal{S} \times \mathcal{A} \to \mathbb{R}$ denote the state value function and the action value function at step $h$ under policy $\pi$. Mathematically, $V_h^\pi(s) := \sum_{h'=h}^H \mathbb{E}_{(s_{h'},a_{h'})\sim(\mathbb{P},\pi)}[r_{h'}(s_{h'},a_{h'}) \mid s_h = s]$. We also use $Q_h^\pi(s,a) := r_h(s,a) + \sum_{h'=h+1}^H \mathbb{E}_{(s_{h'},a_{h'})\sim(\mathbb{P},\pi)}[r_{h'}(s_{h'},a_{h'}) \mid s_h = s, a_h = a]$. Since the state and action spaces and the horizon are all finite, there always exists an optimal policy $\pi^\star$ that achieves the optimal value $V_h^\star(s) = \sup_\pi V_h^\pi(s) = V_h^{\pi^*}(s)$ for all $s \in \mathcal{S}$ and $h \in [H]$ (Azar et al., 2017). The Bellman equation and the Bellman optimality equation can be expressed as

$$
\begin{cases}
V_h^\pi(s) = \mathbb{E}_{a'\sim\pi_h(s)}[Q_h^\pi(s,a')] \\
Q_h^\pi(s,a) := r_h(s,a) + \mathbb{P}_{s,a,h}V_{h+1}^\pi \\
V_{H+1}^\pi(s) = 0, \forall(s,a,h)
\end{cases}
\text{and}
\begin{cases}
V_h^\star(s) = \max_{a'\in\mathcal{A}} Q_h^\star(s,a') \\
Q_h^\star(s,a) := r_h(s,a) + \mathbb{P}_{s,a,h}V_{h+1}^\star \\
V_{H+1}^\star(s) = 0, \forall(s,a,h).
\end{cases}
\quad (1)
$$

## 2.2 THE FEDERATED RL FRAMEWORK

We consider an FRL setting with a central server and $M$ agents, each interacting with an independent copy of $\mathcal{M}$. The agents communicate with the server periodically: after receiving local information, the central server aggregates it and broadcasts certain information to the agents to coordinate their exploration. We assume that the central server knows the reward functions $\{r_h\}_{h=1}^H$ beforehand[2]. We define the communication cost of an algorithm as the number of scalars (integers or real numbers) communicated between the server and agents similar to Zheng et al. (2024a).

For agent $m$, let $U_m$ be the number of generated episodes, $\pi^{m,u}$ be the policy in the $u$-th episode, and $s_1^{m,u}$ be the corresponding initial state. The regret of $M$ agents over $\hat{T} = H \sum_{m=1}^M U_m$ total steps is

$$
\text{Regret}(T) = \sum_{m\in[M]} \sum_{u=1}^{U_m} \left( V_1^\star(s_1^{m,u}) - V_1^{\pi^{m,u}}(s_1^{m,u}) \right).
$$

Here, $T := \hat{T}/M$ is the average total steps for $M$ agents.

## 3 ALGORITHM DESIGN

In this section, we elaborate on our model-free federated RL algorithm termed FedQ-Advantage.

### 3.1 BASIC STRUCTURE: ALIGNED ROUNDS AND UNALIGNED STAGES

We first review the single-agent algorithm in Zhang et al. (2020). The agent generates episodes and splits them into **stages** for each $(s,a,h)$. Denoting $y_t(s,a,h)$ as the number of visits to $(s,a,h)$ in the $t$-th stage for $(s,a,h)$, it requires that $y_{t+1}(s,a,h) = \lfloor(1+1/H)y_t(s,a,h)\rfloor$, and the updates of estimated $Q$-functions at $(s,a,h)$ only happen at the end of each stage. Due to the randomness of the visits, stage renewals for different triples might not happen simultaneously, resulting in unaligned stages. The exponential increase of the stage size leads to a low policy switching cost: the number of

---

[2]To handle unknown reward functions, we only need to slightly modify our algorithm to let agents share this information. This will not affect our Theorems 4.1 and 4.2 on regret and communication cost.

different implemented policies is upper bounded by $O(H^2 SA \log T)$. This provides the potential to parallelize the episodes generated under the same policy to multiple agents.

However, the simple design in Zheng et al. (2024a) does not accommodate the unaligned stages. Thus, FedQ-Advantage designs novel aligned rounds for unaligned stages. Next, we introduce our algorithm design, which is also visually shown in Figure 1. It proceeds in rounds indexed by $k \in [K]$ and agent $m$ generates $n^{m,k}$ episodes in round $k$. The communication between agents and the central server occurs at the end of each round. For each $(s, a, h)$, we divide rounds $k \in [K]$ into stages $t = 1, 2, \ldots$. Each stage contains consecutive multiple rounds: denote $k_h^t(s, a)$ as the index of the first round that belongs to stage $t$, so $k_h^t < k_h^{t+1}$, and stage $t$ is composed of rounds $k_h^t, k_h^t + 1, \ldots, k_h^{t+1} - 1$. Note that the definition of stages is specific to $(s, a, h)$, meaning that a given round may belong to different stages for different $(s, a, h)$. Each round equips the agents with a common policy $\pi^k$ for independent explorations and an event-triggered termination condition that will be explained later. At the end of each round, state renewal is judged for each $(s, a, h)$ separately, resulting in unaligned stages.

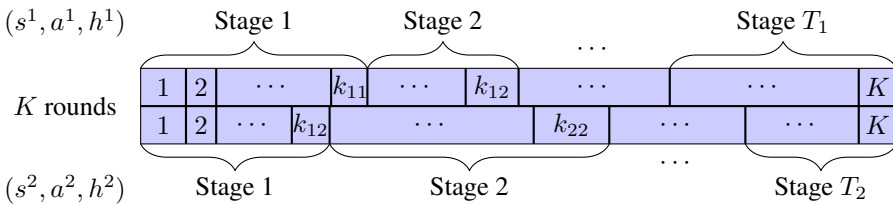

Figure 1: The relationship between rounds and stages for different triples $(s^1, a^1, h^1)$ and $(s^2, a^2, h^2)$. Each square represents a round, and the number inside indicates the round index. A stage is composed of consecutive rounds. Communication occurs at the end of each round and the estimated $Q$-function is updated at the end of each stage. We can find from the figure that a round may belong to different stages for different triples. For example, the round $k_{11}$ is in stage 1 of $(s^1, a^1, h^1)$, while in stage 2 of $(s^2, a^2, h^2)$. Here, $k_{it} = k_{h^i}^{t+1}(s^i, a^i) - 1$ represents the index of the last round in stage $t$ for $(s^i, a^i, h^i)$, $t \in \{1, 2, \cdots, T_i\}$ and $i \in \{1, 2\}$. $T_1$ and $T_2$ are the total number of stages for $(s^1, a^1, h^1)$ and $(s^2, a^2, h^2)$ respectively.

FedQ-Advantage updates the estimated $Q$-function at $(s, a, h)$ only at the end of each stage using stage-wise or global mean values regarding the next states of visits to $(s, a, h)$. Thus, agents only need to prepare and share corresponding local round-wise means for global aggregations. It results in an $O(MHS)$ communication cost within each round that is independent of the number of episodes.

### 3.2 ALGORITHM DETAILS

We provide a notation table in Appendix A to facilitate understanding of this section. For the $j$-th ($j \in [n^{m,k}]$) episode in the $k$-th round, let $s_1^{k,m,j}$ be the initial state for the $m$-th agent, and $\{(s_h^{k,m,j}, a_h^{k,m,j}, r_h^{k,m,j})\}_{h=1}^H$ be the corresponding trajectory. Define $\{V_h^k : \mathcal{S} \to \mathbb{R}\}_{h=1}^{H+1}$, $\{Q_h^k : \mathcal{S} \times \mathcal{A} \to \mathbb{R}\}_{h=1}^{H+1}$ and $\{V_h^{\mathrm{ref},k} : \mathcal{S} \to \mathbb{R}\}_{h=1}^{H+1}$ as the estimated $V$-function, the estimated $Q$-function and the reference function at the beginning of round $k$. Here, $Q_{H+1}^k, V_{H+1}^k, V_{H+1}^{\mathrm{ref},k} = 0$. We use $V_h^{\mathrm{adv},k} = V_h^k - V_h^{\mathrm{ref},k}$ to denote the estimated advantage function. For any predefined functions $g : \mathcal{S} \times \mathcal{A} \to \mathbb{R}$ or $f : \mathcal{S} \to \mathbb{R}$, we will use $g$ or $f$ in replace of $g(s, a)$ or $f(s)$ when there is no ambiguity for simplification. We also denote $t_h^k(s, a)$ as the stage index in round $k$ and $\mathbb{I}_h^{\mathrm{ren},k}(s, a) = \mathbb{I}[t_h^k > t_h^{k-1}]$ as a stage renewal indicator with $\mathbb{I}_h^{\mathrm{ren},1} = 1, \forall(s, a, h)$.

Then we briefly explain each component of the algorithm in round $k$ as follows.

**Step 1. Coordinated exploration for agents.** At the beginning of round $k$, the server holds the values on all states, actions, and steps for functions $\{Q_h^k, V_h^k, V_h^{\mathrm{ref},k}, N_h^k, n_h^k, \tilde{n}_h^k\}$. Here, $N_h^k(s, a)$ is the total number of visits for all agents up to but not including stage $t_h^k$, $n_h^k(s, a)$ is the total number of visits for all agents in the stage $t_h^k - 1$, and $\tilde{n}_h^k(s, a)$ is the number of visits for all the agents in the stage $t_h^k$ before the start of round $k$. Here, $n_h^k = 0$ if $t_h^k = 1$. When $k = 1$, $Q_h^1 = V_h^1 = V_h^{\mathrm{ref},1} = H, \forall(s, a, h)$ and $\pi^1$ is an arbitrary deterministic policy. It also holds

the following global values $\mu_h^{\text{ref},k}(s,a), \sigma_h^{\text{ref},k}(s,a), \mu_h^{\text{adv},k}(s,a), \sigma_h^{\text{adv},k}(s,a), \mu_h^{\text{val},k}(s,a), \forall(s,a,h)$. When $k=1$, they are initialized as 0. Further explanations will be provided in their updates in Step 4. Next, the central server decides a deterministic policy $\pi^k = \{\pi_h^k\}_{h=1}^H$, and then broadcasts $\pi_h^k$ along with $\{n_h^k(s,\pi_h^k(s)), \tilde{n}_h^k(s,\pi_h^k(s))\}_{s,h}$ and $\{V_h^k, V_h^{\text{ref},k}\}_{s,h}$ to all of the agents. Once receiving such information, the agents will execute policy $\pi^k$ and start collecting trajectories.

**Step 2. Event-triggered termination of exploration.** We introduce a Boolean variable $u_{syn}$ as an input to the algorithm for the forced synchronization. During the exploration under $\pi^k$, every agent will monitor its total number of visits for each $(s,a,h)$ triple within the current round. Define

$$c_h^k(s,a) = \begin{cases} \lfloor n_h^k(s,a)/(MH) \rfloor, & \text{if } n_h^k(s,a) > 0, \tilde{n}_h^k(s,a) > (1-1/H)n_h^k(s,a), \\ \max\left\{1, \lceil (n_h^k(s,a) - \tilde{n}_h^k(s,a))/M \rceil\right\}, & \text{otherwise.} \end{cases} \tag{2}$$

If $u_{syn} = \text{TRUE}$, for any agent $m$, at the end of each episode, if any $(s,a,h)$ has been visited by $c_h^k(s,a)$ times, the agent will stop exploration and send a signal to the server that requests all agents to abort the exploration. If $u_{syn} = \text{FALSE}$, the central server will wait until for each agent, there exists a triple $(s,a,h)$ that has been visited by $c_h^k(s,a)$ times.

During this process, each agent $m$ collect $n^{m,k}$ trajectories $\{(s_h^{k,m,j}, a_h^{k,m,j}, r_h^{k,m,j})\}_{h=1}^H, j \in [n^{m,k}]$ and calculates the following local quantities for $a = \pi_h^k(s)$:

$$n_h^{m,k}(s,a), \ \mu_{h,\text{ref}}^{m,k}(s,a), \ \mu_{h,\text{adv}}^{m,k}(s,a), \ \mu_{h,\text{val}}^{m,k}(s,a), \ \sigma_{h,\text{ref}}^{m,k}(s,a), \ \sigma_{h,\text{adv}}^{m,k}(s,a), \forall(s,h). \tag{3}$$

Here, $n_h^{m,k}(s,a)$ is the number of visits to $(s,a,h)$ for agent $m$ in round $k$. Thus, we have

$$\forall(s,a,h,m,k), \ n_h^{m,k}(s,a) \le c_h^k(s,a),$$

$$\forall k, \ \exists(s,a,h,m), \ s.t. \ n_h^{m,k}(s,a) = c_h^k(s,a), \quad \text{if } u_{syn} = \text{TRUE} \tag{4}$$

$$\forall m,k, \ \exists(s,a,h), \ s.t. \ n_h^{m,k}(s,a) = c_h^k(s,a), \quad \text{if } u_{syn} = \text{FALSE}. \tag{5}$$

Other quantities correspond to the summation of the values of five different functions applied to the next states of all the visits to $(s,a,h)$ for agent $m$ in round $k$. These five functions are $V_{h+1}^{\text{ref},k}, V_{h+1}^{\text{adv},k}, V_{h+1}^k, [V_{h+1}^{\text{ref},k}]^2, [V_{h+1}^{\text{adv},k}]^2$. Mathematically, for $f : \mathcal{S} \to \mathbb{R}$, letting $\mathbb{A}_{m,s,a,h}^k(f) = \sum_{j=1}^{n^{m,k}} f(s_{h+1}^{k,m,j}) \times \mathbb{I}[(s_h^{k,m,j}, a_h^{k,m,j}) = (s,a)]$ as the summation of $f$ on the next states for all the visits to $(s,a,h)$ for agent $m$ in round $k$. When there is no ambiguity, we will use the simplified notation $\mathbb{A}_m^k(f) = \mathbb{A}_{m,s,a,h}^k(f)$. Then, $\mu_{h,\text{ref}}^{m,k}(s,a) = \mathbb{A}_m^k(V_{h+1}^{\text{ref},k}), \mu_{h,\text{adv}}^{m,k}(s,a) = \mathbb{A}_m^k(V_{h+1}^{\text{adv},k})$, $\mu_{h,\text{val}}^{m,k}(s,a) = \mathbb{A}_m^k(V_{h+1}^k), \sigma_{h,\text{ref}}^{m,k}(s,a) = \mathbb{A}_m^k([V_{h+1}^{\text{ref},k}]^2)$ and $\sigma_{h,\text{adv}}^{m,k}(s,a) = \mathbb{A}_m^k([V_{h+1}^{\text{adv},k}]^2)$. These quantities correspond to local aggregations of different types of value functions and can be adaptively calculated when collecting the trajectories as shown in Algorithm 2.

**Step 3. Stage renewal.** After the exploration in round $k$, agents share local quantities in Equation (3) on all $(s,a,h)$ such that $a = \pi_h^k(s)$ to the central server. Then it finds existing visits in stage $t_h^k$ as

$$\hat{n}_h^{k+1}(s,a) = \tilde{n}_h^k(s,a) + \sum_{m=1}^M n_h^{m,k}(s,a), \forall(s,a,h), \tag{6}$$

and renew the stages for triples that are sufficiently visited: $\forall(s,a,h)$,

$$t_h^{k+1}(s,a) = t_h^k(s,a) + 1 \iff \hat{n}_h^{k+1}(s,a) \ge \mathbb{I}[n_h^k(s,a) = 0]MH + (1+1/H)n_h^k(s,a). \tag{7}$$

In Equation (7), when $n_h^k = 0$, i.e., $t_h^k = 1$, the state renewal threshold is $MH$. Else, it is $(1+1/H)n_h^k$. With Equation (7), the central can determine the stage renewal indicator $\mathbb{I}_h^{\text{ren},k+1}$ and the counts $N_h^{k+1}, n_h^{k+1}, \tilde{n}_h^{k+1}$ as shown in lines 13 and 18 in Algorithm 1.

**Step 4. Updates of estimated value functions and policies.** According to the stage renewal, the central server updates the global values $\mu_h^{\text{ref},k}, \sigma_h^{\text{ref},k}, \mu_h^{\text{adv},k}, \sigma_h^{\text{adv},k}, \mu_h^{\text{val},k}$ at all $(s,a,h)$ as follows:

$$(\mu,\sigma)_h^{\text{ref},k+1} = (\mu,\sigma)_h^{\text{ref},k} + \sum_m (\mu,\sigma)_{h,\text{ref}}^{m,k}, \tag{8}$$

$$(\mu^{\mathrm{adv}}, \mu^{\mathrm{val}}, \sigma^{\mathrm{adv}})_h^{k+1} = (1 - \mathbb{I}_h^{\mathrm{ren},k})(\mu^{\mathrm{adv}}, \mu^{\mathrm{val}}, \sigma^{\mathrm{adv}})_h^k + \sum_m (\mu_{h,\mathrm{adv}}, \mu_{h,\mathrm{val}}, \sigma_{h,\mathrm{adv}})^{m,k}. \qquad (9)$$

Equation (8) implies that $(\mu, \sigma)_h^{\mathrm{ref},k+1}(s,a)$ gives the sum of the estimated reference functions and squared reference functions at the next states for all agents and all visits to $(s,a,h)$ up to the end of round $k$. $(\mu^{\mathrm{adv}}, \mu^{\mathrm{val}}, \sigma^{\mathrm{adv}})_h^{k+1}(s,a)$ gives the sum of the estimated advantage functions, value functions, and squared advantage functions at the next states for all agents and all visits to $(s,a,h)$ in stage $t_h^k$ and up to the end of round $k$. Here, $(1 - \mathbb{I}_h^{\mathrm{ren},k})$ clears the historical cumulation if $k$ and $k-1$ belong to different stages.

Next, the central server updates the estimated $Q$-function for all $(s,a,h)$ triples with a stage renewal while keeping others unchanged:

$$Q_h^{k+1} = \min\{Q_h^{k+1,1}, Q_h^{k+1,2}, Q_h^k\}\mathbb{I}[t_h^{k+1} > t_h^k] + Q_h^k\mathbb{I}[t_h^{k+1} = t_h^k], \forall(s,a,h), \qquad (10)$$

Here, $Q_h^{k+1,1}, Q_h^{k+1,2}$ represents the Hoeffding-type used in Zheng et al. (2024a), and the reference-advantage type update used in Zhang et al. (2020), respectively:

$$Q_h^{k+1,1}(s,a) = r_h(s,a) + \mu_h^{\mathrm{val},k+1}/n_h^{k+1} + b_h^{k+1,1}(s,a), \qquad (11)$$

$$Q_h^{k+1,2}(s,a) = r_h(s,a) + \mu_h^{\mathrm{ref},k+1}/N_h^{k+1} + \mu_h^{\mathrm{adv},k+1}/n_h^{k+1} + b_h^{k+1,2}(s,a). \qquad (12)$$

In these updates where stage renewal happens, $N_h^{k+1}, n_h^{k+1}$ count all historical visits and visits in stage $t_h^k$, respectively. Thus, $\mu_h^{\mathrm{val},k+1}/n_h^{k+1}$ is the stage-wise mean of the estimated value function, $\mu_h^{\mathrm{adv},k+1}/n_h^{k+1}$ is the stage-wise mean of the estimated advantage function, and $\mu_h^{\mathrm{ref},k+1}/N_h^{k+1}$ gives the all-history estimated mean of reference function. $b_h^{k+1,1}, b_h^{k+1,2}$ are upper confidence bounds (UCB) that dominate the variances in the above empirical mean estimations. Their expressions are provided in line 15 of Algorithm 1.

Next, the central server updates the estimated $V$-function and the policy as follows:

$$V_h^{k+1}(s) = \max_{a' \in \mathcal{A}} Q_h^{k+1}(s,a'), \pi_h^{k+1}(s) \in \arg\max_{a' \in \mathcal{A}} Q_h^{k+1}(s,a'), \forall(s,h) \in \mathcal{S} \times [H]. \qquad (13)$$

**Step 5. Updates of the reference function.** With a constant $N_0 \in \mathbb{R}_+$, the central server conducts

$$V_h^{\mathrm{ref},k+1}(s) = V_h^{\mathrm{ref},1}(s)\mathbb{I}[k < k_{s,h}] + V_h^{k_{s,h}+1}(s)\mathbb{I}[k \geq k_{s,h}], \forall(s,h) \in \mathcal{S} \times [H]. \qquad (14)$$

Here, $k_{s,h} = \inf\{k \in \mathbb{N}_+ : \sum_{a' \in \mathcal{A}} N_h^{k+1}(s,a') \geq N_0\}$. Equation (14) means that at the end of round $k$, for all $(s,h)$ such that the stage for $(s, \pi_h^k(s), h)$ is renewed, we will update the reference function at $(s,h)$ based on the updated value function $V_h^{k+1}$ if round $k$ is the first round such that the global visiting number to $(s,h)$ across all complete stages reaches $N_0$. "First round" indicates that the reference update on each $(s,h)$ happens at most once during the whole learning process with $k = 1, 2 \ldots$, and the reference function on $(s,h)$ will be settled after its update. This design matches the single-agent algorithms in Zhang et al. (2020) and Li et al. (2021).

Now we are ready to provide FedQ-Advantage in Algorithms 1 and 2 for the behaviors of the central server and the agents. In Algorithm 1, $T_0$ limits the total number of steps for all agents. Line 20 in Algorithm 1 updates the reference function at $(s,h)$ when the visiting number exceeds $N_0$ for the first time and keeps it unchanged for other situations, coinciding with Equation (14).

### 3.3 INTUITION BEHIND THE ALGORITHM DESIGN

**Exponentially increasing stage sizes: infrequent policy switching.** FedQ-Advantage guarantees that $y_{t+1}(s,a,h) = (1 + \Theta(1)/H)y_t(s,a,h)$, where $y_t(s,a,h)$ represents the number of visits to $(s,a,h)$ in stage $t$. The exponential increasing rate is controlled by the threshold $c_h^k(s,a)$ in Equation (2) and stage renewal condition in Equation (7). By analyzing the two cases of Equation (2), we can prove that $\hat{n}_h^{k+1} \leq (1 + 2/H)n_h^k$, which implies that $y_{t+1} \leq (1 + 2/H)y_t$. Equation (7) further implies that $y_{t+1} \geq (1 + 1/H)y_t$. The details are provided in (d) and (e) of Lemma D.1. Our visits in stages satisfy a similar exponential increasing pattern as $y_{t+1} = \lfloor(1 + 1/H)y_t\rfloor$ in Zhang et al. (2020), and FedQ-Advantage switches policies infrequently since estimated $Q-$functions and the policies are only updated after stage renewals.

---

**Algorithm 1** FedQ-Advantage (Central Server)

---

1: **Input and Initialization:** $T_0, N_0 \in \mathbb{N}_+, p \in (0,1)$. Functions $Q_h^1 = V_h^1 = V_h^{\text{ref},1} = H$, function $\mathbb{I}_h^{\text{ren},1} = 1$, functions $\mathbb{I}_h^{\text{ref},1} = N_h^1 = n_h^1 = \tilde{n}_h^1 = \mu_h^{\text{ref},1} = \mu_h^{\text{adv},1} = \mu_h^{\text{val},1} = \sigma_h^{\text{ref},1} = \sigma_h^{\text{adv},1} = 0, \forall(s,a,h)$. Arbitrary deterministic policy $\pi^1$. $n_{step} = k = 0$. $u_{syn} \in \{\text{TRUE}, \text{FALSE}\}$.
2: **while** $n_{step} < T_0$ **do**
3:    % Step 1. Coordinated exploration for agents.
4:    Broadcast $\pi^k$ and $(V_h^k, V_h^{\text{ref},k}, n_h^k, \tilde{n}_h^k), \forall(s,a,h)$ with $a = \pi_h^k(s)$ to all clients.
   % Step 2. Event-triggered termination of exploration.
5:    **if** $u_{syn} = \text{TRUE}$ **then**
6:      Wait until receiving an abortion signal and send the signal to all agents.
7:    **else**
8:      Wait until receiving abortion signals from all agents.
9:    **end if**
10:   Receive $n_h^{m,k}$ and $\{\mu_{h,\text{ref}}^{m,k}, \mu_{h,\text{adv}}^{m,k}, \mu_{h,\text{val}}^{m,k}\}, \{\sigma_{h,\text{ref}}^{m,k}, \sigma_{h,\text{adv}}^{m,k}\}, \forall(s, \pi_h^k(s), h)$ from agents.
11:   Calculate $\hat{n}_h^{k+1}, (\mu,\sigma)_h^{\text{ref},k+1}, (\mu^{\text{adv}}, \mu^{\text{val}}, \sigma^{\text{adv}})_h^{k+1}$ via Equations (6), (8) and (9), $\forall(s,a,h)$.
  % Step 3. Stage renewal.
12:   **for** $\forall(s,a,h)$ **do**
13:     **if** $\hat{n}_h^{k+1}(s,a) \geq \mathbb{I}[n_h^k(s,a) = 0]MH + (1 + 1/H)n_h^k(s,a)$, **then**
14:       (Stage renewal) $\mathbb{I}_h^{\text{ren},k+1} = 1, n_h^{k+1} = \hat{n}_h^{k+1}, \tilde{n}_h^{k+1} = 0, N_h^{k+1} = N_h^k(s,a) + n_h^{k+1}$.
      % Step 4. Updates of estimated value functions and policies.
15:       Update $Q_h^{k+1}(s,a)$ based on Equations (10) to (12). Here, $b_h^{k+1,1}(s,a) = \sqrt{2H^2\iota/n_h^{k+1}}$. $b_h^{k+1,2}(s,a) = 2\sqrt{\hat{\text{var}}_h^{\text{ref},k+1}/N_h^{k+1}} + 2\sqrt{\hat{\text{var}}_h^{\text{adv},k+1}/n_h^{k+1}} + 10H((\iota/N_h^{k+1})^{3/4} + (\iota/n_h^{k+1})^{3/4} + \iota/N_h^{k+1} + \iota/n_h^{k+1}), \iota = \log(2/p), \hat{\text{var}}_h^{\text{ref},k+1} = \sigma_h^{\text{ref},k+1}/N_h^{k+1} - (\mu_h^{\text{ref},k+1}/N_h^{k+1})^2, \hat{\text{var}}_h^{\text{adv},k+1} = \sigma_h^{\text{adv},k+1}/n_h^{k+1} - (\mu_h^{\text{adv},k+1}/n_h^{k+1})^2$.
16:     **else**
17:       (Stage unchanged) $Q_h^{k+1} = Q_h^k, \mathbb{I}_h^{\text{ren},k+1} = 0, n_h^{k+1} = n_h^k, \tilde{n}_h^{k+1} = \hat{n}_h^{k+1}, N_h^{k+1} = N_h^k$.
18:     **end if**
19:   **end for**
20:   Find $V_h^{k+1}, \pi_h^{k+1}$ from Equation (13), $\mathbb{I}_h^{\text{ref},k+1} = \mathbb{I}[\sum_{a' \in \mathcal{A}} N_h^{k+1}(s,a') \geq N_0], \forall(s,h)$.
  % Step 5. Updates of the reference function.
21:   $V_h^{\text{ref},k+1} = V_h^{\text{ref},k}\left(1 - \mathbb{I}_h^{\text{ref},k+1}(1 - \mathbb{I}_h^{\text{ref},k})\right) + V_h^{k+1}\mathbb{I}_h^{\text{ref},k+1}(1 - \mathbb{I}_h^{\text{ref},k}), \forall(s,h)$.
22:   $n_{step} \stackrel{\pm}{=} \sum_{m,s,a,h} n_h^{m,k}, k \stackrel{\pm}{=} 1$.
23: **end while**

---

**Algorithm 2** FedQ-Advantage (Agent $m$ in round $k$)

---

1: Receive $\pi^k$ and $(V_h^k, V_h^{\text{ref},k}, n_h^k, \tilde{n}_h^k), \forall(s,a,h)$ with $a = \pi_h^k(s)$ from the central server.
2: Initialization: functions $n_h^m, \mu_{h,\text{ref}}^m, \mu_{h,\text{adv}}^m, \mu_{h,\text{val}}^m, \sigma_{h,\text{ref}}^m, \sigma_{h,\text{adv}}^m \leftarrow 0, \forall(s, \pi_h^k(s), h)$.
3: **while** no abortion signal sent or from the central server **do**
4:   **while** $n_h^m(s,a) < c_h^k(s,a), \forall(s,a,h)$ with $a = \pi_h^k(s)$ **do**
5:     Collect a new trajectory $\{(s_h, a_h, r_h)\}_{h=1}^H$ with $a_h = \pi_h^k(s_h)$.
6:     Update local incremental quantities: $(n_h^m, \mu_{h,\text{ref}}^m, \mu_{h,\text{adv}}^m, \mu_{h,\text{val}}^m, \sigma_{h,\text{ref}}^m, \sigma_{h,\text{adv}}^m)(s_h, a_h)$
7:     $\stackrel{\pm}{=} (1, V_{h+1}^{\text{ref},k}, V_{h+1}^{\text{adv},k}, V_{h+1}^k, [V_{h+1}^{\text{ref},k}]^2, [V_{h+1}^{\text{adv},k}]^2)(s_{h+1}), \forall h$.
8:   **end while**
9:   Send an abortion signal to the central server.
10: **end while**
11: Functions $(n_h^{m,k}, \mu_{h,\text{ref}}^{m,k}, \mu_{h,\text{adv}}^{m,k}, \mu_{h,\text{val}}^{m,k}, \sigma_{h,\text{ref}}^{m,k}, \sigma_{h,\text{adv}}^{m,k}) \leftarrow (n_h^m, \mu_{h,\text{ref}}^m, \mu_{h,\text{adv}}^m, \mu_{h,\text{val}}^m, \sigma_{h,\text{ref}}^m, \sigma_{h,\text{adv}}^m), \forall(s, \pi_h^k(s), h)$ and send $\left\{(n_h^{m,k}, \mu_{h,\text{ref}}^{m,k}, \mu_{h,\text{adv}}^{m,k}, \mu_{h,\text{val}}^{m,k}, \sigma_{h,\text{ref}}^{m,k}, \sigma_{h,\text{adv}}^{m,k})\right\}_{s,h}$ to the central server.

---

**Reference-advantage decompositions: the key to the almost optimal regret.** $Q$-learning algorithms (Jin et al., 2018; Bai et al., 2019; Zhang et al., 2020; Li et al., 2021; Zheng et al., 2024a) update

the estimated $Q$-function in the following form: $Q_h(s,a) \leftarrow r_h(s,a) + \text{EST}(\mathbb{P}_{s,a,h} V^\star_{h+1}) + b$. Here, $b > 0$ is the upper confidence bound (UCB) that promotes exploration, and $\text{EST}(\cdot)$ represents the empirical estimation, which takes the form of a weighted sum of the historically estimated value functions for the next states following the visits to $(s,a,h)$. This update is motivated by the Bellman optimality equation. The error $\text{EST}(\mathbb{P}_{s,a,h} V^\star_{h+1}) - \mathbb{P}_{s,a,h} V^\star_{h+1}$ can be decomposed into the variance from the random transitions to the next states and the bias in the estimated value functions. To handle the bias that is more severe in the early visits, Jin et al. (2018); Bai et al. (2019); Zheng et al. (2024a) required that the weights concentrate on the most recent $\Theta(1/H)$ proportion of visits like Equation (11) that only use visits in the current stage, which causes sample inefficiency and suboptimal regret.

To address this issue, FedQ-Advantage uses the reference-advantage decomposition adopted by Zhang et al. (2020); Li et al. (2021). Equation (12) in Step 4 represents the decomposition. We decompose the estimation of $\mathbb{P}_{s,a,h} V^\star_{h+1}$ into the reference part $\mu^{\text{ref},k+1}_h / N^{k+1}_h$ and the advantage part $\mu^{\text{adv},k+1}_h / n^{k+1}_h$. For the advantage part, we use stage-wise mean to eliminate the large biases in early value estimations. For the reference part, since the reference function will settle after $N_0 = \tilde{O}(1)$ visits as shown in Step 5, we can neglect the bias and use the mean of all historical visits. This design reduces the error in the empirical estimation by improving the sample efficiency in the reference part and restricting the error ranges in the advantage part. The reference-advantage decomposition, together with the exponentially increasing rate of stage sizes, is the key to our improved regret compared to Zheng et al. (2024a) and our linear regret speedup compared to the almost optimal regret given in Zhang et al. (2020).

**Heterogeneous event-triggered communication: the key to our improved communication cost.** FedQ-Advantage uses $c^k_h$ given in Equation (2) to limit the number of new visits for agent $m$ in round $k$. While the first case in Equation (2) is similar to the homogeneous condition in Zheng et al. (2024a), we design the second case to allow more new visits when the number of existing visits in the current stage is small. Specifically, when $n^k_h \geq MH$ and $\tilde{n}^k_h \leq (1 - 1/H)n^k_h$, $c^k_h = \lceil (n^k_h - \tilde{n}^k_h)/M \rceil \geq \lfloor n^k_h/(MH) \rfloor$. Thus, FedQ-Advantage allows more visits in the early rounds of each stage compared to Zheng et al. (2024a) and reduces the number of communication rounds, which is key to our improved communication cost shown in Table 1.

**Optional forced synchronization: accommodating heterogeneous exploration speeds.** Section 3.2 and eqs. (4) and (5) highlight the effect of the optional forced synchronization used in Step 2. Section 3.2 shows a common limitation of new visits, which is sufficient for our linear regret speedup compared to the single-agent algorithm in Zhang et al. (2020). Next, we discuss the robustness and trade-offs of optional forced synchronization when under heterogeneous exploration speeds of agents.

When optional forced synchronization is enabled (i.e., $u_{syn} = \text{TRUE}$), exploration and communication occur as soon as **one** agent reaches the threshold $c^k_h(s,a)$. This allows faster agents to avoid waiting for slower ones, minimizing waiting time. However, Equation (4) guarantees sufficient exploration by only one agent, resulting in varied episode counts across agents. This configuration is suitable for tasks sensitive to waiting time.

When optional forced synchronization is disabled (i.e., $u_{syn} = \text{FALSE}$), communication occurs only after **all** agents meet the threshold $c^k_h(s,a)$. Equation (5) ensures sufficient exploration by all agents, with episode counts being roughly balanced. This allows for more extensive exploration within a round, reducing communication costs but potentially increasing waiting time for faster agents.

## 4 PERFORMANCE GUARANTEES

Next, we provide regret upper bound for FedQ-Advantage as follows.

**Theorem 4.1** (Regret of FedQ-Advantage). *Let $\iota = \log(2/p)$ with $p \in (0,1)$ and $N_0 = 5184 \frac{SAH^5\iota}{\beta^2} + 16 \frac{MSAH^3}{\beta}$ with $\beta \in (0, H]$. For Algorithms 1 and 2, with probability at least $1 - (4SAT^5_1 + SAHT^4_1 + 5SAT^2_1/H + 5SAT_1 + 5)p$, we have*

$$Regret(T) \leq \tilde{O}\left((1 + \beta)\sqrt{MSAH^2T} + Mpoly(HSA, 1/\beta)\right).$$

*Here, $K$ is the total number of rounds, $T = H \sum_{k=1}^{K} n^k$ is the total number of steps for each agent, $T_1 = (2 + \frac{2}{H})T_0 + MSAH(H+1)$, and $\tilde{O}$ hides logarithmic multipliers on $T_0, M, H, S, A, 1/p$ and poly represents some polynomial. This result does not depend on the value of $u_{syn}$. See Equation* (26) *in Appendix E for the complete upper bound.*

Theorem 4.1 indicates that the total regret scales as $\tilde{O}(\sqrt{H^2 MTSA})$ when $T$ is larger than some polynomial of $MHSA$ and $\beta = \Omega(1)$ in $N_0$. This is almost optimal compared to the information bound $\Omega(\sqrt{H^2 SAMT})$ and is better than $\tilde{O}(\sqrt{H^3 SAMT})$ for algorithms in Zheng et al. (2024a). When $M = 1$, our regret bound becomes $\tilde{O}((1 + \beta)\sqrt{H^2 TSA})$ when $T$ is large, which is better than $\tilde{O}((1 + \beta\sqrt{H})\sqrt{H^2 TSA})$ in Zhang et al. (2020) thanks to our tighter regret analysis. This also means that to reach an almost optimal regret bound, Zhang et al. (2020) requires $\beta \leq O(1/\sqrt{H})$ and FedQ-Advantage lays a weaker one $\beta \leq \Omega(1)$. When $M > 1$, focusing on the dominate terms $\tilde{O}\left((1 + \beta)\sqrt{MSAH^2 T}\right)$ when $T$ is large, our algorithm achieves a near-linear regret speedup while the overhead term $\tilde{O}(M\text{poly}(HSA, 1/\beta))$ results from the burn-in cost for using reference-advantage decomposition (Zhang et al., 2020), and the $\Omega(HM)$ visits collected in the first stage for each $(s, a, h)$, which servers as the multi-agent burn-in cost that is common in federated algorithms (see e.g. Zheng et al. (2024a); Woo et al. (2023; 2024)).

Next, we discuss the improved communication cost compared to Zheng et al. (2024a) as follows.

**Theorem 4.2** (Communication rounds of FedQ-Advantage). *Under Algorithms 1 and 2 with $u_{syn} =$ TRUE, the number of communication rounds $K$ and the total number of steps $T$ satisfy that*

$$K \leq MSAH^2 + 4MSAH^2(\log(H) + 3)\log\left(\frac{T}{SAH^3} + 1\right).$$

*If $u_{syn} =$ FALSE, the number of communication rounds $K$ and the total number of steps $T$ satisfy*

$$K \leq SAH^2 + 4SAH^2(\log(H) + 3)\log\left(\frac{T}{SAH^3} + 1\right).$$

Theorem 4.2 implies if $u_{syn} =$ TRUE and $T$ is sufficiently large, $K = O\left(MH^2 SA(\log H)\log T\right)$. Since the total number of communicated scalars is $O(MHS)$ in each round, the total communication cost scales in $O(M^2 H^3 S^2 A(\log H)\log T)$. Thanks to the heterogeneous design in $c_h^k(s, a)$, it is better than $O(M^2 H^4 S^2 A \log T)$ for FedQ-Hoeffding and FedQ-Bernstein in Zheng et al. (2024a). If $u_{syn} =$ FALSE, when $T$ is sufficiently large, $K = O\left(H^2 SA(\log H)\log T\right)$, which is independent of $M$. Since the total number of communicated scalars is $O(MHS)$ in each round, the total communication cost scales in $O(MH^3 S^2 A(\log H)\log T)$.

Our result for $u_{syn} =$ FALSE also implies a low policy switching cost, which is defined as the times of policy switching. Knowing that the cost of Zhang et al. (2020) is $O(H^2 SA \log(T))$ and $K = O\left(H^2 SA(\log H)\log T\right)$ for FedQ-Advantage, our communication round matches the policy switching cost up to a logarithmic factor under restrictions on sharing original trajectories. We also remark Equation (102) in Appendix F shows that FedQ-Advantage can also reach the same local switching cost as Zhang et al. (2020). We refer readers to Zhang et al. (2020) for more information.

We will provide the complete proofs of Theorems 4.1 and 4.2 in Appendices E and F respectively.

## 5 CONCLUSION

This paper develops the model-free FRL algorithm FedQ-Advantage with provably almost optimal regret and logarithmic communication cost. Specifically, it achieves an almost-optimal regret, reaching the information bound up to a logarithmic factor and near-linear regret speedup compared to its single-agent counterpart when the time horizon is sufficiently large. Our algorithm also improves the logarithmic communication cost in the literature. Technically, our algorithm uses the UCB, reference-advantage decomposition and designs separate mechanisms for synchronization and policy switching, which can find broader applications for other RL problems.

## ACKNOWLEDGMENT

The work of Z. Zheng, H. Zhang, and L. Xue was supported by the U.S. National Science Foundation under the grants DMS-1811552, DMS-1953189, and CCF-2007823 and by the U.S. National Institutes of Health under the grant 1R01GM152812.

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

**Organization of the appendix.** In the appendix, we first provide two tables to summarize the notations in Section A. Section B provides related works. Section C presents numerical results. Section D provides basic facts of FedQ-Advantage and a lemma of concentration inequalities for regret analysis. Section E presents the proof of Theorem 4.1 (regret). Section F presents the proof of Theorem 4.2 (communication cost).

## A    NOTATION TABLES

In this section, we provide two notation tables to enhance the readability of the paper. The notations are categorized into two groups: one group consists of global variables utilized for central server aggregation, while the other group consists of local variables employed for agent training.   First,

Table 2: Global Variables

| Variable | Definition |
|---|---|
| $V_h^\pi$ | the state value function at step $h$ under policy $\pi$ |
| $Q_h^\pi$ | the state-action value function at step $h$ under policy $\pi$ |
| $V_h^\star$ | the state value function at step $h$ under the optimal policy $\pi^\star$ |
| $Q_h^\star$ | the state-action value function at step $h$ under the optimal policy $\pi^\star$ |
| $\mathbb{P}_{s,a,h}f$ | $\mathbb{E}_{s_{h+1}\sim\mathbb{P}_h(\cdot\|s,a)}(f(s_{h+1})\|s_h = s, a_h = a)$ |
| $\mathbb{1}_{s,a,h}f$ | $f(s)$ |
| $V_h^{\text{ref}}$ | the reference function at step $h$ |
| $V_h^{\text{adv}}$ | the advantage function at step $h$ |
| $V_h^k$ | the estimated state value function of step $h$ at the beginning of the round $k$ |
| $Q_h^k$ | the estimated state-action value function of step $h$ at the beginning of the round $k$ |
| $V_h^{\text{ref},k}$ | the reference function of step $h$ at the beginning of the round $k$ |
| $V_h^{\text{adv},k}$ | the advantage function of step $h$ at the beginning of the round $k$ |
| $t_h^k(s,a)$ | the stage index of the triple $(s,a,h)$ in the round $k$ |
| $\mathbb{I}_h^{\text{ren},k}$ | $\mathbb{I}[t_h^k > t_h^{k-1}]$, a stage renewal indicator in the round $k-1$ |
| $N_h^k(s,a)$ | the total number of visits to $(s,a,h)$ before the stage $t_h^k(s,a)$ |
| $n_h^k(s,a)$ | the total number of visits to $(s,a,h)$ in the stage $t_h^k(s,a)-1$ |
| $\tilde{n}_h^k(s,a)$ | the number of visits to $(s,a,h)$ in the stage $t_h^k(s,a)$ before the start of round $k$ |
| $\hat{n}_h^k(s,a)$ | the number of visits to $(s,a,h)$ in the stage $t_h^{k-1}(s,a)$ before the start of round $k$ |
| $\mu_h^{\text{ref},k}(s,a)$ | the sum of the reference function at step $h+1$ with regard to all visits to $(s,a,h)$ before round $k$ |
| $\sigma_h^{\text{ref},k}(s,a)$ | the sum of the squared reference function at step $h+1$ with regard to all visits to $(s,a,h)$ before round $k$ |
| $\mu_h^{\text{adv},k}(s,a)$ | the sum of the advantage function at step $h+1$ with regard to all visits to $(s,a,h)$ during stage $t_h^{k-1}$ and before round $k$ |
| $\sigma_h^{\text{adv},k}(s,a)$ | the sum of the squared advantage function at step $h+1$ with regard to all visits to $(s,a,h)$ during stage $t_h^{k-1}$ and before round $k$ |
| $\mu_h^{\text{val},k}(s,a)$ | the sum of the estimated state value function at step $h+1$ with regard to all visits to $(s,a,h)$ during stage $t_h^{k-1}$ and before round $k$ |
| $c_h^k(s,a)$ | the exploration termination constant for triple $(s,a,h)$ |
| $\mathbb{I}_h^{\text{ref},k}$ | a reference function renewal indicator in the round $k-1$ |

we introduce some local quantities. For $f : \mathcal{S} \to \mathbb{R}$, letting $\mathbb{A}_{m,s,a,h}^k(f) = \sum_{j=1}^{n^{m,k}} f(s_{h+1}^{k,m,j})\times$ $\mathbb{I}[(s,a)_h^{k,m,j} = (s,a)]$ as the summation of $f$ on the next states for all the visits to $(s,a,h)$ in round $k$

Table 3: Local Variables

| Variable | Definition |
|---|---|
| $n_h^{m,k}(s,a)$ | the total number of visits to $(s,a,h)$ of the agent $m$ in the round $k$ |
| $\mu_{h,\text{ref}}^{m,k}(s,a)$ | the sum of the reference function at step $h+1$ with regard to all visits to $(s,a,h)$ of the agent $m$ in the round $k$ |
| $\sigma_{h,\text{ref}}^{m,k}(s,a)$ | the sum of the squared reference function at step $h+1$ with regard to all visits to $(s,a,h)$ of the agent $m$ in the round $k$ |
| $\mu_{h,\text{adv}}^{m,k}(s,a)$ | the sum of the advantage function at step $h+1$ with regard to all visits to $(s,a,h)$ of the agent $m$ in the round $k$ |
| $\sigma_{h,\text{adv}}^{m,k}(s,a)$ | the sum of the squared advantage function at step $h+1$ with regard to all visits to $(s,a,h)$ of the agent $m$ in the round $k$ |
| $\mu_{h,\text{val}}^{m,k}(s,a)$ | the sum of the estimated state value function at step $h+1$ with regard to all visits to $(s,a,h)$ of the agent $m$ in the round $k$ |

for agent $m$. When there is no ambiguity, we will use the simplified notation $\mathbb{A}_m^k(f) = \mathbb{A}_{m,s,a,h}^k(f)$. Then, we let $n_h^{m,k}(s,a) = \mathbb{A}_m^k(1)$, $\mu_{h,\text{ref}}^{m,k}(s,a) = \mathbb{A}_m^k(V_{h+1}^{\text{ref},k})$, $\mu_{h,\text{adv}}^{m,k}(s,a) = \mathbb{A}_m^k(V_{h+1}^{\text{adv},k})$, $\mu_{h,\text{val}}^{m,k}(s,a) = \mathbb{A}_m^k(V_{h+1}^k)$, $\sigma_{h,\text{ref}}^{m,k}(s,a) = \mathbb{A}_m^k([V_{h+1}^{\text{ref},k}]^2)$ and $\sigma_{h,\text{adv}}^{m,k}(s,a) = \mathbb{A}_m^k([V_{h+1}^{\text{adv},k}]^2)$. For these functions of $(s,a)$, $n_h^{m,k}$ is the local count of visits for agent $m$ in round $k$, and the remaining ones are local summations related to the reference function, the estimated advantage functions, and the estimated value functions for visits.

Accordingly, we define some global quantities. First, we focus on visiting counts. We let $N_h^k(s,a) = \sum_{k':t_h^{k'}<t_h^k} \sum_m n_h^{m,k'}$ be the total number of visits to $(s,a,h)$ up to but not including stage $t_h^k$, $n_h^k(s,a) = \sum_{k':t_h^{k'}=t_h^k-1} \sum_m n_h^{m,k'}$ be the number of visits to $(s,a,h)$ in the stage $t_h^k - 1$. Here, $n_h^k = 0$ if $t_h^k = 1$. We also let $\tilde{n}_h^k(s,a) = \sum_{k':k'<k,t_h^{k'}=t_h^k} \sum_m n_h^{m,k'}$ and $\hat{n}_h^k(s,a) = \sum_{k':k'<k,t_h^{k'}=t_h^{k-1}} \sum_m n_h^{m,k'}$ be the number of visits to $(s,a,h)$ in the stage $t_h^k$ or $t_h^{k-1}$ before the start of round $k$. Next, we provide quantities of summations. Let $\mu_h^{\text{ref},k}(s,a) = \sum_{k':k'<k} \sum_m \mu_{h,\text{ref}}^{m,k'}$, $\sigma_h^{\text{ref},k}(s,a) = \sum_{k':k'<k} \sum_m \sigma_{h,\text{ref}}^{m,k'}$, $\mu_h^{\text{adv},k}(s,a) = \sum_{k':k'<k,t_h^{k'}=t_h^{k-1}} \sum_m \mu_{h,\text{adv}}^{m,k'}$, $\sigma_h^{\text{adv},k}(s,a) = \sum_{k':k'<k,t_h^{k'}=t_h^{k-1}} \sum_m \sigma_{h,\text{adv}}^{m,k'}$ and $\mu_h^{\text{val},k}(s,a) = \sum_{k':k'<k,t_h^{k'}=t_h^{k-1}} \sum_m \mu_{h,\text{val}}^{m,k'}$. Here, $\mu_h^{\text{ref},k}$ and $\sigma_h^{\text{ref},k}$ represent the sum of the reference function or squared reference function at step $h+1$ with regard to all visits of $(s,a,h)$ before round $k$, and $\mu_h^{\text{adv},k}, \sigma_h^{\text{adv},k}, \mu_h^{\text{val},k}$ are the sum of the advantage function, squared advantage function, and the estimated value function at step $h+1$ with regard to visits of $(s,a,h)$ during stage $t_h^{k-1}$ and before round $k$.

# B   RELATED WORKS

**Single-agent episodic MDPs.** There are mainly two types of algorithms for reinforcement learning: model-based and model-free learning. Model-based algorithms learn a model from past experience and make decisions based on this model, while model-free algorithms only maintain a group of value functions and take the induced optimal actions. Due to these differences, model-free algorithms are usually more space-efficient and time-efficient compared to model-based algorithms. However, model-based algorithms may achieve better learning performance by leveraging the learned model.

Next, we discuss the literature on model-based and model-free algorithms for single-agent episodic MDPs. Auer et al. (2008), Agrawal & Jia (2017), Azar et al. (2017), Kakade et al. (2018), Agarwal et al. (2020), Dann et al. (2019), Zanette & Brunskill (2019),Zhang et al. (2021),Zhou et al. (2023) and Zhang et al. (2023) worked on model-based algorithms. Notably, Zhang et al. (2023) provided an algorithm that achieves a regret of $\tilde{O}\left(\min\{\sqrt{SAH^2T}, T\}\right)$, which matches the information lower bound. Jin et al. (2018), Yang et al. (2021), Zhang et al. (2020), Li et al. (2021) and Ménard et al.

(2021) work on model-free algorithms. The latter three have introduced algorithms that achieve minimax regret of $\tilde{O}\left(\sqrt{SAH^2T}\right)$.

**Variance reduction in RL.** The reference-advantage decomposition used in Zhang et al. (2020) and Li et al. (2021) is a technique of variance reduction that was originally proposed for finite-sum stochastic optimization (see e.g. Gower et al. (2020); Johnson & Zhang (2013); Nguyen et al. (2017)). Later on, model-free RL algorithms also used variance reduction to improve the sample efficiency. For example, it was used in learning with generative models (Sidford et al., 2018; 2023; Wainwright, 2019), policy evaluation (Du et al., 2017; Khamaru et al., 2021; Wai et al., 2019; Xu et al., 2020), offline RL (Shi et al., 2022; Yin et al., 2021), and $Q$-learning (Li et al., 2020; Zhang et al., 2020; Li et al., 2021; Yan et al., 2023).

**RL with low switching cost and batched RL**. Research in RL with low-switching cost aims to minimize the number of policy switches while maintaining comparable regret bounds to fully adaptive counterparts, and it can be applied to federated RL. In batched RL (e.g., Perchet et al. (2016), Gao et al. (2019)), the agent sets the number of batches and the length of each batch upfront, implementing an unchanged policy in a batch and aiming for fewer batches and lower regret. Bai et al. (2019) first introduced the problem of RL with low-switching cost and proposed a $Q$-learning algorithm with lazy updates, achieving $\tilde{O}(SAH^3 \log T)$ switching costs. This work was advanced by Zhang et al. (2020), which improved the regret upper bound and the switching cost. Additionally, Wang et al. (2021) studied RL under the adaptivity constraint. Recently, Qiao et al. (2022) proposed a model-based algorithm with $\tilde{O}(\log \log T)$ switching costs. Zhang et al. (2022) proposed a batched RL algorithm that is well-suited for the federated setting. Both Zheng et al. (2024b) and Zhang et al. (2025) analyzed low switching cost in the gap-dependent setting.

**Multi-agent RL (MARL) with event-triggered communications.** We review a few recent works for on-policy MARL with linear function approximations. Dubey & Pentland (2021) introduced Coop-LSVI for cooperative MARL. Min et al. (2023) proposed an asynchronous version of LSVI-UCB that originates from Jin et al. (2020), matching the same regret bound with improved communication complexity compared to Dubey & Pentland (2021). Hsu et al. (2024) developed two algorithms that incorporate randomized exploration, achieving the same regret and communication complexity as Min et al. (2023). Dubey & Pentland (2021); Min et al. (2023); Hsu et al. (2024) employed event-triggered communication conditions based on determinants of certain quantities. Different from our federated algorithm, during the synchronization in Dubey & Pentland (2021) and Min et al. (2023), local agents share original rewards or trajectories with the server. On the other hand, Hsu et al. (2024) reduces communication cost by sharing compressed statistics in the non-tabular setting with linear function approximation.

**Federated and distributed RL**. Existing literature on federated and distributed RL algorithms highlights various aspects. For value-based algorithms, Guo & Brunskill (2015), Zheng et al. (2024a), and Woo et al. (2023) focused on linear speed up. Agarwal et al. (2021) proposed a parallel RL algorithm with low communication cost. Woo et al. (2023) and Woo et al. (2024) discussed the improved covering power of heterogeneity. Wu et al. (2021) and Chen et al. (2023) worked on robustness. Particularly, Chen et al. (2023) proposed algorithms in both offline and online settings, obtaining near-optimal sample complexities and achieving superior robustness guarantees. In addition, several works have investigated value-based algorithms such as $Q$-learning in different settings, including Beikmohammadi et al. (2024), Jin et al. (2022), Khodadadian et al. (2022), Fan et al. (2023), Woo et al. (2023), and Woo et al. (2024); Anwar & Raychowdhury (2021); Zhao et al. (2023); Yang et al. (2023); Zhang et al. (2024). The convergence of decentralized temporal difference algorithms has been analyzed by Doan et al. (2019), Doan et al. (2021), Chen et al. (2021b), Sun et al. (2020), Wai (2020), Wang et al. (2020), Zeng et al. (2021), and Liu & Olshevsky (2023).

Some other works focus on policy gradient-based algorithms. Communication-efficient policy gradient algorithms have been studied by Chen et al. (2021a) and Fan et al. (2021). Lan et al. (2023) further reduces the communication complexity and also demonstrates a linear speedup in the synchronous setting. Optimal sample complexity for global convergence in federated RL, even in the presence of adversaries, is studied in Ganesh et al. (2024). Lan et al. (2024) proposes an algorithm to address the challenge of lagged policies in asynchronous settings.

The convergence of distributed actor-critic algorithms has been analyzed by Shen et al. (2023) and Chen et al. (2022). Federated actor-learner architectures have been explored by Assran et al. (2019),

Espeholt et al. (2018), and Mnih et al. (2016). Distributed inverse reinforcement learning has been examined by Banerjee et al. (2021); Gong et al. (2023); Liu & Zhu (2022; 2023; 2024; 2025).

## C  NUMERICAL EXPERIMENTS

### C.1  EXPERIMENTS COMPARING CANDIDATE ALGORITHMS

In this subsection, we conduct experiments[3] in a synthetic environment to demonstrate the better regret and communication cost of FedQ-Advantage compared to FedQ-Hoeffding and FedQ-Bernstein proposed by (Zheng et al., 2024a). We follow Zheng et al. (2024a) to use forced synchronization and generate a synthetic environment to evaluate the proposed algorithms on a tabular episodic MDP. We set $H = 10$, $S = 5$, and $A = 5$. The reward $r_h(s, a)$ for each $(s, a, h)$ is generated independently and uniformly at random from $[0, 1]$. $\mathbb{P}_h(\cdot \mid s, a)$ is generated on the $S$-dimensional simplex independently and uniformly at random for $(s, a, h)$. Under the given MDP, we set $M = 10$ and generate $10^5$ episodes for each agent, resulting in a total of $10^6$ episodes for all algorithms. For each episode, we randomly choose the initial state uniformly from the $S$ states. In FedQ-Hoeffding and FedQ-Bernstein, we use their hyper-parameter settings based on their publicly available code[4]. For FedQ-Advantage, we set $\iota = 1$ and $N_0 = 200$. To show error bars, we collect 10 sample paths for all algorithms under the same MDP environment and show the regret and communication cost in Figure 2. For both panels, the solid line represents the median of the 10 sample paths, while the shaded area shows the 10th and 90th percentiles.

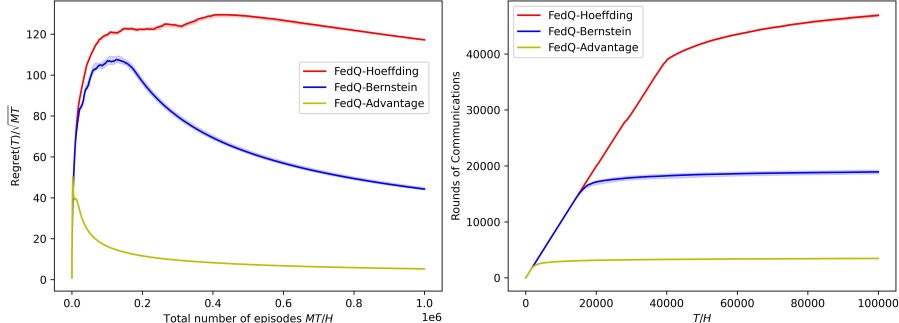

Figure 2: Numerical comparison of regrets and communication costs.

The left panel of Figure 2 plots $\text{Regret}(T)/\sqrt{MT}$ versus $MT/H$, the total number of episodes for all agents, showing the lower regret of FedQ-Advantage compared to FedQ-Hoeffding and FedQ-Bernstein. The right panel tracks the number of communication rounds throughout the learning process. All three federated algorithms show a sublinear pattern when $T$ is large, and FedQ-Advantage requires the fewest communication rounds. Since the communication cost for one synchronization is $O(MHS)$ for each of the three algorithms, FedQ-Advantage enjoys the least communication cost. These numerical results are consistent with our theoretical results in Table 1. We also provide numerical experiments to replicate the setting in Zheng et al. (2024a), show the performance on different combinations of $H, S, A$, and explore the multi-agent speedup in Appendix C.3. The conclusions are consistent with those for Figure 2.

### C.2  EXPERIMENTS FOR MULTI-AGENT SPEEDUP.

In this subsection, we provide experiments on the multi-agent speedup of FedQ-Advantage under the same experimental setting as Appendix C.1. Figure 3 reports $R(T)/\sqrt{T}$ versus $T/H$ based on the 10 sample trajectories for FedQ-Advantage and UCB-A. Here, $R(T) = \text{Regret}(T)/M$. UCB-A is our

---

[3]All the experiments are run on a server with Intel Xeon E5-2650v4 (2.2GHz) and 100 cores. Each replication is limited to a single core and 4GB RAM. The total execution time is less than 2 hours. The code for the numerical experiments is included in the supplementary materials along with the submission.

[4]`https://openreview.net/attachment?id=fe6ANBxcKM&name=supplementary_material`

single-agent counterpart from Zhang et al. (2020), and we show the experimental results for both $10^5$ episodes for the single-agent experiment and $10^6$ episodes, representing the situation where a single agent generates all episodes for FedQ-Advantage under a high communication cost. When showing $R(T)$ for UCB-A with $10^6$ episodes, we pretend that $M = 10$ and the total number of episodes is $10^5$ so that the three situations are comparable. We find that FedQ-Advantage shows a multi-agent speedup compared to UCB-A with $10^5$ episodes. However, it exhibits larger regret compared to UCB-A with $10^6$ episodes, which results from the multi-agent burn-in cost discussed in Section 4.

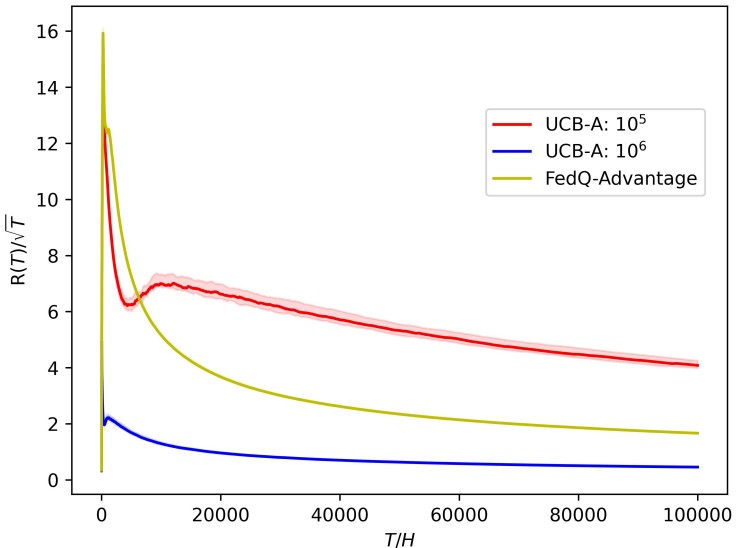

Figure 3: Multi-agent speedup

### C.3    OTHER COMBINATIONS OF $H, S, A$

Figure 3 gives additional numerical experiment under different combinations of $H, S, A$. The top panels show 10 replications on $S = 3, A = 2, H = 5$, which replicates the experiments in Zheng et al. (2024a). The second one chooses relatively large parameters where $H = 20, S = 20, A = 5$ and shows one replication. The conclusion is the same as that in Appendix C.1.

## D    BASIC FACTS AND CONCENTRATION INEQUALITIES

In this section, we provide some basic facts and lemmas of concentration inequalities for FedQ-Advantage. For any triple $(s, a, h) \in \mathcal{S} \times \mathcal{A} \times [H]$, we let $Y_h^t(s, a) = \sum_{m,k:k \leq K} n_h^{m,k} \mathbb{I}[t_h^k \leq t]$ be the number of visits to $(s, a, h)$ up to and including stage $t$ and $y_h^t(s, a) = \sum_{m,k:k \leq K} n_h^{m,k} \mathbb{I}[t_h^k = t]$ be the visiting number in stage $t$. Here, we have that $Y_h^0 = y_h^0 = 0, N_h^k = Y_h^{t_h^k - 1}, n_h^k = y_h^{t_h^k - 1}$. We also denote $T_h(s, a) = t_h^K(s, a)$ as the total number of stages for $(s, a, h)$. Here, we emphasize that the stage renewal condition might not be triggered in FedQ-Advantage for the last stage $T_h(s, a)$.

Next, we assign an order to the visits of any $(s, a, h)$. Let $L_i(s, a, h)$ denote the $i$-th visit to $(s, a, h)$ in FedQ-Advantage for $i \in \mathbb{N}_+$, and $(k_{L_i}(s, a, h), m_{L_i}(s, a, h), j_{L_i}(s, a, h))$ be the corresponding (round, agent, episode) index of the $i$-th visit. Similarly, let $l_i(s, a, h, k), i \in [n_h^k(s, a)]$ denote the $i$-th visit to $(s, a, h)$ during the stage $t_h^k(s, a) - 1$, and $(k_{l_i}(s, a, h, k), m_{l_i}(s, a, h, k), j_{l_i}(s, a, h, k))$ be the corresponding (round, agent, episode) index of the $i$-th visit $l_i(s, a, h, k)$. The indices follow the chronological order of the visits. Specifically, under the synchronization assumption $n^{m,k} = n^k, \forall m \in [M]$, a viable order can be determined by the "round index first, episode index second, agent

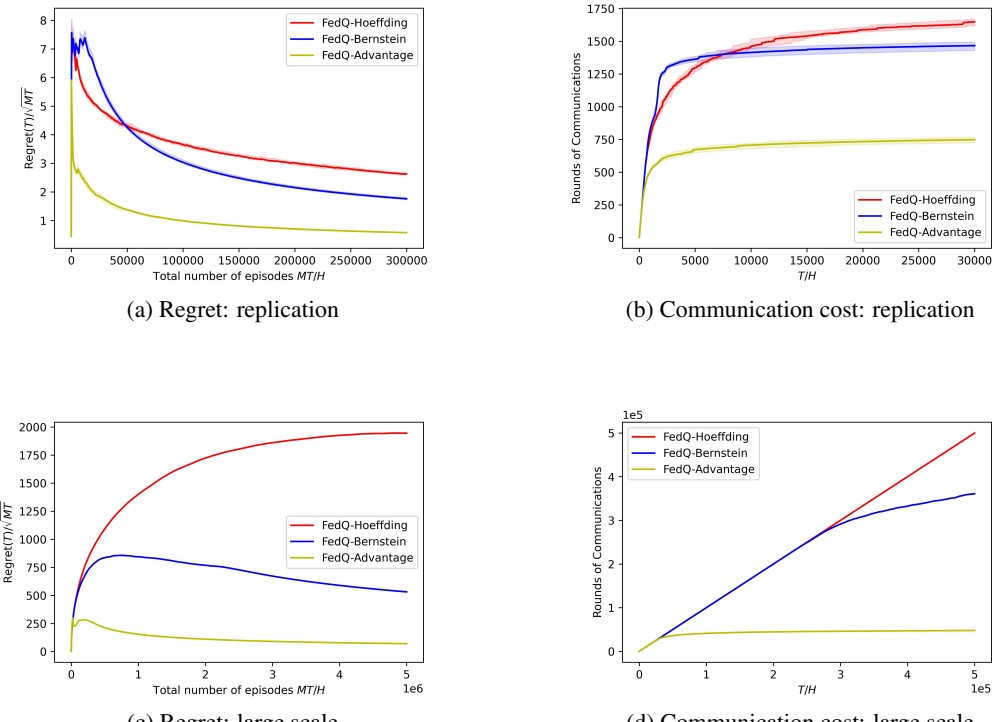

Figure 4: Additional Experiments. The upper panels (a) and (b) replicate the numerical setting in Zheng et al. (2024a) where $S = 3, A = 2, H = 5$. The bottom panels (c) and (d) perform experiments with $H = 20, S = 20, A = 5$.

index third" rule. When $(s, a, h, k)$ is clear from the context, we use $L_i, l_i, (k, m, j)_{L_i}, (k, m, j)_{l_i}$ for simplicity.

Next, we provide Lemma D.1 on some basic relationships for quantities in FedQ-Advantage.

**Lemma D.1.** *For any* $(s, a, h, k) \in \mathcal{S} \times \mathcal{A} \times [H] \times [K]$, *the following relationships hold for FedQ-Advantage.*

(a) $T_0 \leq \hat{T}$.

(b) $n_h^{m,k}(s, a) \leq c_h^k(s, a), \forall m \in [M]$. *In addition,* $\forall k \in [K]$, *there exists* $(s, a, h, m) \in \mathcal{S} \times \mathcal{A} \times [H] \times [M]$ *such that* $n_h^{m,k}(s, a) = c_h^k(s, a)$.

(c) *If* $T_h(s, a) \geq 2$, *we have* $MH \leq Y_h^1(s, a) \leq MH + M$.

(d) *If* $n_h^k(s, a) > 0$, *we have that* $\hat{n}_h^{k+1}(s, a) \leq (1 + 2/H)n_h^k(s, a), \forall k \in [K]$.

(e) $1 + \frac{1}{H} \leq \frac{y_h^{t+1}(s,a)}{y_h^t(s,a)} \leq 1 + \frac{2}{H}, \forall t \in [1, T_h(s, a) - 2], t \in \mathbb{N}$. *In addition,* $0 \leq \frac{y_h^{T_h(s,a)}(s,a)}{y_h^{T_h-1}(s,a)} \leq 1 + \frac{2}{H}$.

(f) $1 \leq \frac{Y_h^{t+1}(s,a)}{Y_h^t(s,a)} \leq 2 + \frac{2}{H} \leq 4, \forall t \in [T_h(s, a) - 1]$.

(g) *The following relationships hold.*

$$\frac{Y_h^t(s, a)}{y_h^t(s, a)} \leq 2H, \forall t \in [T_h(s, a) - 1]. \tag{15}$$

$$\sqrt{y_h^t(s,a)} \le 3\sqrt{H}\left(\sqrt{Y_h^t(s,a)} - \sqrt{Y_h^{t-1}(s,a)}\right), \forall t \in [T_h(s,a) - 1]. \qquad (16)$$

$$\frac{Y_h^t(s,a)}{y_h^t(s,a)} \ge \frac{H}{4}, \forall t \in [H, T_h(s,a)], t \in \mathbb{N}. \qquad (17)$$

(h) $\sum_{s,a,h} y_h^{T_h(s,a)-1}(s,a) \le 9MSAH(H+1) + \frac{4T_0}{H}$. $\sum_{s,a,h} y_h^{T_h(s,a)}(s,a) \le 9MSAH(H+1) + \frac{4T_0}{H}$.

(i) Denote $T_1 = (2 + \frac{2}{H})T_0 + MSAH(H+1)$, we have

$$\hat{T} \le T_1 \le (2 + \frac{2}{H})\hat{T} + MSAH(H+1).$$

(j) $Q_h^{k+1}(s,a) \le Q_h^k(s,a), V_h^{k+1}(s) \le V_h^k(s), V_h^{\mathrm{ref},k+1}(s) \le V_h^{\mathrm{ref},k}(s), \forall (s,a,h,k) \in \mathcal{S} \times \mathcal{A} \times [H] \times [K-1]$.

*Here, we use $\sum_{s,a}$ as a simplified notation for $\sum_{s\in\mathcal{S}}\sum_{a\in\mathcal{A}}$ and $\sum_{s,a,h}$ as a simplified notation for $\sum_{s\in\mathcal{S}}\sum_{a\in\mathcal{A}}\sum_{h=1}^{H}$. $T_h$ is the simplified notation of $T_h(s,a)$. Those simplifications will also be used later.*

*Proof of Lemma D.1.*      (a) This relationship holds from the stopping condition of the loop (line 2) in Algorithm 1.

(b) From the triggering condition for terminating the exploration in a round (line 4 in Algorithm 2), we can prove the relationship.

(c) Since $T_h(s,a) \ge 2$, there exists a round $k$ satisfying $t_h^k(s,a) = 1$ and $t_h^{k+1}(s,a) = 2$. Then according to Equation (7), we have $Y_h^1(s,a) = \hat{n}_h^{k+1}(s,a) \ge MH$. Meanwhile, according to (b), we have:

$$Y_h^1(s,a) = \hat{n}_h^{k+1}(s,a) = \tilde{n}_h^k(s,a) + \sum_{m=1}^{M} n_h^{m,k}(s,a)$$
$$\le \tilde{n}_h^k(s,a) + Mc_h^k(s,a) \le \tilde{n}_h^k(s,a) + M.$$

Since round $k$ is in stage 1, we know stage 1 is not renewed before the start of round $k$. Then according to Equation (7) and the definition of $\tilde{n}_h^k(s,a)$, we have $\tilde{n}_h^k(s,a) \le MH$ and $Y_h^1(s,a) \le \tilde{n}_h^k(s,a) + M = MH + M$.

(d) According to Equation (7) and the definition of $\tilde{n}_h^k(s,a)$, we have

$$\tilde{n}_h^k(s,a) \le (1 + 1/H)n_h^k(s,a).$$

If $0 \le \tilde{n}_h^k(s,a) \le (1 - \frac{1}{H})n_h^k(s,a)$, then according to Equation (2) we have:

$$\hat{n}_h^{k+1}(s,a) \le \tilde{n}_h^k(s,a) + Mc_h^k(s,a) \le \tilde{n}_h^k(s,a) + M\left(\frac{n_h^k(s,a) - \tilde{n}_h^k(s,a)}{M} + 1\right)$$
$$= n_h^k(s,a) + M.$$

Since $n_h^k(s,a) > 0$, we know $t_h^k(s,a) > 1$ and then $n_h^k(s,a) \ge Y_h^1(s,a) \ge MH$. Therefore, $\hat{n}_h^{k+1}(s,a) \le n_h^k(s,a) + M \le (1 + \frac{2}{H})n_h^k(s,a)$.

If $(1 - 1/H)n_h^k(s,a) < \tilde{n}_h^k(s,a) \le (1 + 1/H)n_h^k(s,a)$, then according to Equation (2) we have:

$$\hat{n}_h^{k+1}(s,a) \le \tilde{n}_h^k(s,a) + Mc_h^k(s,a) \le \tilde{n}_h^k(s,a) + M \cdot \frac{1}{MH}n_h^k(s,a) \le (1 + \frac{2}{H})n_h^k(s,a).$$

Therefore, $\tilde{n}_h^k(s,a) \le (1 + \frac{2}{H})n_h^k(s,a)$.

(e) For $t \le T_h(s,a) - 2$, there exists a round $k$ satisfying $t_h^k(s,a) = t+1$ and $t_h^{k+1}(s,a) = t+2$. Then according to Equation (7), we have $y_h^{t+1}(s,a) = \hat{n}_h^{k+1}(s,a) \ge (1 + 1/H)n_h^k(s,a) = (1 + 1/H)y_h^t(s,a)$. Moreover, according to (d), we have $y_h^{t+1}(s,a) = \hat{n}_h^{k+1}(s,a) \le (1 + 2/H)n_h^k(s,a) = (1 + 2/H)y_h^t(s,a)$.

For $t = T_h(s,a) - 1$, we have $y_h^{T_h}(s,a) = \hat{n}_h^{K+1}(s,a) \le (1 + 2/H)n_h^K(s,a) = (1 + 2/H)y_h^{T_h - 1}(s,a)$.

(f) According to (e), for $t \le T_h(s,a) - 1$ we have $y_h^{t+1}(s,a) \le (1 + 2/H)y_h^t(s,a)$. Then:

$$1 \le \frac{Y_h^{t+1}(s,a)}{Y_h^t(s,a)} = 1 + \frac{y_h^{t+1}(s,a)}{Y_h^t(s,a)} \le 1 + \frac{(1 + \frac{2}{H})y_h^t(s,a)}{Y_h^t(s,a)} \le 2 + \frac{2}{H} \le 4.$$

(g) We will use the mathematical induction to prove the Equation (15).

For $t = 1$,
$$\frac{Y_h^1(s,a)}{y_h^1(s,a)} = 1 \le 2H.$$

If $\frac{Y_h^{t-1}(s,a)}{y_h^{t-1}(s,a)} \le 2H$, then for $t$ ($2 \le t \le T_h(s,a) - 1$), according to (e), we have $y_h^t(s,a) \ge (1 + 1/H)y_h^{t-1}(s,a)$. Then:

$$\frac{Y_h^t(s,a)}{y_h^t(s,a)} = 1 + \frac{Y_h^{t-1}(s,a)}{y_h^t(s,a)} \le 1 + \frac{Y_h^{t-1}(s,a)}{(1 + \frac{1}{H})y_h^{t-1}(s,a)} \le 1 + \frac{2H}{1 + \frac{1}{H}} \le 2H.$$

Therefore, we finish the proof of the Equation (15).

For Equation (16), we have:

$$3\sqrt{H}\left(\sqrt{Y_h^t(s,a)} - \sqrt{Y_h^{t-1}(s,a)}\right)$$
$$= \sqrt{y_h^t(s,a)} \cdot 3\sqrt{H}\left(\sqrt{\frac{Y_h^t(s,a)}{y_h^t(s,a)}} - \sqrt{\frac{Y_h^t(s,a)}{y_h^t(s,a)} - 1}\right)$$
$$= \sqrt{y_h^t(s,a)} \cdot 3\sqrt{H}\frac{1}{\sqrt{\frac{Y_h^t(s,a)}{y_h^t(s,a)}} + \sqrt{\frac{Y_h^t(s,a)}{y_h^t(s,a)} - 1}}$$
$$\ge \sqrt{y_h^t(s,a)} \cdot 3\sqrt{H}\frac{1}{\sqrt{2H} + \sqrt{2H - 1}} \ge \sqrt{y_h^t(s,a)}.$$

The second last inequality is because $\frac{Y_h^t(s,a)}{y_h^t(s,a)} \le 2H$ according to Equation (15).

For Equation (17), according to (e), we have $y_h^t(s,a) \le (1 + 2/H)^{h-1}y_h^{t-h+1}(s,a)$ for any $h \in [H]$. Then:

$$\frac{Y_h^t(s,a)}{y_h^t(s,a)} \ge \frac{\sum_{h=1}^H y_h^{t-h+1}(s,a)}{y_h^t(s,a)} \ge \frac{\sum_{h=1}^H (1 + \frac{2}{H})^{1-h}y_h^t(s,a)}{y_h^t(s,a)}$$
$$= \frac{H}{2}(1 + \frac{2}{H})(1 - (1 + \frac{2}{H})^{-H}).$$

Because $(1 + \frac{2}{H})^H$ is increasing in $H$, we have $(1 + \frac{2}{H})^H \ge 3$ and $1 - (1 + \frac{2}{H})^{-H} \ge \frac{2}{3}$. Therefore,

$$\frac{Y_h^t(s,a)}{y_h^t(s,a)} \ge \frac{H}{2}(1 + \frac{2}{H})(1 - (1 + \frac{2}{H})^{-H}) \ge \frac{H}{4}.$$

We finish the proof of (g).

(h)
$$\sum_{s,a,h} y_h^{T_h(s,a)-1}(s,a) = \sum_{s,a,h} y_h^{T_h-1}(s,a)\mathbb{I}[T_h(s,a) < H + 1]$$
$$+ \sum_{s,a,h} y_h^{T_h-1}(s,a)\mathbb{I}[T_h(s,a) \ge H + 1]$$

Because of (e), we have:

$$y_h^{T_h-1}(s,a)\mathbb{I}[T_h(s,a) < H+1] \le (1+\frac{2}{H})^{T_h-1}y_h^1(s,a)\mathbb{I}[T_h(s,a) < H+1]$$

$$\le (1+\frac{2}{H})^H(MH+M).$$

The last inequality is because $y_h^1(s,a) = Y_h^1(s,a) \le MH+M$ according to (c). Moreover, according to Equation (17), we have

$$\sum_{s,a,h} y_h^{T_h-1}(s,a)\mathbb{I}[T_h(s,a) \ge H+1] \le \sum_{s,a,h} \frac{4}{H}Y_h^{T_h-1}(s,a)\mathbb{I}[T_h(s,a) \ge H+1] \le \frac{4T_0}{H}.$$

The last inequality holds because $\sum_{s,a,h} Y_h^{T_h-1}(s,a) \le T_0$ according to the algorithm. Therefore, we have

$$\sum_{s,a,h} y_h^{T_h(s,a)-1}(s,a) \le \sum_{s,a,h}(1+\frac{2}{H})^H(MH+M) + \frac{4T_0}{H} \le 9MSAH(H+1) + \frac{4T_0}{H}.$$

Similarly, we have:

$$\sum_{s,a,h} y_h^{T_h(s,a)}(s,a)$$

$$= \sum_{s,a,h} y_h^{T_h}(s,a)\mathbb{I}[T_h(s,a) < H+1] + \sum_{s,a,h} y_h^{T_h-1}(s,a)\mathbb{I}[T_h(s,a) \ge H+1]$$

$$\le (1+\frac{2}{H})^{T_h-1}y_h^1(s,a)\mathbb{I}[T_h(s,a) < H+1] + \sum_{s,a,h} \frac{4}{H}Y_h^{T_h-1}(s,a)\mathbb{I}[T_h(s,a) \ge H+1]$$

$$\le 9MSAH(H+1) + \frac{4T_0}{H}.$$

(i) First, according to (f), we have:

$$Y_h^{T_h}(s,a) \le Y_h^1(s,a)\mathbb{I}[T_h(s,a) = 1] + (2+\frac{2}{H})Y_h^{T_h-1}(s,a)\mathbb{I}[T_h(s,a) > 1]$$

$$\le (2+\frac{2}{H})Y_h^{T_h-1}(s,a) + MH+M.$$

The last inequality is because of (c). Then we have:

$$\hat{T} = \sum_{s,a,h} Y_h^{T_h}(s,a) \le \sum_{s,a,h}\left((2+\frac{2}{H})Y_h^{T_h-1}(s,a) + MH+M\right)$$

$$\le (2+\frac{2}{H})T_0 + MSAH(H+1).$$

The last inequality is because of $\sum_{s,a,h} Y_h^{T_h-1}(s,a) \le T_0$ according to the algorithm. Because $T_0 \le \hat{T}$ from (a), we have:

$$\hat{T} \le T_1 \le (2+\frac{2}{H})\hat{T} + MSAH(H+1).$$

(j) According to Equation (10), we know $Q_h^k(s,a)$ is non-increasing with respect to $k$. Then based on the update rule Equation (13) for $V_h^k(s,a)$ and the update rule Equation (14) for $V_h^{\text{ref},k}(s,a)$, we find that they are also non-increasing with respect to $k$.

$\square$

Next, we provide Lemma D.2 that discusses the weighted sum of all the steps.

**Lemma D.2.** *For any non-negative weight sequence $\{\omega_h(s,a)\}_{s,a,h}$ and any $\alpha \in (0,1)$, it holds that*

$$\sum_{h=1}^{H} \sum_{k,m,j} \frac{\omega_h(s_h^{k,m,j}, a_h^{k,m,j})}{N_h^k(s_h^{k,m,j}, a_h^{k,m,j})^\alpha} \mathbb{I}\left[t_h^{k,m,j} > 1\right] \le \frac{4^\alpha}{1-\alpha} \sum_{s,a,h} \omega_h(s,a) Y_h^{T_h}(s,a)^{1-\alpha},$$

*and*

$$\sum_{h=1}^{H} \sum_{k,m,j} \frac{\omega_h(s_h^{k,m,j}, a_h^{k,m,j})}{n_h^k(s_h^{k,m,j}, a_h^{k,m,j})^\alpha} \mathbb{I}\left[t_h^{k,m,j} > 1\right] \le \frac{(8H)^\alpha}{1-\alpha} \sum_{s,a,h} \omega_h(s,a) Y_h^{T_h}(s,a)^{1-\alpha}.$$

*For $\alpha = 1$, it holds that*

$$\sum_{h=1}^{H} \sum_{k,m,j} \frac{\omega_h(s_h^{k,m,j}, a_h^{k,m,j})}{N_h^k(s_h^{k,m,j}, a_h^{k,m,j})} \mathbb{I}\left[t_h^{k,m,j} > 1\right] \le 4 \sum_{s,a,h} \omega_h(s,a) \log(Y_h^{T_h}(s,a)),$$

*and*

$$\sum_{h=1}^{H} \sum_{k,m,j} \frac{\omega_h(s_h^{k,m,j}, a_h^{k,m,j})}{n_h^k(s_h^{k,m,j}, a_h^{k,m,j})} \mathbb{I}\left[t_h^{k,m,j} > 1\right] \le 8H \sum_{s,a,h} \omega_h(s,a) \log(Y_h^{T_h}(s,a)).$$

*Here, $t_h^{k,m,j} = t_h^k(s_h^{k,m,j}, a_h^{k,m,j})$.*

*Proof of Lemma D.2.* According to Equation (15), for any $(s_h^{k,m,j}, a_h^{k,m,j}, h) \in \mathcal{S} \times \mathcal{A} \times [H]$ and $t_h^{k,m,j} > 1$,

$$\frac{N_h^k(s_h^{k,m,j}, a_h^{k,m,j})}{n_h^k(s_h^{k,m,j}, a_h^{k,m,j})} = \frac{Y_h^{t_h^{k,m,j}-1}(s_h^{k,m,j}, a_h^{k,m,j})}{y_h^{t_h^{k,m,j}-1}(s_h^{k,m,j}, a_h^{k,m,j})} \le 2H.$$

Therefore, we only need to prove the first and the third inequalities.

We first provide two conclusions. For $1 \le x \le 4$ and $0 < \alpha < 1$, it holds:

$$x^{1-\alpha} - 1 \ge (1-\alpha)(x-1)x^{-\alpha} \ge 4^{-\alpha}(1-\alpha)(x-1) \tag{18}$$

$$\log(x) \ge \frac{x-1}{x} \ge \frac{x-1}{4}. \tag{19}$$

Next, we go back to the proof. For any $(s,a,h) \in \mathcal{S} \times \mathcal{A} \times [H]$ and $2 \le t \le T_h(s,a)$, let $x = \frac{Y_h^t(s,a)}{Y_h^{t-1}(s,a)}$. According to (f) in Lemma D.1, we have $1 \le x \le 4$. Using Equation (18), and Equation (19), it holds:

$$Y_h^t(s,a)^{1-\alpha} - Y_h^{t-1}(s,a)^{1-\alpha} \ge 4^{-\alpha}(1-\alpha)\frac{y_h^t(s,a)}{Y_h^{t-1}(s,a)^\alpha}, \tag{20}$$

and

$$\log(Y_h^t(s,a)) - \log(Y_h^{t-1}(s,a)) \ge \frac{y_h^t(s,a)}{4Y_h^{t-1}(s,a)}. \tag{21}$$

Now,

$$\sum_{h=1}^{H} \sum_{k,m,j} \frac{\omega_h(s_h^{k,m,j}, a_h^{k,m,j})}{N_h^k(s_h^{k,m,j}, a_h^{k,m,j})^\alpha} \mathbb{I}\left[t_h^{k,m,j} > 1\right]$$

$$= \sum_{h=1}^{H} \sum_{k,m,j} \frac{\omega_h(s_h^{k,m,j}, a_h^{k,m,j})}{N_h^k(s_h^{k,m,j}, a_h^{k,m,j})^\alpha} \mathbb{I}\left[t_h^{k,m,j} > 1\right] \left(\sum_{s,a} \mathbb{I}\left[(s_h^{k,m,j}, a_h^{k,m,j}) = (s,a)\right]\right)$$

$$= \sum_{s,a,h} \sum_{k,m,j} \frac{\omega_h(s,a)}{N_h^k(s,a)^\alpha} \mathbb{I}\left[t_h^k(s,a) > 1\right] \mathbb{I}\left[(s_h^{k,m,j}, a_h^{k,m,j}) = (s,a)\right]$$

$$= \sum_{s,a,h} \sum_{k,m,j} \frac{\omega_h(s,a)}{N_h^k(s,a)^\alpha} \mathbb{I}\left[(s_h^{k,m,j}, a_h^{k,m,j}) = (s,a)\right] \left(\sum_{t=2}^{T_h} \mathbb{I}\left[t_h^k(s,a) = t\right]\right)$$

$$= \sum_{s,a,h} \sum_{t=2}^{T_h} \frac{\omega_h(s,a)}{N_h^k(s,a)^\alpha} \sum_{k,m,j} \mathbb{I}\left[(s_h^{k,m,j}, a_h^{k,m,j}) = (s,a), t_h^k(s,a) = t\right]$$

In the last equality, $\mathbb{I}[(s_h^{k,m,j}, a_h^{k,m,j}) = (s,a), t_h^k(s,a) = t] = 1$ if and only if $(s_h^{k,m,j}, a_h^{k,m,j}) = (s,a)$ and $k$ in stage $t$ of $(s,a,h)$, so $\sum_{k,m,j} \mathbb{I}[(s_h^{k,m,j}, a_h^{k,m,j}) = (s,a), t_h^k(s,a) = t] = y_h^t(s,a)$. In this case, $N_h^k(s,a) = Y_h^{t-1}(s,a)$ and then we have :

$$\sum_{h=1}^{H} \sum_{k,m,j} \frac{\omega_h(s_h^{k,m,j}, a_h^{k,m,j})}{N_h^k(s_h^{k,m,j}, a_h^{k,m,j})^\alpha} \mathbb{I}\left[t_h^{k,m,j} > 1\right] = \sum_{s,a,h} \sum_{t=2}^{T_h} \frac{\omega_h(s,a) y_h^t(s,a)}{Y_h^{t-1}(s,a)^\alpha}. \tag{22}$$

Summing Equation (20) for $2 \le t \le T_h(s,a)$, for any $0 < \alpha < 1$, we have:

$$\sum_{t=2}^{T_h} \frac{y_h^t(s,a)}{Y_h^{t-1}(s,a)^\alpha} \le \frac{4^\alpha}{1-\alpha} \sum_{t=2}^{T_h} (Y_h^t(s,a)^{1-\alpha} - Y_h^{t-1}(s,a)^{1-\alpha}) \le \frac{4^\alpha}{1-\alpha} Y_h^{T_h}(s,a)^{1-\alpha} \tag{23}$$

Combining Equation (22) and Equation (23), we can finish the proof of the first inequality.

In Equation (22), let $\alpha = 1$, we have:

$$\sum_{h=1}^{H} \sum_{k,m,j} \frac{\omega_h(s_h^{k,m,j}, a_h^{k,m,j})}{N_h^k(s_h^{k,m,j}, a_h^{k,m,j})} \mathbb{I}\left[t_h^{k,m,j} > 1\right] = \sum_{s,a,h} \omega_h(s,a) \sum_{t=1}^{T_h} \frac{y_h^t(s,a)}{Y_h^{t-1}(s,a)} \tag{24}$$

Summing Equation (21) for $2 \le t \le T_h(s,a)$ , we have:

$$\sum_{t=2}^{T_h} \frac{y_h^t(s,a)}{Y_h^{t-1}(s,a)} \le 4 \sum_{t=2}^{T_h} \left(\log(Y_h^t(s,a)) - \log(Y_h^{t-1}(s,a))\right) \le 4\log(Y_h^{T_h}(s,a)) \tag{25}$$

Combining Equation (24) with Equation (25), we finish the proof of the third inequality. Then we finish the proof. $\qquad\square$

Next, we provide auxiliary lemmas.

**Lemma D.3.** (Azuma-Hoeffding Inequality) *Suppose $\{X_k\}_{k=0}^{\infty}$ is a martingale and $|X_k - X_{k-1}| \le c_k, \forall k \in \mathbb{N}_+$ almost surely. Then for any positive integers $N$ and any positive real number $\epsilon$, it holds that:*

$$\mathbb{P}\left(|X_N - X_0| \ge \epsilon\right) \le 2\exp\left(-\frac{\epsilon^2}{2\sum_{k=1}^{N} c_k^2}\right).$$

**Lemma D.4.** (Lemma 10 of Zhang et al. (2020)) *Let $\{M_n\}_{n=0}^{\infty}$ be a martingale such that $M_0 = 0$ and $|M_n - M_{n-1}| \le c$. Let $Var_n = \sum_{k=1}^{n} \mathbb{E}[(M_k - M_{k-1})^2 | \mathcal{F}_{k-1}]$, where $\mathcal{F}_{k-1} = \sigma(M_0, M_1, ..., M_{k-1})$. Then for any positive integer $n$ and any $\epsilon, p > 0$, we have that:*

$$\mathbb{P}\left(|M_n| \ge 2\sqrt{Var_n \log(1/p)} + 2\sqrt{\epsilon \log(1/p)} + 2c\log(1/p)\right) \le \left(2nc^2/\epsilon + 2\right) p.$$

At the end of this section, we provide a lemma of concentration inequalities.

**Lemma D.5.** *Let $\iota = \log(2/p)$ with $p \in (0,1)$. Using $\forall(s,a,h,k)$ as the simplified notation for $\forall(s,a,h,k) \in \mathcal{S} \times \mathcal{A} \times [H] \times [K]$. For any function $f : \mathcal{S} \to \mathbb{R}$, we denote $\mathbb{V}_{s,a,h}(f) = \mathbb{P}_{s,a,h}f^2 - (\mathbb{P}_{s,a,h}f)^2$. Next, we define the following events.*

$$\mathcal{E}_1 = \left\{ \frac{1}{n_h^k} \left| \sum_{i=1}^{n_h^k} \left( V_{h+1}^\star(s_{h+1}^{(k,m,j)_{l_i}}) - \mathbb{P}_{s,a,h}V_{h+1}^\star \right) \right| \leq \sqrt{\frac{2H^2\iota}{n_h^k}}, \forall(s,a,h,k) \right\}.$$

$$\mathcal{E}_2 = \left\{ |\chi_1| \leq \frac{2}{N_h^k} \left( \sqrt{\sum_{i=1}^{N_h^k} \mathbb{V}_{s,a,h}(V_{h+1}^{\mathrm{ref},k_{L_i}})\iota} + \frac{\sqrt{\iota}}{T_1} + H\iota \right), \forall(s,a,h,k) \right\},$$

*in which $\chi_1$ is the abbreviation for*

$$\chi_1(s,a,h,k) = \frac{1}{N_h^k(s,a)} \sum_{i=1}^{N_h^k} \left( \mathbb{P}_{s,a,h} - \mathbb{1}_{s_{h+1}^{(k,m,j)_{L_i}}} \right) V_{h+1}^{\mathrm{ref},k_{L_i}}.$$

$$\mathcal{E}_3 = \left\{ \left| \sum_{i=1}^{N_h^k} \left( \mathbb{P}_{s,a,h} - \mathbb{1}_{s_{h+1}^{(k,m,j)_{L_i}}} \right) V_{h+1}^{\mathrm{ref},k_{L_i}} \right| \leq H\sqrt{2N_h^k(s,a)\iota}, \forall(s,a,h,k) \right\}.$$

$$\mathcal{E}_4 = \left\{ \left| \sum_{i=1}^{N_h^k} \left( \mathbb{P}_{s,a,h} - \mathbb{1}_{s_{h+1}^{(k,m,j)_{L_i}}} \right) (V_{h+1}^{\mathrm{ref},k_{L_i}})^2 \right| \leq H^2\sqrt{2N_h^k(s,a)\iota}, \forall(s,a,h,k) \right\}.$$

$$\mathcal{E}_5 = \left\{ |\chi_2| \leq \frac{2}{n_h^k} \left( \sqrt{\sum_{i=1}^{n_h^k} \mathbb{V}_{s,a,h}(V_{h+1}^{k_{l_i}} - V_{h+1}^{\mathrm{ref},k_{l_i}})\iota} + \frac{\sqrt{\iota}}{T_1} + 2H\iota \right), \forall(s,a,h,k) \right\},$$

*in which $\chi_2$ is the abbreviation for*

$$\chi_2(s,a,h,k) = \frac{1}{n_h^k} \sum_{i=1}^{n_h^k} \left( \mathbb{P}_{s,a,h} - \mathbb{1}_{s_{h+1}^{(k,m,j)_{l_i}}} \right) (V_{h+1}^{k_{l_i}} - V_{h+1}^{\mathrm{ref},k_{l_i}}).$$

$$\mathcal{E}_6 = \left\{ \left| \sum_{i=1}^{n_h^k} \left( \mathbb{P}_{s,a,h} - \mathbb{1}_{s_{h+1}^{(k,m,j)_{l_i}}} \right) (V_{h+1}^{k_{l_i}} - V_{h+1}^{\mathrm{ref},k_{l_i}}) \right| \leq 2H\sqrt{2n_h^k(s,a)\iota}, \forall(s,a,h,k) \right\}.$$

$$\mathcal{E}_7 = \left\{ \left| \sum_{i=1}^{n_h^k} \left( \mathbb{P}_{s,a,h} - \mathbb{1}_{s_{h+1}^{(k,m,j)_{l_i}}} \right) (V_{h+1}^{k_{l_i}} - V_{h+1}^{\mathrm{ref},k_{l_i}})^2 \right| \leq 2H^2\sqrt{2n_h^k(s,a)\iota}, \forall(s,a,h,k) \right\}.$$

$$\mathcal{E}_8 = \left\{ \left| \sum_{h=1}^{H} \sum_{k,m,j} \left( \mathbb{P}_{s_h^{k,m,j},a_h^{k,m,j},h} - \mathbb{1}_{s_{h+1}^{k,m,j}} \right) \lambda_{h+1}^k(s_{h+1}^{k,m,j}) \right| \leq \sqrt{2T_1\iota} \right\}.$$

Here, $\lambda_h^k(s) = \mathbb{I}[N_h^k(s) < N_0]$ with $N_h^k(s) = \sum_{a \in \mathcal{A}} N_h^k(s,a)$. Especially, $\lambda_{H+1}^k(s) = 0$. $\sum_{k,m,j}$ is the abbreviation of $\sum_{k=1}^{K} \sum_{m=1}^{M} \sum_{j=1}^{n^{m,k}}$. We will also use the abbreviation later.

$$\mathcal{E}_9 = \left\{ \sum_{h=1}^{H} \sum_{k,m,j} (1 + \frac{2}{H})^{h-1}(1 + \frac{1}{H}) \left( \mathbb{P}_{s_h^{k,m,j},a_h^{k,m,j},h} - \mathbb{1}_{s_{h+1}^{k,m,j}} \right) (V_{h+1}^k - V_{h+1}^\star) \right.$$

$$\left. \leq 18H\sqrt{2T_1\iota} \right\}$$

$$\mathcal{E}_{10} = \left\{ |V(s,a,h,t)| \leq H\sqrt{2y_h^t(s,a)\iota}, \forall(s,a,h) \text{ and } \forall t \in [T_h(s,a)] \right\}.$$

*Here,*

$$V(s, a, h, t) = \sum_{k,m,j} \left( \mathbb{P}_{s,a,h} - \mathbb{1}_{s_{h+1}^{k,m,j}} \right) (V_{h+1}^k - V_{h+1}^\star) \, \mathbb{I} \left[ (s_h^{k,m,j}, a_h^{k,m,j}) = (s,a), t_h^k(s,a) = t \right].$$

$$\mathcal{E}_{11} = \left\{ \sum_{h=1}^H \sum_{k,m,j} (1 + \frac{2}{H})^{h-1} \mathbb{I} \left[ t_h^{k,m,j} > 1 \right] \left( \mathbb{P}_{s_h^{k,m,j}, a_h^{k,m,j}, h} - \mathbb{1}_{s_{h+1}^{k,m,j}} \right) \left( V_{h+1}^\star - V_{h+1}^{\pi^k} \right) \right.$$

$$\left. \leq 9H\sqrt{2T_1 \iota} \right\}.$$

$$\mathcal{E}_{12} = \left\{ \frac{1}{N_h^k(s,a)} \left| \sum_{i=1}^{N_h^k(s,a)} \left( \mathbb{P}_{s,a,h} - \mathbb{1}_{s_{h+1}^{(k,m,j)_{L_i}}} \right) \lambda_{h+1}^{k_{L_i}} \right| \leq \sqrt{\frac{2\iota}{N_h^k(s,a)}}, \forall(s,a,h,k) \right\}.$$

$$\mathcal{E}_{13} = \left\{ \left| \sum_{h=1}^H \sum_{k,m,j} \left( \mathbb{P}_{s_h^{k,m,j}, a_h^{k,m,j}, h}(V_{h+1}^\star) - V_{h+1}^\star(s_{h+1}^{k,m,j}) \right) \right| \leq H\sqrt{2T_1 \iota} \right\}.$$

$$\mathcal{E}_{14} = \left\{ \left| \sum_{h=1}^H \sum_{k,m,j} \left( \mathbb{P}_{s_h^{k,m,j}, a_h^{k,m,j}, h}(V_{h+1}^\star)^2 - V_{h+1}^\star(s_{h+1}^{k,m,j})^2 \right) \right| \leq H^2\sqrt{2T_1 \iota} \right\}.$$

$$\mathcal{E}_{15} = \left\{ \left| \frac{1}{N_h^k(s,a)} \sum_{i=1}^{N_h^k(s,a)} \left( \mathbb{P}_{s,a,h} - \mathbb{1}_{s_{h+1}^{(k,m,j)_{L_i}}} \right) \left( V_{h+1}^{\text{ref},k_{L_i}} - V_{h+1}^\star \right) \right| \leq 2H\sqrt{\frac{2\iota}{N_h^k(s,a)}} \right\}.$$

*Then we have*

$$\mathbb{P}(\mathcal{E}_i) \geq 1 - SAT_1^2/Hp, i \in \{1,6,7,10\},$$
$$\mathbb{P}(\mathcal{E}_i) \geq 1 - SAT_1 p, i \in \{3,4,12,15\},$$
$$\mathbb{P}(\mathcal{E}_i) \geq 1 - p, i \in \{8,9,11,13,14\},$$
$$\mathbb{P}(\mathcal{E}_2) \geq 1 - SAT_1(HT_1^3 + 1)p$$

*and*

$$\mathbb{P}(\mathcal{E}_5) \geq 1 - SAT_1^2(4HT_1^3 + 1)/Hp.$$

*Proof of Lemma D.5.* First, we will prove with probability at least $1 - SAT_1^2/Hp$, $\mathcal{E}_1$ holds. The sequence $\{V_{h+1}^\star(s_{h+1}^{(k,m,j)_{l_i}}) - \mathbb{P}_{s,a,h}V_{h+1}^\star\}_{i \in \mathbb{N}^+}$ is a martingale sequence with its absolute values bounded by $H$. Then according to Azuma-Hoeffding inequality, for any $p \in (0,1)$, with probability at least $1 - p$, it holds for given $n_h^k(s,a) = n \in \mathbb{N}_+$ that:

$$\frac{1}{n} \left| \sum_{i=1}^n \left( V_{h+1}^\star(s_{h+1}^{(k,m,j)_{l_i}}) - \mathbb{P}_{s,a,h}V_{h+1}^\star \right) \right| \leq \sqrt{\frac{2H^2\iota}{n}}.$$

For any $k \in [K]$, we have $n_h^k(s,a) \in [\frac{T_1}{H}]$. Considering all the possible combinations $(s,a,h,k) \in \mathcal{S} \times \mathcal{A} \times [H] \times [\frac{T_1}{H}]$ and $n_h^k(s,a) \in [\frac{T_1}{H}]$, with probability at least $1 - SAT_1^2/Hp$, it holds simultaneously for all $(s,a,h,k) \in \mathcal{S} \times \mathcal{A} \times [H] \times [K]$ that:

$$\frac{1}{n_h^k(s,a)} \left| \sum_{i=1}^{n_h^k} \left( V_{h+1}^\star(s_{h+1}^{(k,m,j)_{l_i}}) - \mathbb{P}_{s,a,h}V_{h+1}^\star \right) \right| \leq \sqrt{\frac{2H^2\iota}{n_h^k(s,a)}}.$$

This conclusion also holds for for $\mathcal{E}_6$ and $\mathcal{E}_7$ as $\{(\mathbb{P}_{s,a,h} - \mathbb{1}_{s_{h+1}^{(k,m,j)_{l_i}}})(V_{h+1}^{k_{l_i}} - V_{h+1}^{\text{ref},k_{l_i}})\}_{i \in \mathbb{N}^+}$ is a martingale sequence with its absolute values bounded by $2H$, and $\{(\mathbb{P}_{s,a,h} - \mathbb{1}_{s_{h+1}^{(k,m,j)_{l_i}}})(V_{h+1}^{k_{l_i}} - V_{h+1}^{\text{ref},k_{l_i}})^2\}_{i \in \mathbb{N}^+}$ is a martingale sequence with its absolute values bounded by $2H^2$.

Next, we will prove with probability at least $1 - SAT_1p$, $\mathcal{E}_3$ holds. $\{(\mathbb{P}_{s,a,h} - \mathbb{1}_{s_{h+1}^{(k,m,j)_{L_i}}})V_{h+1}^{\text{ref},k_{L_i}}\}_{i \in \mathbb{N}^+}$ is a martingale sequence bounded by $H$. Then according to Azuma-Hoeffding inequality, for any $p \in (0,1)$, with probability at least $1 - p$, it holds for a given $N_h^k(s,a) = N \in \mathbb{N}_+$ that:

$$\left| \sum_{i=1}^{N} \left( \mathbb{P}_{s,a,h} - \mathbb{1}_{s_{h+1}^{(k,m,j)_{L_i}}} \right) V_{h+1}^{\text{ref},k_{L_i}} \right| \leq H\sqrt{2N\iota}.$$

For any $k \in [K]$, we have $N_h^k(s,a) \in [\frac{T_1}{H}]$. Considering all the possible combinations $(s,a,h,N) \in \mathcal{S} \times \mathcal{A} \times [H] \times [\frac{T_1}{H}]$, with probability at least $1 - SAT_1p$, it holds simultaneously for all $(s,a,h,k) \in \mathcal{S} \times \mathcal{A} \times [H] \times [K]$ that:

$$\left| \sum_{i=1}^{N_h^k} \left( \mathbb{P}_{s,a,h} - \mathbb{1}_{s_{h+1}^{(k,m,j)_{L_i}}} \right) V_{h+1}^{\text{ref},k_{L_i}} \right| \leq H\sqrt{2N_h^k(s,a)\iota}.$$

This conclusion also holds for $\mathcal{E}_4$, $\mathcal{E}_{12}$ and $\mathcal{E}_{15}$ because of the similar martingale structures as follows. For $\mathcal{E}_4$, the sequence $\{(\mathbb{P}_{s,a,h} - \mathbb{1}_{s_{h+1}^{(k,m,j)_{L_i}}})(V_{h+1}^{\text{ref},k_{L_i}})^2\}_{i \in \mathbb{N}^+}$ is a martingale sequence with its absolute values bounded by $H^2$. For $\mathcal{E}_{12}$, the sequence $\{(\mathbb{P}_{s,a,h} - \mathbb{1}_{s_{h+1}^{(k,m,j)_{L_i}}})\lambda_{h+1}^{k_{L_i}}\}_{i \in \mathbb{N}^+}$ is a martingale sequence with its absolute values bounded by $1$. For $\mathcal{E}_{15}$, the sequence $\{(\mathbb{P}_{s,a,h} - \mathbb{1}_{s_{h+1}^{(k,m,j)_{L_i}}})(V_{h+1}^{\text{ref},k_{L_i}} - V_{h+1}^{\star})\}_{i \in \mathbb{N}^+}$ is a martingale sequence with its absolute values bounded by $2H$.

Now, we will prove, with probability at least $1 - p$, $\mathcal{E}_8$ holds. Because of (i) in Lemma D.1, we can append multiple 0s to the summation such that there are $T_1$ terms. Since the sequence $\{(\mathbb{P}_{s_h^{k,m,j},a_h^{k,m,j},h} - \mathbb{1}_{s_{h+1}^{k,m,j}})\lambda_{h+1}^k(s_{h+1}^{k,m,j})\}_{h,k,m,j}$ can be reordered chronologically to a martingale sequence with its absolute values bounded by $1$, it is still a martingale sequence with its absolute values bounded by $1$ after appending some 0 terms. According to Azuma-Hoeffding inequality, for any $p \in (0,1)$, with probability at least $1 - p$, it holds that:

$$\left| \sum_{h=1}^{H} \sum_{k,m,j} \left( \mathbb{P}_{s_h^{k,m,j},a_h^{k,m,j},h} - \mathbb{1}_{s_{h+1}^{k,m,j}} \right) \lambda_{h+1}^k(s_{h+1}^{k,m,j}) \right| \leq \sqrt{2T_1\iota}.$$

Similarly, the conclusion also holds for $\mathcal{E}_9$, $\mathcal{E}_{11}$, $\mathcal{E}_{13}$ and $\mathcal{E}_{14}$ because of their similar martingale structures as follows. For $\mathcal{E}_9$, the sequence $\{(1+2/H)^h(\mathbb{P}_{s_h^{k,m,j},a_h^{k,m,j},h} - \mathbb{1}_{s_{h+1}^{k,m,j}})(V_{h+1}^k - V_{h+1}^{\star})\}_{k,m,j,h}$ can be reordered to a martingale sequence with the absolute values bounded by $18H$. For $\mathcal{E}_{11}$, the sequence $\{(1+2/H)^{h-1}\mathbb{I}[t_h^{k,m,j} > 1](\mathbb{P}_{s_h^{k,m,j},a_h^{k,m,j},h} - \mathbb{1}_{s_{h+1}^{k,m,j}})(V_{h+1}^{\star} - V_{h+1}^{\pi^k})\}_{k,m,j,h}$ can be reordered to a martingale sequence with its absolute values bounded by $9H$. For $\mathcal{E}_{13}$, the sequence $\{\mathbb{P}_{s_h^{k,m,j},a_h^{k,m,j},h}(V_{h+1}^{\star}) - V_{h+1}^{\star}(s_{h+1}^{k,m,j})\}_{k,m,j,h}$ can be reordered to a martingale sequence with the absolute values bounded by $H$. For $\mathcal{E}_{14}$, the sequence $\{\mathbb{P}_{s_h^{k,m,j},a_h^{k,m,j},h}(V_{h+1}^{\star})^2 - (V_{h+1}^{\star}(s_{h+1}^{k,m,j}))^2\}_{k,m,j,h}$ can be reordered to a martingale sequence with its absolute values bounded by $H^2$.

Now, we will prove with probability at least $1 - SAT_1(HT_1^3+1)p$, $\mathcal{E}_2$ holds. According to the Lemma D.4 with $\epsilon = \frac{1}{T_1^2}$, $c = H$ and $p \leftarrow \frac{p}{2}$, we have that with probability at least $1 - (NH^2T_1^2 + 1)p$, it holds for a given $N_h^k(s,a) = N \in \mathbb{N}_+$ that:

$$|\chi_1| \leq \frac{2}{N} \left( \sqrt{\sum_{i=1}^{N} \mathbb{V}_{s,a,h}(V_{h+1}^{\text{ref},k_{L_i}})\iota} + \frac{\sqrt{\iota}}{T_1} + H\iota \right),$$

For any $k \in [K]$, we have $N_h^k(s,a) \in [\frac{T_1}{H}]$. Considering all the possible combination $(s,a,h,N) \in \mathcal{S} \times \mathcal{A} \times [H] \times [\frac{T_1}{H}]$, then with probability at least $1 - SAT_1(HT_1^3 + 1)p$, it holds simultaneously

for all $(s, a, h, k) \in \mathcal{S} \times \mathcal{A} \times [H] \times [K]$ that:

$$|\chi_1| \leq \frac{2}{N_h^k(s,a)} \left( \sqrt{\sum_{i=1}^{N_h^k} \mathbb{V}_{s,a,h}(V_{h+1}^{\text{ref},k_{L_i}})\iota} + \frac{\sqrt{\iota}}{T_1} + H\iota \right).$$

Similarly, with probability at least $1 - SAT_1^2(4HT_1^3 + 1)/Hp$, $\mathcal{E}_5$ holds.

Finally, we will prove, with probability at least $1 - SAT_1^2/Hp$, $\mathcal{E}_{10}$ holds. $V(s, a, h, t)$ is the summation for all the visits to $(s, a, h)$ in stage $t$, which is a martingale sequence with the order assigned chronologically. According to Azuma-Hoeffding Inequality, for any $p \in (0, 1)$, with probability at least $1 - p$, it holds for a given $y_h^t(s, a) = y \in \mathbb{N}_+$ that:

$$\left| \sum_{k,m,j} \left( \mathbb{P}_{s,a,h} - \mathbb{1}_{s_{h+1}^{k,m,j}} \right) \left( V_{h+1}^k - V_{h+1}^\star \right) \mathbb{I}\left[ (s_h^{k,m,j}, a_h^{k,m,j}) = (s, a), t_h^k(s, a) = t \right] \right| \leq 2H\sqrt{2y\iota}.$$

For any $t \in [T_h(s, a)]$, $y_h^t(s, a) \in [\frac{T_1}{H}]$. Considering all combination of $(s, a, h, y) \in \mathcal{S} \times \mathcal{A} \times H \times [\frac{T_1}{H}]$, with probability at least $1 - SAT_1^2/Hp$, it holds simultaneously for any $(s, a, h) \in \mathcal{S} \times \mathcal{A} \times H$ and any $t \in [T_1/H]$ that:

$$|V(s, a, h, t)| \leq 2H\sqrt{2y_h^t(s,a)\iota}.$$

$\square$

# E  PROOF OF THEOREM 4.1

In this section, we provide the proof of Theorem 4.1. Throughout this section, we will discuss under the event $\bigcap_{i=1}^{15} \mathcal{E}_i$ and show

$\text{Regret}(T)$

$$\leq O((1 + \sqrt{\beta} + \beta)\sqrt{MSAH^2T\iota} + H\sqrt{MT\iota}\log(T) + H\sqrt{MT\iota}\log(MSAH^2)$$
$$+ M^{\frac{1}{4}}SAH^{\frac{11}{4}}T^{\frac{1}{4}}\iota^{\frac{3}{4}} + SH^2N_0\log(T) + \sqrt{MSA}H^2\log(T)\sqrt{\iota} + S^{\frac{3}{2}}AH^3\sqrt{N_0}\log(T)\iota$$
$$+ SAH^{\frac{5}{2}}(M\iota)^{\frac{1}{2}} + MSAH^2\sqrt{\iota} + MSAH^2\sqrt{\beta\iota} + MSAH^2\sqrt{\beta^2\iota} + M^{\frac{1}{4}}S^{\frac{5}{4}}A^{\frac{5}{4}}H^{\frac{13}{4}}\iota^{\frac{3}{4}}$$
$$+ S^{\frac{3}{2}}AH^3\sqrt{N_0}\log(MSAH^2)\iota + SH^2N_0\log(MSAH^2) + \sqrt{MSA}H^2\log(MSAH^2)\sqrt{\iota}),$$
$$(26)$$

where $\mathcal{E}_i$s are the events in Lemma D.5 which shows that $\mathbb{P}(\bigcap_{i=1}^{15} \mathcal{E}_i) \geq 1 - (4SAT_1^5 + SAHT_1^4 + 5SAT_1^2/H + 5SAT_1 + 5)p$. Thus, showing Equation (26) will complete the proof. Before we start, we introduce some stage-wise notations. Let $\tilde{\mu}_h^{\text{ref},k}(s, a) = \sum_{k':t_h^{k'} < t_h^k} \sum_m \mu_{h,\text{ref}}^{m,k'}$, $\tilde{\sigma}_h^{\text{ref},k}(s, a) = \sum_{k':t_h^{k'} < t_h^k} \sum_m \sigma_{h,\text{ref}}^{m,k'}$, $\tilde{\mu}_h^{\text{adv},k}(s, a) = \sum_{k':t_h^{k'} = t_h^k - 1} \sum_m \mu_{h,\text{adv}}^{m,k'}$, $\tilde{\sigma}_h^{\text{adv},k}(s, a) = \sum_{k':t_h^{k'} = t_h^k - 1} \sum_m \sigma_{h,\text{adv}}^{m,k'}$, $\tilde{\mu}_h^{\text{val},k}(s, a) = \sum_{k':t_h^{k'} = t_h^k - 1} \sum_m \mu_{h,\text{val}}^{m,k'}$, $\tilde{v}_h^{\text{ref},k} = \frac{\tilde{\sigma}_h^{\text{ref},k}}{N_h^k} - (\frac{\tilde{\mu}_h^{\text{ref},k}}{N_h^k})^2$ and $\tilde{v}_h^{\text{adv},k} = \frac{\tilde{\sigma}_h^{\text{adv},k}}{n_h^k} - (\frac{\tilde{\mu}_h^{\text{adv},k}}{n_h^k})^2$. Here, $\tilde{\mu}_h^{\text{ref},k}$ and $\tilde{\sigma}_h^{\text{ref},k}$ represent the sum of the reference function or squared reference function at step $h + 1$ with regard to all visits of $(s, a, h)$ before stage $t_h^k(s, a)$, and $\tilde{\mu}_h^{\text{adv},k}, \tilde{\sigma}_h^{\text{adv},k}, \tilde{\mu}_h^{\text{val},k}$ are the sum of the advantage function, squared advantage function, and the estimated value function at step $h + 1$ with regard to visits of $(s, a, h)$ during stage $t_h^k(s, a) - 1$. Using the definition of $L_i(s, a, h)$ and $l_i(s, a, h, k)$, we have the following equalities:

$$\tilde{\mu}_h^{\text{ref},k}(s, a) = \sum_{i=1}^{N_h^k} V_{h+1}^{\text{ref},k_{L_i}}(s_{h+1}^{(k,m,j)_{L_i}}), \quad \tilde{\sigma}_h^{\text{ref},k}(s, a) = \sum_{i=1}^{N_h^k} \left( V_{h+1}^{\text{ref},k_{L_i}}(s_{h+1}^{(k,m,j)_{L_i}}) \right)^2,$$

$$\tilde{\mu}_h^{\text{adv},k}(s, a) = \sum_{i=1}^{n_h^k} (V_{h+1}^{k_{l_i}} - V_{h+1}^{\text{ref},k_{l_i}})(s_{h+1}^{(k,m,j)_{l_i}}), \quad \tilde{\sigma}_h^{\text{adv},k}(s, a) = \sum_{i=1}^{n_h^k} (V_{h+1}^{k_{l_i}} - V_{h+1}^{\text{ref},k_{l_i}})^2(s_{h+1}^{(k,m,j)_{l_i}}),$$

$$\tilde{\mu}_h^{\text{val},k}(s,a) = \sum_{i=1}^{n_h^k} V_{h+1}^{k_{l_i}}(s_{h+1}^{(k,m,j)_{l_i}}).$$

We also denote

$$\tilde{b}_h^{k+1,1} = b_h^{k+1,1} = \sqrt{2H^2\iota/n_h^{k+1}},$$

$$\tilde{b}_h^{k+1,2}(s,a) = 2\sqrt{\tilde{v}_h^{\text{ref},k+1}/N_h^{k+1}} + 2\sqrt{\tilde{v}_h^{\text{adv},k+1}/n_h^{k+1}}$$
$$+ 10H\left((\iota/N_h^{k+1})^{3/4} + (\iota/n_h^{k+1})^{3/4} + \iota/N_h^{k+1} + \iota/n_h^{k+1}\right),$$

and

$$\tilde{Q}_h^{k+1,1}(s,a) = r_h(s,a) + \tilde{\mu}_h^{\text{val},k+1}/n_h^{k+1} + \tilde{b}_h^{k+1,1}(s,a),$$

$$\tilde{Q}_h^{k+1,2}(s,a) = r_h(s,a) + \tilde{\mu}_h^{\text{ref},k+1}/N_h^{k+1} + \tilde{\mu}_h^{\text{adv},k+1}/n_h^{k+1} + \tilde{b}_h^{k+1,2}(s,a).$$

For $k \in \mathbb{N}_+$ such that $t_h^{k+1} > t_h^k$, we have the following relationships:

$$\tilde{\mu}_h^{\text{adv},k+1}(s,a) = \mu_h^{\text{adv},k+1}(s,a),$$

$$\tilde{\sigma}_h^{\text{adv},k+1}(s,a) = \sigma_h^{\text{adv},k+1}(s,a),$$

$$\tilde{\mu}_h^{\text{val},k+1}(s,a) = \mu_h^{\text{val},k+1}(s,a).$$

In this case, we have $\tilde{Q}_h^{k+1,1}(s,a) = Q_h^{k+1,1}(s,a)$ and $\tilde{Q}_h^{k+1,2}(s,a) = Q_h^{k+1,2}(s,a)$. Therefore, based on the update rule Equation (7), for $t_h^{k+1}(s,a) > t_h^k(s,a)$, we have $Q_h^{k+1}(s,a) = \min\{\tilde{Q}_h^{k+1,1}(s,a), \tilde{Q}_h^{k+1,2}(s,a), Q_h^k(s,a)\}$. Since these stage-wise notations $\tilde{\mu}_h^{\text{ref},k}$, $\tilde{\mu}_h^{\text{adv},k}$ and $\tilde{\mu}_h^{\text{val},k}$ have the same value for different rounds in the same stage, for $t_h^{k+1}(s,a) = t_h^k(s,a)$, we have $\tilde{Q}_h^{k+1,1}(s,a) = \tilde{Q}_h^{k,1}(s,a)$ and $\tilde{Q}_h^{k+1,2}(s,a) = \tilde{Q}_h^{k,2}(s,a)$. According to the update rule Equation (7), in this case we have $Q_h^{k+1}(s,a) = Q_h^k(s,a)$. In each stage, using mathematical induction, we can find that for any $k \in \mathbb{N}_+$, it holds:

$$Q_h^{k+1}(s,a) = \mathbb{I}\left[t_h^{k+1} = 1\right]H + \mathbb{I}\left[t_h^{k+1} > 1\right]\min\{\tilde{Q}_h^{k+1,1}(s,a), \tilde{Q}_h^{k+1,2}(s,a), Q_h^k(s,a)\}. \quad (27)$$

Here, $t_h^{k+1}$ is the abbreviation of $t_h^{k+1}(s,a)$. Since $Q_h^k(s,a)$ is non-increasing with respect to $k$, in the following Lemma E.1, we will give a lower bound of $Q_h^k(s,a)$.

**Lemma E.1.** *Under the event $\bigcap_{i=1}^7 \mathcal{E}_i$ in Lemma D.5, it holds that for any $(s,a,h,k) \in \mathcal{S} \times \mathcal{A} \times [H] \times [K]$:*

$$Q_h^k(s,a) \geq Q_h^\star(s,a).$$

*Then we have $V_h^k(s) \geq V_h^\star(s)$ for any $(s,a,h,k) \in \mathcal{S} \times \mathcal{A} \times [H] \times [K]$.*

*Proof.* We first claim that based on the event $\mathcal{E}_3 \cap \mathcal{E}_4$ in Lemma D.5, it holds for any $(s,a,h,k) \in \mathcal{S} \times \mathcal{A} \times [H] \times [K]$ that

$$\sum_{i=1}^{N_h^k} \mathbb{V}_{s,a,h}(V_{h+1}^{\text{ref},k_{L_i}}) \leq N_h^k(s,a)\tilde{v}_h^{\text{ref},k}(s,a) + 5H^2\sqrt{N_h^k(s,a)\iota}, \quad (28)$$

and based on the event $\mathcal{E}_6 \cap \mathcal{E}_7$, for any $(s,a,h,k) \in \mathcal{S} \times \mathcal{A} \times [H] \times [K]$, we have:

$$\sum_{i=1}^{n_h^k} \mathbb{V}_{s,a,h}(V_{h+1}^{k_{l_i}} - V_{h+1}^{\text{ref},k_{l_i}}) \leq n_h^k(s,a)\tilde{v}_h^{\text{adv},k}(s,a) + 10H^2\sqrt{n_h^k(s,a)\iota}. \quad (29)$$

We will prove Equation (28) and Equation (29) at the end of the proof for Lemma E.1. Combining Equation (28) with the event $\mathcal{E}_2$, for any $(s, a, h, k) \in \mathcal{S} \times \mathcal{A} \times [H] \times [K]$, we have:

$$
\begin{aligned}
|\chi_1| &\leq \frac{2}{N_h^k(s,a)} \left( \sqrt{\sum_{i=1}^{N_h^k} \mathbb{V}_{s,a,h}(V_{h+1}^{\text{ref},k_{L_i}})\iota} + \frac{\sqrt{\iota}}{T_1} + H\iota \right) \\
&\leq \frac{2}{N_h^k(s,a)} \left( \sqrt{N_h^k(s,a)\tilde{v}_h^{\text{ref},k}(s,a)\iota} + \sqrt{5H^2\iota\sqrt{N_h^k(s,a)\iota}} + \frac{\sqrt{\iota}}{T_1} + H\iota \right) \\
&= 2 \left( \sqrt{\frac{\tilde{v}_h^{\text{ref},k}(s,a)\iota}{N_h^k(s,a)}} + \frac{\sqrt{5}H\iota^{\frac{3}{4}}}{N_h^k(s,a)^{\frac{3}{4}}} + \frac{\sqrt{\iota}}{N_h^k(s,a)T_1} + \frac{H\iota}{N_h^k(s,a)} \right) \\
&\leq 2\sqrt{\frac{\tilde{v}_h^{\text{ref},k}(s,a)\iota}{N_h^k(s,a)}} + 5H \left( \frac{\iota}{N_h^k(s,a)} \right)^{\frac{3}{4}} + 4H\frac{\iota}{N_h^k(s,a)}.
\end{aligned}
$$

Similarly, combining Equation (29) with the event $\mathcal{E}_5$ in Lemma D.5, for any $(s, a, h, k) \in \mathcal{S} \times \mathcal{A} \times [H] \times [K]$, we have:

$$
|\chi_2| \leq 2\sqrt{\frac{\tilde{v}_h^{\text{adv},k}(s,a)\iota}{n_h^k(s,a)}} + 10H \left( \frac{\iota}{n_h^k(s,a)} \right)^{\frac{3}{4}} + 6H\frac{\iota}{n_h^k(s,a)}.
$$

Therefore, according to the definition of $\tilde{b}_h^{k,2}(s,a)$, for any $(s, a, h, k) \in \mathcal{S} \times \mathcal{A} \times [H] \times [K]$, it holds that:

$$
\tilde{b}_h^{k,2}(s,a) \geq |\chi_1| + |\chi_2|. \tag{30}
$$

Now we use mathematical induction on $k$ to prove $Q_h^k(s,a) \geq Q_h^\star(s,a)$ for any $(s, a, h, k) \in \mathcal{S} \times \mathcal{A} \times [H] \times [K]$. For $k = 1$, $Q_h^1(s,a) = H \geq Q_h^\star(s,a)$ for any $(s, a, h) \in \mathcal{S} \times \mathcal{A} \times [H]$. For $k \geq 2$, assume we already have $Q_h^{k'}(s,a) \geq Q_h^\star(s,a)$ for any $(s, a, h, k') \in \mathcal{S} \times \mathcal{A} \times [H] \times [k-1]$, then we will prove for any $(s, a, h) \in \mathcal{S} \times \mathcal{A} \times [H]$, $Q_h^k(s,a) \geq Q_h^\star(s,a)$. According to Equation (27), the following relationship holds:

$$
Q_h^k(s,a) = \mathbb{I}\left[t_h^k = 1\right] H + \mathbb{I}\left[t_h^k > 1\right] \min\{\tilde{Q}_h^{k,1}(s,a), \tilde{Q}_h^{k,2}(s,a), Q_h^{k-1}(s,a)\}.
$$

Then for any given $(s, a, h) \in \mathcal{S} \times \mathcal{A} \times [H]$, we have the following four cases:

(a) If $t_h^k(s,a) = 1$, then $Q_h^k(s,a) = H \geq Q_h^\star(s,a)$.

(b) If $t_h^k(s,a) > 1$ and $Q_h^k(s,a) = Q_h^{k-1}(s,a)$, then the conclusion holds.

(c) If $t_h^k(s,a) > 1$ and $Q_h^k(s,a) = \tilde{Q}_h^{k,1}(s,a) = r_h(s,a) + \tilde{\mu}_h^{\text{val},k}/n_h^k + \tilde{b}_h^{k,1}(s,a)$.

Because of Equation (1), we have the following equality:

$$
Q_h^\star(s,a) = r_h(s,a) + \mathbb{P}_{s,a,h}V_{h+1}^\star.
$$

Then we have:

$$
\begin{aligned}
Q_h^k(s,a) - Q_h^\star(s,a) &= \tilde{\mu}_h^{\text{val},k}/n_h^k + \tilde{b}_h^{k,1}(s,a) - \mathbb{P}_{s,a,h}V_{h+1}^\star \\
&= \frac{1}{n_h^k} \sum_{i=1}^{n_h^k} \left( V_{h+1}^{k_{l_i}}(s_{h+1}^{(k,m,j)_{l_i}}) - \mathbb{P}_{s,a,h}V_{h+1}^\star \right) + \tilde{b}_h^{k,1}(s,a). \tag{31}
\end{aligned}
$$

According to the definition of $l_i(s,a,h,k)$, we know $k_{l_i} < k$ for $i \in [n_h^k(s,a)]$. Then $Q_h^{k_{l_i}}(s,a) \geq Q_h^\star(s,a)$ based on the induction. Therefore, according to the update rule Equation (13) and Equation (1), for any $(s, h) \in \mathcal{S} \times [H]$ and any $i \in [n_h^k(s,a)]$, we have:

$$
V_{h+1}^{k_{l_i}}(s) = \max_{a \in \mathcal{A}} Q_{h+1}^{k_{l_i}}(s,a) \geq \max_{a \in \mathcal{A}} Q_{h+1}^\star(s,a) = V_{h+1}^\star(s), \tag{32}
$$

and for any $i \in [n_h^k(s,a)]$ it holds:

$$
\mathbb{P}_{s,a,h}V_{h+1}^{k_{l_i}} \geq \mathbb{P}_{s,a,h}V_{h+1}^\star. \tag{33}
$$

Combining Equation (31) and Equation (32), we have:

$$Q_h^k(s,a) - Q_h^\star(s,a) \geq \frac{1}{n_h^k(s,a)} \sum_{i=1}^{n_h^k} \left( V_{h+1}^\star(s_{h+1}^{(k,m,j)_{l_i}}) - \mathbb{P}_{s,a,h} V_{h+1}^\star \right) + \tilde{b}_h^{k,1}(s,a) \geq 0.$$

The last inequality is because $\tilde{b}_h^{k,1} = \sqrt{2H^2\iota/n_h^k}$ and the event $\mathcal{E}_1$ in Lemma D.5.

(d) If $t_h^k(s,a) > 1$ and $Q_h^k(s,a) = \tilde{Q}_h^{k,2}(s,a) = r_h(s,a) + \tilde{\mu}_h^{\mathrm{ref},k+1}/N_h^{k+1} + \tilde{\mu}_h^{\mathrm{adv},k+1}/n_h^{k+1} + \tilde{b}_h^{k,2}(s,a)$. We have that

$$
\begin{aligned}
&Q_h^k(s,a) - Q_h^\star(s,a) \\
&= \tilde{\mu}_h^{\mathrm{ref},k}/N_h^k + \tilde{\mu}_h^{\mathrm{adv},k}/n_h^k + \tilde{b}_h^{k,2}(s,a) - \mathbb{P}_{s,a,h} V_{h+1}^\star \\
&= \frac{\sum_{i=1}^{N_h^k} V_{h+1}^{\mathrm{ref},k_{L_i}}(s_{h+1}^{(k,m,j)_{L_i}})}{N_h^k(s,a)} + \frac{\sum_{i=1}^{n_h^k}(V_{h+1}^{k_{l_i}} - V_{h+1}^{\mathrm{ref},k_{l_i}})(s_{h+1}^{(k,m,j)_{l_i}})}{n_h^k(s,a)} + \tilde{b}_h^{k,2}(s,a) - \mathbb{P}_{s,a,h} V_{h+1}^\star \\
&= \frac{\sum_{i=1}^{N_h^k} \mathbb{P}_{s,a,h} V_{h+1}^{\mathrm{ref},k_{L_i}}}{N_h^k(s,a)} + \frac{\sum_{i=1}^{n_h^k} \mathbb{P}_{s,a,h}(V_{h+1}^{k_{l_i}} - V_{h+1}^{\mathrm{ref},k_{l_i}})}{n_h^k(s,a)} - \mathbb{P}_{s,a,h} V_{h+1}^\star + \tilde{b}_h^{k,2}(s,a) - \chi_1 - \chi_2 \\
&= \left( \frac{\sum_{i=1}^{N_h^k} \mathbb{P}_{s,a,h} V_{h+1}^{\mathrm{ref},k_{L_i}}}{N_h^k(s,a)} - \frac{\sum_{i=1}^{n_h^k} \mathbb{P}_{s,a,h} V_{h+1}^{\mathrm{ref},k_{l_i}}}{n_h^k(s,a)} \right) + \frac{\sum_{i=1}^{n_h^k} \mathbb{P}_{s,a,h} V_{h+1}^{k_{l_i}} - \mathbb{P}_{s,a,h} V_{h+1}^\star}{n_h^k(s,a)} \\
&\quad + \left( \tilde{b}_h^{k,2}(s,a) - \chi_1 - \chi_2 \right).
\end{aligned}
\tag{34}
$$

As $V_{h+1}^{\mathrm{ref},k}$ is non-increasing with regard to $k$ based on (j) in Lemma D.1, we have:

$$\frac{1}{N_h^k(s,a)} \sum_{i=1}^{N_h^k} \mathbb{P}_{s,a,h} V_{h+1}^{\mathrm{ref},k_{L_i}} \geq \frac{1}{n_h^k(s,a)} \sum_{i=1}^{n_h^k} \mathbb{P}_{s,a,h} V_{h+1}^{\mathrm{ref},k_{l_i}}. \tag{35}$$

Based on Equation (35), Equation (33), and Equation (30), we know each term in Equation (34) is nonnegative. Therefore, in this case $Q_h^k(s,a) - Q_h^\star(s,a) \geq 0$.

In summary, we prove the conclusion that $Q_h^k(s,a) \geq Q_h^\star(s,a)$ for any $(s,a,h,k) \in \mathcal{S} \times \mathcal{A} \times [H] \times [K]$. The only thing left is to prove Equation (28) and Equation (29).

**Proof of Equation (28) and Equation (29).** For any given $(s,a,h,k) \in \mathcal{S} \times \mathcal{A} \times [H] \times [K]$, let:

$$\chi_3(s,a,h,k) = \sum_{i=1}^{N_h^k} \left( \mathbb{P}_{s,a,h} - \mathbb{1}_{s_{h+1}^{(k,m,j)_{L_i}}} \right) \left( V_{h+1}^{\mathrm{ref},k_{L_i}} \right)^2, \tag{36}$$

$$\chi_4(s,a,h,k) = \frac{\left( \sum_{i=1}^{N_h^k} V_{h+1}^{\mathrm{ref},k_{L_i}}(s_{h+1}^{(k,m,j)_{L_i}}) \right)^2}{N_h^k(s,a)} - \frac{\left( \sum_{i=1}^{N_h^k} \mathbb{P}_{s,a,h} V_{h+1}^{\mathrm{ref},k_{L_i}} \right)^2}{N_h^k(s,a)}, \tag{37}$$

$$\chi_5(s,a,h,k) = \frac{\left( \sum_{i=1}^{N_h^k} \mathbb{P}_{s,a,h} V_{h+1}^{\mathrm{ref},k_{L_i}} \right)^2}{N_h^k(s,a)} - \sum_{i=1}^{N_h^k} \left( \mathbb{P}_{s,a,h} V_{h+1}^{\mathrm{ref},k_{L_i}} \right)^2. \tag{38}$$

Without ambiguity, we will use the abbreviations $\chi_3$, $\chi_4$, and $\chi_5$ in the following proof.

First, we focus on bounding $|\chi_3|$. Using the definition of $\tilde{v}_h^{\mathrm{ref},k}(s,a)$, we have:

$$N_h^k(s,a) \tilde{v}_h^{\mathrm{ref},k}(s,a) = \sum_{i=1}^{N_h^k} \left( V_{h+1}^{\mathrm{ref},k_{L_i}}(s_{h+1}^{(k,m,j)_{L_i}}) \right)^2 - \frac{\left( \sum_{i=1}^{N_h^k} V_{h+1}^{\mathrm{ref},k_{L_i}}(s_{h+1}^{(k,m,j)_{L_i}}) \right)^2}{N_h^k(s,a)}. \tag{39}$$

Summing Equation (36), Equation (37), Equation (38) and Equation (39), we can find that:

$$\sum_{i=1}^{N_h^k} \mathbb{V}_{s,a,h}(V_{h+1}^{\mathrm{ref},k_{L_i}}) = \sum_{i=1}^{N_h^k} \mathbb{P}_{s,a,h}(V_{h+1}^{\mathrm{ref},k_{L_i}})^2 - \sum_{i=1}^{N_h^k} \left(\mathbb{P}_{s,a,h}V_{h+1}^{\mathrm{ref},k_{L_i}}\right)^2$$

$$= N_h^k(s,a)\tilde{v}_h^{\mathrm{ref},k} + \chi_3 + \chi_4 + \chi_5. \tag{40}$$

Because of the event $\mathcal{E}_4$ in Lemma D.5, we know:

$$|\chi_3| \leq H^2\sqrt{2N_h^k(s,a)\iota}. \tag{41}$$

Next, we focus on bounding $|\chi_4|$. Using the absolute value inequality, it holds that:

$$\left|\sum_{i=1}^{N_h^k}\left(V_{h+1}^{\mathrm{ref},k_{L_i}}(s_{h+1}^{(k,m,j)_{L_i}}) + \mathbb{P}_{s,a,h}V_{h+1}^{\mathrm{ref},k_{L_i}}\right)\right|$$

$$\leq \sum_{i=1}^{N_h^k}\left(\left|V_{h+1}^{\mathrm{ref},k_{L_i}}(s_{h+1}^{(k,m,j)_{L_i}})\right| + \left|\mathbb{P}_{s,a,h}V_{h+1}^{\mathrm{ref},k_{L_i}}\right|\right) \leq 2H.$$

Then we have: :

$$|\chi_4| = \frac{1}{N_h^k(s,a)}\left|\sum_{i=1}^{N_h^k}\left(V_{h+1}^{\mathrm{ref},k_{L_i}}(s_{h+1}^{(k,m,j)_{L_i}}) + \mathbb{P}_{s,a,h}V_{h+1}^{\mathrm{ref},k_{L_i}}\right)\right| \times$$

$$\left|\sum_{i=1}^{N_h^k}\left(V_{h+1}^{\mathrm{ref},k_{L_i}}(s_{h+1}^{(k,m,j)_{L_i}}) - \mathbb{P}_{s,a,h}V_{h+1}^{\mathrm{ref},k_{L_i}}\right)\right|$$

$$\leq 2H\left|\sum_{i=1}^{N_h^k}\left(\mathbb{P}_{s,a,h} - \mathbb{1}_{s_{h+1}^{(k,m,j)_{L_i}}}\right)V_{h+1}^{\mathrm{ref},k_{L_i}}\right| \leq 2H^2\sqrt{2N_h^k(s,a)\iota}. \tag{42}$$

The last inequality is because of the event $\mathcal{E}_3$ in Lemma D.5.

For $\chi_5$, according to the Cauchy-Schwarz Inequality, we have $\chi_5 \leq 0$.

Applying the upper bound of $\chi_3$ Equation (41), $\chi_4$ Equation (42) and $\chi_5$ to Equation (40), we have:

$$\sum_{i=1}^{N_h^k} \mathbb{V}_{s,a,h}(V_{h+1}^{\mathrm{ref},k_{L_i}}) \leq N_h^k(s,a)\tilde{v}_h^{\mathrm{ref},k} + 5H^2\sqrt{N_h^k(s,a)\iota},$$

Then we finish the proof of Equation (28). The proof for Equation (29) is similar, in which we just need to substitute $N_h^k(s,a)$ with $n_h^k(s,a)$ and $H^2$ with $2H^2$. $\qquad\square$

$\qquad\square$

With Lemma E.1, Equation (13) and Equation (1), for any $(s,h,k) \in \mathcal{S} \times [H] \times [K]$, we have:

$$V_h^k(s) = \max_{a'\in\mathcal{A}} Q_h^k(s,a') \geq \max_{a'\in\mathcal{A}} Q_h^\star(s,a') = V_h^\star(s). \tag{43}$$

The following lemma gives a viable value of $N_0$ to learn the reference function $V_h^{\mathrm{ref},k}(s)$. Denote $V_h^{\mathrm{REF}}(s) = V_h^{\mathrm{ref},K+1}(s)$ as the final value of the reference function $V_h^{\mathrm{ref},k}(s)$.

**Lemma E.2.** *Under the event $\bigcap_{i=1}^7 \mathcal{E}_i$ in Lemma D.5, it holds for any $h \in [H]$ and $\beta \in (0,H]$ that:*

$$\sum_{k,m,j} \mathbb{I}\left[V_h^k(s_h^{k,m,j}) - V_h^\star(s_h^{k,m,j}) \geq \beta\right] < 5184\frac{SAH^5\iota}{\beta^2} + 16\frac{MSAH^3}{\beta}.$$

*In addition, letting*

$$N_0 = 5184 \frac{SAH^5\iota}{\beta^2} + 16 \frac{MSAH^3}{\beta}, \beta \in (0, H],$$

*we have that for any $(s, h) \in \mathcal{S} \times [H]$,*

$$V_h^{\text{REF}}(s) = H \text{ or } V_h^\star(s) \le V_h^{\text{REF}}(s) \le V_h^\star(s) + \beta.$$

*Proof.* We claim that for any non-negative weight sequence $\{\omega_{k,m,j}\}_{k,m,j}$ and any $h \in [H]$,

$$\sum_{k,m,j} \omega_{k,m,j} \left(V_h^k - V_h^\star\right)(s_h^{k,m,j}) \le 8H^2 \left(MSAH||\omega||_\infty + 9\sqrt{SAH\iota||\omega||_\infty||\omega||_1}\right). \tag{44}$$

Here, $||\omega||_\infty = \max_{k,m,j} \omega_{k,m,j}$ and $||\omega||_1 = \sum_{k,m,j} \omega_{k,m,j}$. If we have proved Equation (44), then letting $\omega_{k,m,j} = \mathbb{I}[V_h^k(s_h^{k,m,j}) - V_h^\star(s_h^{k,m,j}) \ge \beta]$, according to Equation (44) and Equation (43), we have:

$$
\begin{aligned}
||\omega||_1 &= \sum_{k,m,j} \mathbb{I}\left[V_h^k(s_h^{k,m,j}) - V_h^\star(s_h^{k,m,j}) \ge \beta\right] \\
&\le \frac{1}{\beta} \sum_{k,m,j} \mathbb{I}\left[V_h^k(s_h^{k,m,j}) - V_h^\star(s_h^{k,m,j}) \ge \beta\right] \left(V_h^k(s_h^{k,m,j}) - V_h^\star(s_h^{k,m,j})\right) \\
&\le \frac{1}{\beta} \cdot 8H^2(MSAH + 9\sqrt{SAH\iota||\omega||_1}).
\end{aligned}
$$

Letting $b = 72H^2\sqrt{SAH\iota}$ and $c = 8MSAH^3$, we have:

$$\beta||\omega||_1 - b\sqrt{||\omega||_1} - c \le 0.$$

Solving the inequality, we have:

$$0 \le \sqrt{||\omega||_1} \le \frac{b + \sqrt{b^2 + 4\beta c}}{2\beta}.$$

Then:

$$||\omega||_1 \le \left(\frac{b + \sqrt{b^2 + 4\beta c}}{2\beta}\right)^2 < \frac{b^2 + b^2 + 4\beta c}{2\beta^2} = 5184\frac{SAH^5\iota}{\beta^2} + 16\frac{MSAH^3}{\beta}.$$

Therefore, for any $h \in [H]$, it holds that:

$$\sum_{k,m,j} \mathbb{I}\left[V_h^k(s_h^{k,m,j}) - V_h^\star(s_h^{k,m,j}) \ge \beta\right] = ||\omega||_1 < 5184\frac{SAH^5\iota}{\beta^2} + 16\frac{MSAH^3}{\beta}.$$

Especially, for any $(s, h)$ we have:

$$\sum_{k,m,j} \mathbb{I}\left[V_h^k(s) - V_h^\star(s) \ge \beta, s_h^{k,m,j} = s\right] < N_0.$$

Since $V_h^k(s)$ is non-increasing with regard to $k$ under the event $\bigcap_{i=1}^7 \mathcal{E}_i$ according to Lemma E.1 and (j) in Lemma D.1, $V_h^k(s) - V_h^\star(s)$ is also non-increasing. Before we update the reference function at $(s, h)$, $V_h^{\text{ref},k}(s) = V_h^1(s) = H$. Therefore, if the reference function at $(s, h)$ is not updated in the algorithm, $V_h^{\text{REF}}(s) = H$. Next, we discuss the situation in which the reference function at $(s, h)$ is updated in FedQ-Advantage. When we update the reference function at the end of round $k$, we have $\sum_{k',m,j:k' \le k} \mathbb{I}\left[s_h^{k',m,j} = s\right] \ge N_0$ and thus $0 \le V_h^k(s) - V_h^\star(s) < \beta$. Therefore, for the final value $V_h^{\text{REF}}(s) = V_h^{\text{ref},k+1}(s) = V_h^{k+1}(s)$, it holds that $0 \le V_h^{\text{REF}}(s) - V_h^\star(s) < \beta$. Then we have $V_h^{\text{REF}}(s) = H$ or $V_h^\star(s) \le V_h^{\text{REF}}(s) \le V_h^\star(s) + \beta$ under the event $\bigcap_{i=1}^7 \mathcal{E}_i$. Now, we only need to prove Equation (44).

**Proof of Equation (44).** According to the update rule Equation (13) and Equation (27), for $h \in [H]$, we have:

$$
V_h^k(s_h^{m,k,j}) = \max_{a \in \mathcal{A}} Q_h^k(s_h^{k,m,j}, a) = Q_h^k(s_h^{k,m,j}, a_h^{k,m,j})
$$
$$
\leq \mathbb{I}\left[t_h^{k,m,j} = 1\right] H + \mathbb{I}\left[t_h^{k,m,j} > 1\right] \tilde{Q}_h^{k,1}(s_h^{k,m,j}, a_h^{k,m,j})
$$
$$
= \mathbb{I}\left[t_h^{k,m,j} = 1\right] H + \mathbb{I}\left[t_h^{k,m,j} > 1\right] \times
$$
$$
\left( r_h(s_h^{k,m,j}, a_h^{k,m,j}) + \frac{\sum_{i=1}^{n_h^k} V_{h+1}^{k_{l_i}}(s_{h+1}^{(k,m,j)_{l_i}})}{n_h^k(s_h^{k,m,j}, a_h^{k,m,j})} + \tilde{b}_h^{k,1}(s_h^{k,m,j}, a_h^{k,m,j}) \right),
$$

and according to the Bellman equality Equation (1), we have:

$$
V_h^\star(s_h^{m,k,j}) = \max_{a \in \mathcal{A}} Q_h^\star(s_h^{k,m,j}, a) \geq Q_h^\star(s_h^{k,m,j}, a_h^{k,m,j})
$$
$$
\geq \mathbb{I}\left[t_h^{k,m,j} > 1\right] \left( r_h(s_h^{k,m,j}, a_h^{k,m,j}) + \mathbb{P}_{s_h^{k,m,j}, a_h^{k,m,j}, h} V_{h+1}^\star \right).
$$

Combined these two inequalities, it holds for any $h \in [H]$ that:

$$
\left(V_h^k - V_h^\star\right)(s_h^{k,m,j})
$$
$$
\leq \mathbb{I}\left[t_h^{k,m,j} = 1\right] H + \mathbb{I}\left[t_h^{k,m,j} > 1\right] \times \tag{45}
$$
$$
\left( \frac{\sum_{i=1}^{n_h^k} V_{h+1}^{k_{l_i}}(s_{h+1}^{(k,m,j)_{l_i}})}{n_h^k(s_h^{k,m,j}, a_h^{k,m,j})} + \tilde{b}_h^{k,1}(s_h^{k,m,j}, a_h^{k,m,j}) - \mathbb{P}_{s_h^{k,m,j}, a_h^{k,m,j}, h} V_{h+1}^\star \right)
$$
$$
\leq \mathbb{I}\left[t_h^{k,m,j} = 1\right] H + \mathbb{I}\left[t_h^{k,m,j} > 1\right] \left( \frac{\sum_{i=1}^{n_h^k}(V_{h+1}^{k_{l_i}} - V_{h+1}^\star)(s_{h+1}^{(k,m,j)_{l_i}})}{n_h^k(s_h^{k,m,j}, a_h^{k,m,j})} + 2\tilde{b}_h^{k,1} \right).
$$

The last inequality is because of the event $\mathcal{E}_1$ in Lemma D.5. Then for any $h \in [H]$:

$$
\sum_{k,m,j} \omega_{k,m,j}(V_h^k - V_h^\star)(s_h^{k,m,j}) \leq \sum_{k,m,j} \omega_{k,m,j} \mathbb{I}\left[t_h^{k,m,j} = 1\right] H + \mathbb{I}\left[t_h^{k,m,j} > 1\right] \times
$$
$$
\left( \sum_{k,m,j} \omega_{k,m,j} \frac{\sum_{i=1}^{n_h^k}(V_{h+1}^{k_{l_i}} - V_{h+1}^\star)(s_{h+1}^{(k,m,j)_{l_i}})}{n_h^k(s_h^{k,m,j}, a_h^{k,m,j})} + 2\sum_{k,m,j} \omega_{k,m,j} \tilde{b}_h^{k,1}(s_h^{k,m,j}, a_h^{k,m,j}) \right). \tag{46}
$$

For the first term in Equation (46), we have:

$$
\sum_{k,m,j} \mathbb{I}\left[t_h^{k,m,j} = 1\right] H = H \sum_{k,m,j} \mathbb{I}\left[t_h^k(s_h^{k,m,j}, a_h^{k,m,j}) = 1\right] \left( \sum_{(s,a)} \mathbb{I}\left[(s_h^{k,m,j}, a_h^{k,m,j}) = (s,a)\right] \right)
$$
$$
= H \sum_{s,a} \sum_{k,m,j} \mathbb{I}\left[(s_h^{k,m,j}, a_h^{k,m,j}) = (s,a), t_h^k(s,a) = 1\right].
$$

For any $(s,a,h) \in \mathcal{S} \times \mathcal{A} \times [H]$, $\mathbb{I}[(s_h^{k,m,j}, a_h^{k,m,j}) = (s,a), t_h^k(s,a) = 1] = 1$ if and only if $(s_h^{k,m,j}, a_h^{k,m,j}) = (s,a)$ and $t_h^k(s,a) = 1$. Therefore, $\sum_{k,m,j} \mathbb{I}[(s_h^{k,m,j}, a_h^{k,m,j}) = (s,a), t_h^k(s,a) = 0] = Y_h^1(s,a)$. Because of (c) in Lemma D.1, we have:

$$
\sum_{k,m,j} \mathbb{I}\left[t_h^{k,m,j} = 1\right] H = H \sum_{s,a} Y_h^1(s,a) \leq H \cdot SAM(H+1) \leq 2MSAH^2. \tag{47}
$$

Then it holds for any $h \in [H]$ that:

$$
\sum_{k,m,j} \omega_{k,m,j} \mathbb{I}[t_h^{k,m,j} = 1] H \leq ||\omega||_\infty \sum_{k,m,j} \mathbb{I}[t_h^{k,m,j} = 1] H \leq 2MSAH^2 ||\omega||_\infty. \tag{48}
$$

For the second term in Equation (46), we have:

$$\sum_{k,m,j} \omega_{k,m,j} \frac{\sum_{i=1}^{n_h^k} \left(V_{h+1}^{k_{l_i}} - V_{h+1}^{\star}\right)(s_{h+1}^{(k,m,j)_{l_i}})}{n_h^k(s_h^{k,m,j}, a_h^{k,m,j})} \mathbb{I}[t_h^{k,m,j} > 1]$$

$$= \sum_{k,m,j} \sum_{i=1}^{n_h^k} \omega_{k,m,j} \frac{\left(V_{h+1}^{k_{l_i}} - V_{h+1}^{\star}\right)(s_{h+1}^{(k,m,j)_{l_i}})}{n_h^k(s_h^{k,m,j}, a_h^{k,m,j})} \mathbb{I}[t_h^{k,m,j} > 1] \cdot \sum_{k',m',j'} \mathbb{I}[(k,m,j)_{l_i} = (k',m',j')]$$

$$= \sum_{k,m,j} \sum_{i=1}^{n_h^k} \sum_{k',m',j'} \omega_{k,m,j} \frac{\left(V_{h+1}^{k'} - V_{h+1}^{\star}\right)(s_{h+1}^{k',m',j'})}{n_h^k(s_h^{k,m,j}, a_h^{k,m,j})} \mathbb{I}[t_h^{k,m,j} > 1] \cdot \mathbb{I}[(k,m,j)_{l_i} = (k',m',j')]$$

$$= \sum_{k',m',j'} \left( \sum_{k,m,j} \omega_{k,m,j} \frac{\sum_{i=1}^{n_h^k} \mathbb{I}[(k,m,j)_{l_i} = (k',m',j'), t_h^{k,m,j} > 1]}{n_h^k(s_h^{k,m,j}, a_h^{k,m,j})} \right) (V_{h+1}^{k'} - V_{h+1}^{\star})(s_{h+1}^{k',m',j'}).$$

Let:

$$\tilde{\omega}_{k',m',j'} = \sum_{k,m,j} \omega_{k,m,j} \frac{\sum_{i=1}^{n_h^k} \mathbb{I}[(k,m,j)_{l_i} = (k',m',j'), t_h^{k,m,j} > 1]}{n_h^k(s_h^{k,m,j}, a_h^{k,m,j})} \geq 0,$$

and

$$||\tilde{\omega}||_\infty = \max_{k',m',j'} \tilde{\omega}_{k',m',j'}, \; ||\tilde{\omega}||_1 = \sum_{k',m',j'} \tilde{\omega}_{k',m',j'}.$$

We have:

$$\sum_{k,m,j} \omega_{k,m,j} \frac{\sum_{i=1}^{n_h^k} \left(V_{h+1}^{k_{l_i}} - V_{h+1}^{\star}\right)(s_{h+1}^{(k,m,j)_{l_i}})}{n_h^k(s_h^{k,m,j}, a_h^{k,m,j})} \mathbb{I}[t_h^{k,m,j} > 1]$$

$$= \sum_{k',m',j'} \tilde{\omega}_{k',m',j'}(V_{h+1}^{k'} - V_{h+1}^{\star})(s_{h+1}^{k',m',j'}). \tag{49}$$

Next, we will explore the relationship between the norm of $\{\omega_{k,m,j}\}_{k,m,j}$ and $\{\tilde{\omega}_{k,m,j}\}_{k,m,j}$. For a given triple $(k',m',j')$, according to the definition of $l_i(s_h^{k,m,j}, a_h^{k,m,j}, h, k)$, $\sum_{i=1}^{n_h^k} \mathbb{I}[(k,m,j)_{l_i} = (k',m',j'), t_h^{k,m,j} > 1]) = 1$ if and only if $(s_h^{k,m,j}, a_h^{k,m,j}) = (s_h^{k',m',j'}, a_h^{k',m',j'})$ and $1 < t_h^k(s_h^{k,m,j}, a_h^{k,m,j}) = t_h^{k'}(s_h^{k,m,j}, a_h^{k,m,j}) + 1$. In this case, we have $t_h^{k'}(s_h^{k,m,j}, a_h^{k,m,j}) > 0$ and $n_h^k(s_h^{k,m,j}, a_h^{k,m,j}) = y_h^{t_h^{k'}}(s_h^{k',m',j'}, a_h^{k',m',j'})$. Then for a given triple $(k',m',j')$, it holds that:

$$\sum_{k,m,j} \sum_{i=1}^{n_h^k} \mathbb{I}[(k,m,j)_{l_i} = (k',m',j'), t_h^{k,m,j} > 1]$$

$$= \sum_{k,m,j} \mathbb{I}\left[(s_h^{k,m,j}, a_h^{k,m,j}) = (s_h^{k',m',j'}, a_h^{k',m',j'}), t_h^k(s_h^{k,m,j}, a_h^{k,m,j}) = t_h^{k'}(s_h^{k,m,j}, a_h^{k,m,j}) + 1\right]$$

$$= y_h^{t_h^{k'}+1}(s_h^{k',m',j'}, a_h^{k',m',j'}).$$

Then according to (e) in Lemma D.1, it holds that:

$$\sum_{k,m,j} \frac{\sum_{i=1}^{n_h^k} \mathbb{I}[(k,m,j)_{l_i} = (k',m',j'), t_h^{k,m,j} > 1]}{n_h^k(s_h^{k,m,j}, a_h^{k,m,j})} = \frac{y_h^{t_h^{k'}+1}(s_h^{k',m',j'}, a_h^{k',m',j'})}{y_h^{t_h^{k'}}(s_h^{k',m',j'}, a_h^{k',m',j'})} \leq 1 + \frac{2}{H}.$$

Then we have:

$$||\tilde{\omega}||_\infty \leq ||\omega||_\infty \sum_{k,m,j} \frac{\sum_{i=1}^{n_h^k} \mathbb{I}[(k,m,j)_{l_i} = (k',m',j'), t_h^{k,m,j} > 1]}{n_h^k(s_h^{k,m,j}, a_h^{k,m,j})} \leq (1 + \frac{2}{H})||\omega||_\infty.$$

We also have:

$$||\tilde{\omega}||_1 = \sum_{k',m',j'} \sum_{k,m,j} \omega_{k,m,j} \sum_{i=1}^{n_h^k} \frac{\mathbb{I}[(k,m,j)_{l_i} = (k',m',j'), t_h^{k,m,j} > 1]}{n_h^k(s_h^{k,m,j}, a_h^{k,m,j})}$$

$$= \sum_{k,m,j} \omega_{k,m,j} \sum_{i=1}^{n_h^k} \sum_{k',m',j'} \frac{\mathbb{I}[(k,m,j)_{l_i} = (k',m',j'), t_h^{k,m,j} > 1]}{n_h^k(s_h^{k',m',j'}, a_h^{k',m',j'})}$$

$$\leq \sum_{k,m,j} \omega_{k,m,j} = ||\omega||_1.$$

For the third term in Equation (46), we have:

$$\sum_{k,m,j} \omega_{k,m,j} \tilde{b}_h^{k,1}(s_h^{k,m,j}, a_h^{k,m,j}) \mathbb{I}[t_h^{k,m,j} > 1]$$

$$= \sum_{k,m,j} \omega_{k,m,j} \sqrt{\frac{2H^2\iota}{n_h^k(s_h^{k,m,j}, a_h^{k,m,j})}} \mathbb{I}\left[t_h^{k,m,j} > 1\right] \left( \sum_{s,a} \mathbb{I}\left[(s_h^{k,m,j}, a_h^{k,m,j}) = (s,a)\right] \right)$$

$$= \sum_{s,a} \sum_{k,m,j} \omega_{k,m,j} \sqrt{\frac{2H^2\iota}{n_h^k(s,a)}} \mathbb{I}\left[(s_h^{k,m,j}, a_h^{k,m,j}) = (s,a)\right] \left( \sum_{t=2}^{T_h(s,a)} \mathbb{I}\left[t_h^k(s,a) = t\right] \right)$$

$$= \sum_{s,a} \sum_{k,m,j} \sum_{t=2}^{T_h(s,a)} \omega_{k,m,j} \sqrt{\frac{2H^2\iota}{y_h^{t-1}(s,a)}} \mathbb{I}\left[(s_h^{k,m,j}, a_h^{k,m,j}) = (s,a), t_h^k(s,a) = t\right]$$

$$= \sum_{s,a} \sum_{t=1}^{T_h(s,a)-1} \left( \sum_{k,m,j} \omega_{k,m,j} \mathbb{I}\left[(s_h^{k,m,j}, a_h^{k,m,j}) = (s,a), t_h^k(s,a) = t+1\right] \right) \sqrt{\frac{2H^2\iota}{y_h^t(s,a)}}.$$

Denote

$$q(s,a,t) = \sum_{k,m,j} \omega_{k,m,j} \mathbb{I}\left[(s_h^{k,m,j}, a_h^{k,m,j}) = (s,a), t_h^k(s,a) = t+1\right],$$

and

$$q(s,a) = \sum_{t=1}^{T_h(s,a)-1} q(s,a,t) \text{ for } T_h(s,a) \geq 2. \tag{50}$$

Then we have:

$$\sum_{k,m,j} \omega_{k,m,j} \tilde{b}_h^{k,1}(s_h^{k,m,j}, a_h^{k,m,j}) \mathbb{I}[t_h^{k,m,j} > 1] = \sum_{s,a} \sum_{t=1}^{T_h(s,a)-1} q(s,a,t) \sqrt{\frac{2H^2\iota}{y_h^t(s,a)}}. \tag{51}$$

For the coefficient $q(s,a,t)$, we have the following properties:

$$q(s,a,t) \leq ||\omega||_\infty \sum_{k,m,j} \mathbb{I}\left[(s_h^{k,m,j}, a_h^{k,m,j}) = (s,a), t_h^k(s,a) = t+1\right] = ||\omega||_\infty y_h^{t+1}(s,a), \tag{52}$$

and

$$\sum_{s,a} q(s,a) = \sum_{k,m,j} \omega_{k,m,j} \sum_{s,a} \sum_{t=1}^{T_h(s,a)-1} \mathbb{I}\left[(s_h^{k,m,j}, a_h^{k,m,j}) = (s,a), t_h^k(s,a) = t+1\right]$$

$$\leq \sum_{k,m,j} \omega_{k,m,j} \sum_{s,a} \mathbb{I}\left[(s_h^{k,m,j}, a_h^{k,m,j}) = (s,a), t_h^k(s,a) > 1\right]$$

$$\leq \sum_{k,m,j} \omega_{k,m,j} = ||w||_1. \tag{53}$$

Because $y_h^t(s, a)$ is increasing for $1 \leq t \leq T_h(s, a) - 1$, given the equation Equation (50), when the weights $q(s, a, t)$ concentrates on former terms, we can obtain the larger value of the right term in Equation (51). There exists some positive integer $t_0 \leq T_h(s, a) - 1$ satisfying:

$$||\omega||_\infty \sum_{t=1}^{t_0-1} y_h^{t+1}(s, a) < q(s, a) \leq ||\omega||_\infty \sum_{t=1}^{T_h-1} y_h^{t+1}(s, a). \tag{54}$$

and

$$||\omega||_\infty \sum_{t=1}^{t_0} y_h^{t+1}(s, a) \geq q(s, a).$$

Then according to Equation (52), we have

$$\sum_{t=1}^{T_h(s,a)-1} q(s, a, t) \sqrt{\frac{1}{y_h^t(s, a)}} \leq \sum_{t=1}^{t_0} ||\omega||_\infty y_h^{t+1}(s, a) \sqrt{\frac{1}{y_h^t(s, a)}}. \tag{55}$$

Since $t_0 \leq T_h(s, a) - 1$, according to (e) and Equation (16) in Lemma D.1, we have:

$$y_h^{t+1}(s, a) \sqrt{\frac{1}{y_h^t(s, a)}} \leq (1 + \frac{2}{H}) \sqrt{y_h^t(s, a)} \leq 3(1 + \frac{2}{H}) \sqrt{H} \left( \sqrt{Y_h^t(s, a)} - \sqrt{Y_h^{t-1}(s, a)} \right). \tag{56}$$

If $t_0 \geq 2$, we also have:

$$||\omega||_\infty \sum_{t=1}^{t_0-1} y_h^{t+1}(s, a) \geq (1 + \frac{1}{H})||\omega||_\infty \sum_{t=1}^{t_0-1} y_h^t(s, a) = (1 + \frac{1}{H})||\omega||_\infty Y_h^{t_0-1}(s, a)$$

$$\geq \frac{||\omega||_\infty}{2} Y_h^{t_0}(s, a). \tag{57}$$

The last inequality is because of (f) in Lemma D.1. Then according to Equation (54), it holds that:

$$||\omega||_\infty Y_h^{t_0}(s, a) \leq 2||\omega||_\infty \sum_{t=1}^{t_0-1} y_h^{t+1}(s, a) \leq 2q(s, a).$$

Applying inequalities Equation (56) and Equation (57) to Equation (55), we have:

$$\sum_{t=1}^{T_h(s,a)-1} q(s, a, t) \sqrt{\frac{1}{y_h^t(s, a)}}$$

$$\leq 3(1 + \frac{2}{H}) \sqrt{H} ||\omega||_\infty \cdot \sum_{t=1}^{t_0} \left( \sqrt{Y_h^t(s, a)} - \sqrt{Y_h^{t-1}(s, a)} \right)$$

$$= 3(1 + \frac{2}{H}) \sqrt{H ||\omega||_\infty} \cdot \sqrt{||\omega||_\infty Y_h^{t_0}(s, a)}$$

$$\leq 9 \sqrt{2H ||\omega||_\infty q(s, a)}.$$

Here, the first inequality is because of Equation (16). The last inequality uses Equation (57) and $1 + \frac{2}{H} \leq 3$.

If $t_0 = 1$, then $q(s, a) \leq ||\omega||_\infty y_h^2(s, a)$. Therefore, according to Equation (55), we have:

$$\sum_{t=1}^{T_h(s,a)-1} q(s, a, t) \sqrt{\frac{1}{y_h^t(s, a)}} \leq q(s, a) \sqrt{\frac{1}{y_h^1(s, a)}} \leq \sqrt{\frac{||\omega||_\infty y_h^2(s, a) q(s, a)}{y_h^1(s, a)}}.$$

Based on (e) in Lemma D.1, it holds:

$$\sum_{t=1}^{T_h(s,a)-1} q(s, a, t) \sqrt{\frac{1}{y_h^t(s, a)}} \leq \sqrt{(1 + \frac{2}{H})||\omega||_\infty q(s, a)} \leq 9 \sqrt{2H ||\omega||_\infty q(s, a)}.$$

Therefore, for any $t_0 \le T_h(s,a) - 1$ defined in Equation (54), we have:

$$\sum_{t=1}^{T_h(s,a)-1} q(s,a,t) \sqrt{\frac{1}{y_h^t(s,a)}} \le 9\sqrt{2H||\omega||_\infty q(s,a)}.$$

Combined with Equation (51), we have

$$\sum_{k,m,j} \omega_{k,m,j} \tilde{b}_h^{k,1}(s_h^{k,m,j}, a_h^{k,m,j}) \mathbb{I}[t_h^{k,m,j} > 1] \le 18\sqrt{H^3||\omega||_\infty \iota} \sum_{s,a} \sqrt{q(s,a)}$$

$$\le 18\sqrt{SAH^3||\omega||_\infty ||\omega||_1 \iota}. \tag{58}$$

The last inequality uses the Cauchy-Schwarz inequality.

Based on Equation (45), by applying Equation (48), Equation (49) and Equation (58), for any $h \in [H]$, we have:

$$\sum_{k,m,j} \omega_{k,m,j} (V_h^k - V_h^\star)(s_h^{k,m,j}) \le 2MSAH^2||\omega||_\infty + 18\sqrt{SAH^3 \iota ||\omega||_\infty ||\omega||_1}$$

$$+ \sum_{k,m,j} \tilde{\omega}_{k,m,j} (V_{h+1}^k - V_{h+1}^\star)(s_{h+1}^{k,m,j}), \tag{59}$$

where $||\tilde{\omega}||_\infty \le (1 + \frac{2}{H})||\omega||_\infty$, and $||\tilde{\omega}||_1 = ||\omega||_1$.

Using Equation (59), with induction on $h = H, H-1, ..., 1$, we can prove that for any $h \in [H]$:

$$\sum_{k,m,j} \omega_{k,m,j} (V_h^k - V_h^\star)(s_h^{k,m,j}) \le C_h \left( 2MSAH^2||\omega||_\infty + 18\sqrt{SAH^3 \iota ||\omega||_\infty ||\omega||_1} \right), \tag{60}$$

where $C_h = (1 + \frac{H}{2})(1 + \frac{2}{H})^{H-h} - \frac{H}{2}$. Note that $C_h \le 4H$, based on Equation (60), it holds for any $h \in [H]$ that:

$$\sum_{k,m,j} \omega_{k,m,j} (V_h^k - V_h^\star)(s_h^{k,m,j}) \le 8H^2(MSAH||\omega||_\infty + 9\sqrt{SAH\iota||\omega||_\infty ||\omega||_1}).$$

Therefore, we finish the proof of Equation (44). $\qquad\qquad\square$

$\square$

Next, we go back to the proof of Equation (26). In the following content, $\sum_{k,m,j}$ is the simplified notation of $\sum_{k=1}^K \sum_{m=1}^M \sum_{j=1}^{n^{m,k}}$. $N_h^k, n_h^k, L_i, l_i$ represent simplified notations for $N_h^k(s_h^{k,m,j}, a_h^{k,m,j})$, $n_h^k(s_h^{k,m,j}, a_h^{k,m,j})$, $L_i(s_h^{k,m,j}, a_h^{k,m,j}, h)$ and $l_i(s_h^{k,m,j}, a_h^{k,m,j}, h, h, k)$ respectively.

For $h \in [H+1]$, denote:

$$\delta_h^k = \sum_{m=1}^M \sum_{j=1}^{n^{m,k}} \left( V_h^k - V_h^\star \right)(s_h^{k,m,j}),$$

$$\zeta_h^k = \sum_{m=1}^M \sum_{j=1}^{n^{m,k}} \left( V_h^k - V_h^{\pi^k} \right)(s_h^{k,m,j}).$$

Here, $\delta_{H+1}^k = \zeta_{H+1}^k = 0$. Because $V_h^\star(s) = \sup_\pi V_h^\pi(s)$, we have $\delta_h^k \le \zeta_h^k$ for any $h \in [H+1]$. In addition, as $V_h^k(s) \ge V_h^\star(s)$ for all $(s,h,k) \in \mathcal{S} \times [H] \times [K]$, according to Lemma E.1, we have:

$$\text{Regret}(T) = \sum_{k,m,j} \left( V_1^\star(s_1^{k,m,j}) - V_1^{\pi^k}(s_1^{k,m,j}) \right) \le \sum_{k,m,j} \left( V_1^k(s_1^{k,m,j}) - V_1^{\pi^k}(s_1^{k,m,j}) \right) = \sum_{k=1}^K \zeta_1^k.$$

Thus, we only need to bound $\sum_{k=1}^K \zeta_1^k$. Let:

$$\psi_{h+1}^k = \sum_{m=1}^M \sum_{j=1}^{n^{m,k}} \frac{\mathbb{I}\left[t_h^k(s_h^{k,m,j}, a_h^{k,m,j}) > 1\right]}{N_h^k(s_h^{k,m,j}, a_h^{k,m,j})} \sum_{i=1}^{N_h^k} \mathbb{P}_{s_h^{k,m,j}, a_h^{k,m,j}, h} \left( V_{h+1}^{\text{ref},k_{L_i}} - V_{h+1}^{\text{REF}} \right), \tag{61}$$

$$\epsilon_{h+1}^k = \sum_{m=1}^M \sum_{j=1}^{n^{m,k}} \frac{\mathbb{I}\left[t_h^k(s_h^{k,m,j}, a_h^{k,m,j}) > 1\right]}{n_h^k(s_h^{k,m,j}, a_h^{k,m,j})} \sum_{i=1}^{n_h^k} \left(\mathbb{P}_{s_h^{k,m,j}, a_h^{k,m,j}, h} - \mathbb{1}_{s_{h+1}^{(k,m,j)_{l_i}}}\right)\left(V_{h+1}^{k_{l_i}} - V_{h+1}^\star\right),$$

(62)

$$\phi_{h+1}^k = \sum_{m=1}^M \sum_{j=1}^{n^{m,k}} \mathbb{I}\left[t_h^k(s_h^{k,m,j}, a_h^{k,m,j}) > 1\right] \left(\mathbb{P}_{s_h^{k,m,j}, a_h^{k,m,j}, h} - \mathbb{1}_{s_{h+1}^{k,m,j}}\right)\left(V_{h+1}^\star - V_{h+1}^{\pi^k}\right), \quad (63)$$

where $\psi_{H+1}^k = \epsilon_{H+1}^k = \phi_{H+1}^k = 0$. According to the update rule Equation (27), we have:

$$V_h^k(s_h^{k,m,j}) = \max_{a \in \mathcal{A}} Q_h^k(s_h^{k,m,j}, a) = Q_h^k(s_h^{k,m,j}, a_h^{k,m,j})$$

$$\leq \mathbb{I}\left[t_h^{k,m,j} = 1\right] H + \mathbb{I}\left[t_h^{k,m,j} > 1\right] \tilde{Q}_h^{k,2}(s_h^{k,m,j}, a_h^{k,m,j})$$

$$\leq \mathbb{I}\left[t_h^{k,m,j} = 1\right] H + \mathbb{I}\left[t_h^{k,m,j} > 1\right] \left(r_h(s_h^{k,m,j}, a_h^{k,m,j}) + \frac{\sum_{i=1}^{N_h^k} V_{h+1}^{\mathrm{ref}, k_{L_i}}(s_{h+1}^{(k,m,j)_{L_i}})}{N_h^k(s_h^{k,m,j}, a_h^{k,m,j})}\right.$$

$$\left. + \frac{\sum_{i=1}^{n_h^k}\left(V_{h+1}^{k_{l_i}} - V_{h+1}^{\mathrm{ref}, k_{l_i}}\right)(s_{h+1}^{(k,m,j)_{l_i}})}{n_h^k(s_h^{k,m,j}, a_h^{k,m,j})} + \tilde{b}_h^{k,2}(s_h^{k,m,j}, a_h^{k,m,j})\right).$$

Also using Equation (1), we have:

$$V_h^{\pi^k}(s_h^{k,m,j}) = Q_h^{\pi^k}(s_h^{k,m,j}, a_h^{k,m,j}) \geq \mathbb{I}[t_h^{k,m,j} > 1]\left(r_h(s_h^{k,m,j}, a_h^{k,m,j}) + \mathbb{P}_{s_h^{k,m,j}, a_h^{k,m,j}, h} V_{h+1}^{\pi^k}\right).$$

Then with Equation (30), it holds that:

$$\zeta_h^k = \sum_{m=1}^M \sum_{j=1}^{n^{m,k}} \left(V_h^k - V_h^{\pi^k}\right)(s_h^{k,m,j})$$

$$\leq \sum_{m=1}^M \sum_{j=1}^{n^{m,k}} \left(\mathbb{I}\left[t_h^{k,m,j} = 1\right] H + \mathbb{I}\left[t_h^{k,m,j} > 1\right] \tilde{b}_h^{k,2}(s_h^{k,m,j}, a_h^{k,m,j})\right) + \sum_{m=1}^M \sum_{j=1}^{n^{m,k}} \mathbb{I}\left[t_h^{k,m,j} > 1\right]$$

$$\times \left(\frac{\sum_{i=1}^{N_h^k} V_{h+1}^{\mathrm{ref}, k_{L_i}}(s_{h+1}^{(k,m,j)_{L_i}})}{N_h^k(s_h^{k,m,j}, a_h^{k,m,j})} + \frac{\sum_{i=1}^{n_h^k}\left(V_{h+1}^{k_{l_i}} - V_{h+1}^{\mathrm{ref}, k_{l_i}}\right)(s_{h+1}^{(k,m,j)_{l_i}})}{n_h^k(s_h^{k,m,j}, a_h^{k,m,j})} - \mathbb{P}V_{h+1}^{\pi^k}\right)$$

$$\leq \sum_{m=1}^M \sum_{j=1}^{n^{m,k}} \left(\mathbb{I}\left[t_h^{k,m,j} = 1\right] H + 2\mathbb{I}\left[t_h^{k,m,j} > 1\right] \tilde{b}_h^{k,2}(s_h^{k,m,j}, a_h^{k,m,j})\right) + \sum_{m=1}^M \sum_{j=1}^{n^{m,k}}$$

$$\mathbb{I}\left[t_h^{k,m,j} > 1\right] \left(\frac{\sum_{i=1}^{N_h^k} \mathbb{P}V_{h+1}^{\mathrm{ref}, k_{L_i}}}{N_h^k(s_h^{k,m,j}, a_h^{k,m,j})} + \frac{\sum_{i=1}^{n_h^k} \mathbb{P}\left(V_{h+1}^{k_{l_i}} - V_{h+1}^{\mathrm{ref}, k_{l_i}}\right)}{n_h^k(s_h^{k,m,j}, a_h^{k,m,j})} - \mathbb{P}V_{h+1}^{\pi^k}\right). \quad (64)$$

Here $\mathbb{P}$ is the simplified notation for $\mathbb{P}_{s_h^{k,m,j}, a_h^{k,m,j}, h}$. Because the reference function is non-increasing based on (j) in Lemma D.1, we have $V_{h+1}^{\mathrm{ref}, k_{l_i}}(s) \geq V_{h+1}^{\mathrm{REF}}(s)$ for any $s \in \mathcal{S}$ and any positive integer $i$, $\mathbb{P}_{s_h^{k,m,j}, a_h^{k,m,j}, h} V_{h+1}^{\mathrm{ref}, k_{l_i}} \geq \mathbb{P}_{s_h^{k,m,j}, a_h^{k,m,j}, h} V_{h+1}^{\mathrm{REF}}$ and

$$\frac{\sum_{i=1}^{n_h^k} \mathbb{P}_{s_h^{k,m,j}, a_h^{k,m,j}, h} V_{h+1}^{\mathrm{ref}, k_{l_i}}}{n_h^k(s_h^{k,m,j}, a_h^{k,m,j})} \geq \mathbb{P}_{s_h^{k,m,j}, a_h^{k,m,j}, h} V_{h+1}^{\mathrm{REF}}. \quad (65)$$

According to the definition of $\delta_{h+1}^k, \psi_{h+1}^k, \epsilon_{h+1}^k, \phi_{h+1}^k$ and Equation (65), we have:

$$\sum_{m=1}^M \sum_{j=1}^{n^{m,k}} \mathbb{I}\left[t_h^{k,m,j} > 1\right] \left(\frac{\sum_{i=1}^{N_h^k} \mathbb{P}_{s_h^{k,m,j}, a_h^{k,m,j}, h} V_{h+1}^{\mathrm{ref}, k_{L_i}}}{N_h^k(s_h^{k,m,j}, a_h^{k,m,j})} - \frac{\sum_{i=1}^{n_h^k} \mathbb{P}_{s_h^{k,m,j}, a_h^{k,m,j}, h} V_{h+1}^{\mathrm{ref}, k_{l_i}}}{n_h^k(s_h^{k,m,j}, a_h^{k,m,j})}\right)$$

$$\leq \psi_{h+1}^k, \quad (66)$$

$$\sum_{m=1}^{M} \sum_{j=1}^{n^{m,k}} \mathbb{I}\left[t_h^{k,m,j} > 1\right] \frac{\sum_{i=1}^{n_h^k} \mathbb{P}_{s_h^{k,m,j}, a_h^{k,m,j}, h}\left(V_{h+1}^{k_{l_i}} - V_{h+1}^{\star}\right)}{n_h^k(s_h^{k,m,j}, a_h^{k,m,j})}$$

$$= \epsilon_{h+1}^k + \sum_{m=1}^{M} \sum_{j=1}^{n^{m,k}} \mathbb{I}\left[t_h^{k,m,j} > 1\right] \frac{\sum_{i=1}^{n_h^k}\left(V_{h+1}^{k_{l_i}} - V_{h+1}^{\star}\right)(s_{h+1}^{(k,m,j)_{l_i}})}{n_h^k(s_h^{k,m,j}, a_h^{k,m,j})}, \tag{67}$$

and

$$\sum_{m=1}^{M} \sum_{j=1}^{n^{m,k}} \mathbb{I}\left[t_h^{k,m,j} > 1\right] \mathbb{P}_{s_h^{k,m,j}, a_h^{k,m,j}, h}\left(V_{h+1}^{\star} - V_{h+1}^{\pi^k}\right) = \phi_{h+1}^k + \zeta_{h+1}^k - \delta_{h+1}^k. \tag{68}$$

Summing Equation (66), Equation (67) and Equation (68), we can bound the second term in Equation (64) as follows:

$$\sum_{m=1}^{M} \sum_{j=1}^{n^{m,k}} \mathbb{I}\left[t_h^{k,m,j} > 1\right] \left(\frac{\sum_{i=1}^{N_h^k} \mathbb{P}V_{h+1}^{\mathrm{ref}, k_{L_i}}}{N_h^k(s_h^{k,m,j}, a_h^{k,m,j})} + \frac{\sum_{i=1}^{n_h^k} \mathbb{P}\left(V_{h+1}^{k_{l_i}} - V_{h+1}^{\mathrm{ref}, k_{l_i}}\right)}{n_h^k(s_h^{k,m,j}, a_h^{k,m,j})} - \mathbb{P}V_{h+1}^{\pi^k}\right) \le$$

$$\sum_{m=1}^{M} \sum_{j=1}^{n^{m,k}} \mathbb{I}[t_h^{k,m,j} > 1]\frac{\sum_{i=1}^{n_h^k}\left(V_{h+1}^{k_{l_i}} - V_{h+1}^{\star}\right)(s_{h+1}^{(k,m,j)_{l_i}})}{n_h^k(s_h^{k,m,j}, a_h^{k,m,j})} + \psi_{h+1}^k + \epsilon_{h+1}^k + \phi_{h+1}^k + \zeta_{h+1}^k - \delta_{h+1}^k.$$

Together with Equation (64), we have

$$\zeta_h^k \le \sum_{m=1}^{M} \sum_{j=1}^{n^{m,k}} (\mathbb{I}\left[t_h^{k,m,j} = 1\right] H + 2\mathbb{I}\left[t_h^{k,m,j} > 1\right] \tilde{b}_h^{k,2}) + \sum_{m=1}^{M} \sum_{j=1}^{n^{m,k}} \mathbb{I}\left[t_h^{k,m,j} > 1\right] \times$$

$$\frac{\sum_{i=1}^{n_h^k}\left(V_{h+1}^{k_{l_i}} - V_{h+1}^{\star}\right)(s_{h+1}^{(k,m,j)_{l_i}})}{n_h^k(s_h^{k,m,j}, a_h^{k,m,j})} + \psi_{h+1}^k + \epsilon_{h+1}^k + \phi_{h+1}^k + \zeta_{h+1}^k - \delta_{h+1}^k.$$

Summing the above inequality for $k = 1, 2, .., K$, we have:

$$\sum_{k=1}^{K} \zeta_h^k \le \sum_{k,m,j} (\mathbb{I}\left[t_h^{k,m,j} = 1\right] H + 2\mathbb{I}\left[t_h^{k,m,j} > 1\right] \tilde{b}_h^{k,2}(s_h^{k,m,j}, a_h^{k,m,j})) + \sum_{k,m,j} \mathbb{I}\left[t_h^{k,m,j} > 1\right] \times$$

$$\frac{\sum_{i=1}^{n_h^k}\left(V_{h+1}^{k_{l_i}} - V^{\star}\right)(s_{h+1}^{(k,m,j)_{l_i}})}{n_h^k(s_h^{k,m,j}, a_h^{k,m,j})} + \sum_{k=1}^{K}(\psi_{h+1}^k + \epsilon_{h+1}^k + \phi_{h+1}^k + \zeta_{h+1}^k - \delta_{h+1}^k). \tag{69}$$

We claim the following conclusions:

$$\sum_{k,m,j} \mathbb{I}\left[t_h^{k,m,j} = 1\right] H \le 2MH^2 SA,$$

$$\sum_{k,m,j} \mathbb{I}\left[t_h^{k,m,j} > 1\right] \frac{\sum_{i=1}^{n_h^k}\left(V_{h+1}^{k_{l_i}} - V_{h+1}^{\star}\right)(s_{h+1}^{(k,m,j)_{l_i}})}{n_h^k(s_h^{k,m,j}, a_h^{k,m,j})} \le (1 + \frac{2}{H})\sum_{k=1}^{K} \delta_{h+1}^k. \tag{70}$$

The first conclusion has been proved in Equation (47), and we will prove the second conclusion in Lemma E.3 in the last subsection. Applying the two conclusions to Equation (69), it holds:

$$\sum_{k=1}^{K} \zeta_h^k \le 2MH^2 SA + (1 + \frac{2}{H})\sum_{k=1}^{K} \delta_{h+1}^k + \sum_{k=1}^{K}(\psi_{h+1}^k + \epsilon_{h+1}^k + \phi_{h+1}^k + \zeta_{h+1}^k - \delta_{h+1}^k)$$

$$+ 2\sum_{k,m,j} \mathbb{I}\left[t_h^{k,m,j} > 1\right] \tilde{b}_h^{k,2}(s_h^{k,m,j}, a_h^{k,m,j})$$

$$\le 2MH^2 SA + (1 + \frac{2}{H})\sum_{k=1}^{K} \zeta_{h+1}^k + \sum_{k=1}^{K}(\psi_{h+1}^k + \epsilon_{h+1}^k + \phi_{h+1}^k)$$

$$+ 2\sum_{k,m,j} \mathbb{I}\left[t_h^{k,m,j} > 1\right] \tilde{b}_h^{k,2}(s_h^{k,m,j}, a_h^{k,m,j}).$$

Here, the last inequality is because $\delta_{h+1}^k \le \zeta_{h+1}^k$. By recursion on $H, H-1, H-2\dots,1$, with $\zeta_{H+1}^K = 0$, we have:

$$
\begin{aligned}
\sum_{k=1}^{K}\zeta_1^k &\le 2\sum_{h=1}^{H}(1+\frac{2}{H})^{h-1}MH^2SA + \sum_{h=1}^{H}\sum_{k=1}^{K}(1+\frac{2}{H})^{h-1}(\psi_{h+1}^k + \epsilon_{h+1}^k + \phi_{h+1}^k)\\
&\quad + 2\sum_{h=1}^{H}(1+\frac{2}{H})^{h-1}\sum_{k,m,j}\mathbb{I}\left[t_h^{k,m,j} > 1\right]\tilde{b}_h^{k,2}(s_h^{k,m,j},a_h^{k,m,j})\\
&\le 18MH^3SA + \sum_{h=1}^{H}\sum_{k=1}^{K}(1+\frac{2}{H})^{h-1}(\psi_{h+1}^k + \epsilon_{h+1}^k + \phi_{h+1}^k)\\
&\quad + 18\sum_{h=1}^{H}\sum_{k,m,j}\mathbb{I}\left[t_h^{k,m,j} > 1\right]\tilde{b}_h^{k,2}(s_h^{k,m,j},a_h^{k,m,j})\\
&= O(MH^3SA + \sum_{h=1}^{H}\sum_{k=1}^{K}(1+\frac{2}{H})^{h-1}(\psi_{h+1}^k + \epsilon_{h+1}^k + \phi_{h+1}^k)\\
&\quad + \sum_{h=1}^{H}\sum_{k,m,j}\mathbb{I}\left[t_h^{k,m,j} > 1\right]\tilde{b}_h^{k,2}(s_h^{k,m,j},a_h^{k,m,j})).
\end{aligned}
\tag{71}
$$

Here, the second inequality is because $(1+\frac{2}{H})^{h-1} \le (1+\frac{2}{H})^H \le e^2 < 9$. Based on the Lemma E.4, Lemma E.5, Lemma E.6, and Lemma E.7 provided in the last subsection, we have:

$$
\sum_{h=1}^{H}\sum_{k=1}^{K}(1+\frac{2}{H})^{h-1}\psi_{h+1}^k \le O\left(H\sqrt{T_1\iota}\log(T_1) + H^2SN_0\log(T_1)\right),
$$

$$
\sum_{h=1}^{H}\sum_{k=1}^{K}(1+\frac{2}{H})^{h-1}\epsilon_{h+1}^k \le O\left(\sqrt{SAH^2T_1\iota} + SAH^{\frac{5}{2}}(M\iota)^{\frac{1}{2}}\right),
$$

$$
\sum_{h=1}^{H}\sum_{k=1}^{K}(1+\frac{2}{H})^{h-1}\phi_{h+1}^k \le O(H\sqrt{T_1\iota}),
$$

$$
\sum_{h=1}^{H}\sum_{k,m,j}\mathbb{I}\left[t_h^{k,m,j} > 1\right]\tilde{b}_h^{k,2}(s_h^{k,m,j},a_h^{k,m,j}) \le
$$
$$
O\left(\sqrt{SAH^2T_1\iota} + \sqrt{\beta SAH^2T_1\iota} + \sqrt{\beta^2 SAH^2T_1\iota} + SAH^{\frac{11}{4}}T_1^{\frac{1}{4}}\iota^{\frac{3}{4}} + S^{\frac{3}{2}}AH^3\sqrt{N_0}\log(T_1)\iota\right).
$$

Inserting these relationships into Equation (71), we have

$$
\begin{aligned}
&\text{Regret}(T)\\
&\le O((1+\sqrt{\beta}+\beta)\sqrt{SAH^2T_1\iota} + H\sqrt{T_1\iota}\log(T_1) + SAH^{\frac{11}{4}}T_1^{\frac{1}{4}}\iota^{\frac{3}{4}} + SH^2N_0\log(T_1)\\
&\quad + S^{\frac{3}{2}}AH^3\sqrt{N_0}\log(T_1)\iota + SAH^{\frac{5}{2}}(M\iota)^{\frac{1}{2}} + MSAH^3)\\
&\le O((1+\sqrt{\beta}+\beta)\sqrt{MSAH^2T\iota} + H\sqrt{MT\iota}\log(T) + H\sqrt{MT\iota}\log(MSAH^2)\\
&\quad + M^{\frac{1}{4}}SAH^{\frac{11}{4}}T^{\frac{1}{4}}\iota^{\frac{3}{4}} + SH^2N_0\log(T) + \sqrt{MSA}H^2\log(T)\sqrt{\iota} + S^{\frac{3}{2}}AH^3\sqrt{N_0}\log(T)\iota\\
&\quad + SAH^{\frac{5}{2}}(M\iota)^{\frac{1}{2}} + MSAH^2\sqrt{\iota} + MSAH^2\sqrt{\beta\iota} + MSAH^2\sqrt{\beta^2\iota} + M^{\frac{1}{4}}S^{\frac{5}{4}}A^{\frac{5}{4}}H^{\frac{13}{4}}\iota^{\frac{3}{4}}\\
&\quad + S^{\frac{3}{2}}AH^3\sqrt{N_0}\log(MSAH^2)\iota + SH^2N_0\log(MSAH^2) + \sqrt{MSA}H^2\log(MSAH^2)\sqrt{\iota}).
\end{aligned}
$$

In the last step, we use $T_1 \le (2+2/H)MT + MSAH(H+1)$ according to (i) in Lemma D.1. This finishes the proof of Theorem 4.1

### E.1 PROOF OF SOME INDIVIDUAL COMPONENT

This subsection collects the proof of some individual components for Theorem 4.1.

**Lemma E.3** (Proof of Equation (70)). *Under the event $\bigcap_{i=1}^{15} \mathcal{E}_i$, we have that Equation (70) holds.*

*Proof.*

$$
\sum_{k,m,j} \mathbb{I}\left[t_h^{k,m,j} > 1\right] \frac{\sum_{i=1}^{n_h^k}\left(V_{h+1}^{k_{l_i}} - V_{h+1}^{\star}\right)(s_{h+1}^{(k,m,j)_{l_i}})}{n_h^k(s_h^{k,m,j}, a_h^{k,m,j})}
$$

$$
= \sum_{k,m,j} \sum_{i=1}^{n_h^k} \mathbb{I}\left[t_h^{k,m,j} > 1\right] \frac{\left(V_{h+1}^{k_{l_i}} - V_{h+1}^{\star}\right)(s_{h+1}^{(k,m,j)_{l_i}})}{n_h^k(s_h^{k,m,j}, a_h^{k,m,j})} \left(\sum_{k',m',j'} \mathbb{I}[(k,m,j)_{l_i} = (k',m',j')]\right)
$$

$$
= \sum_{k,m,j} \sum_{k',m',j'} \sum_{i=1}^{n_h^k} \frac{(V_{h+1}^{k'} - V_{h+1}^{\star})(s_{h+1}^{k',m',j'})}{n_h^k(s_h^{k,m,j}, a_h^{k,m,j})} \mathbb{I}[(k,m,j)_{l_i} = (k',m',j'), t_h^{k,m,j} > 1]
$$

$$
= \sum_{k',m',j'} \sum_{k,m,j} \left(\sum_{i=1}^{n_h^k} \mathbb{I}[(k,m,j)_{l_i} = (k',m',j'), t_h^{k,m,j} > 1]\right) \frac{(V_{h+1}^{k'} - V_{h+1}^{\star})(s_{h+1}^{k',m',j'})}{n_h^k(s_h^{k,m,j}, a_h^{k,m,j})}
$$

$$
= \sum_{k',m',j'} \left(\sum_{k,m,j} \frac{\sum_{i=1}^{n_h^k} \mathbb{I}[(k,m,j)_{l_i} = (k',m',j'), t_h^{k,m,j} > 1]}{n_h^k(s_h^{k,m,j}, a_h^{k,m,j})}\right) (V_{h+1}^{k'} - V_{h+1}^{\star})(s_{h+1}^{k',m',j'}).
$$

For a given triple $(k', m', j')$, according to the definition of $l_i(s_h^{k,m,j}, a_h^{k,m,j}, h, k)$, $\sum_{i=1}^{n_h^k} \mathbb{I}[(k,m,j)_{l_i} = (k',m',j'), t_h^{k,m,j} > 1]) = 1$ if and only if $(s_h^{k,m,j}, a_h^{k,m,j}) = (s_h^{k',m',j'}, a_h^{k',m',j'})$ and $1 < t_h^k(s_h^{k,m,j}, a_h^{k,m,j}) = t_h^{k'}(s_h^{k,m,j}, a_h^{k,m,j}) + 1$. In this case, we have $n_h^k(s_h^{k,m,j}, a_h^{k,m,j}) = y_h^{t_h^{k'}}(s_h^{k',m',j'}, a_h^{k',m',j'})$ and

$$
\sum_{k,m,j} \sum_{i=1}^{n_h^k} \mathbb{I}[(k,m,j)_{l_i} = (k',m',j'), t_h^{k,m,j} > 1]
$$

$$
= \sum_{k,m,j} \mathbb{I}\left[(s_h^{k,m,j}, a_h^{k,m,j}) = (s_h^{k',m',j'}, a_h^{k',m',j'}), t_h^k(s_h^{k,m,j}, a_h^{k,m,j}) = t_h^{k'}(s_h^{k,m,j}, a_h^{k,m,j}) + 1\right]
$$

$$
= y_h^{t_h^{k'}+1}(s_h^{k',m',j'}, a_h^{k',m',j'}).
$$

Then according to (f) in Lemma D.1:

$$
\sum_{k,m,j} \frac{\sum_{i=1}^{n_h^k} \mathbb{I}\left[(k,m,j)_{l_i} = (k',m',j'), t_h^{k,m,j} > 1\right]}{n_h^k(s_h^{k,m,j}, a_h^{k,m,j})} = \frac{y_h^{t_h^{k'}+1}(s_h^{k',m',j'}, a_h^{k',m',j'})}{y_h^{t_h^{k'}}(s_h^{k',m',j'}, a_h^{k',m',j'})} \leq 1 + \frac{2}{H}.
$$

Therefore, since $V_{h+1}^{k'} \geq V_{h+1}^{\star}$ according to Lemma E.1, we have:

$$
\sum_{k,m,j} \mathbb{I}\left[t_h^{k,m,j} > 1\right] \frac{\sum_{i=1}^{n_h^k}\left(V_{h+1}^{k_{l_i}} - V_{h+1}^{\star}\right)(s_{h+1}^{(k,m,j)_{l_i}})}{n_h^k(s_h^{k,m,j}, a_h^{k,m,j})}
$$

$$
\leq (1 + \frac{2}{H}) \sum_{k',m',j'} (V_{h+1}^{k'} - V_{h+1}^{\star})(s_{h+1}^{k',m',j'})
$$

$$
= (1 + \frac{2}{H}) \sum_{k=1}^{K} \delta_{h+1}^k.
$$

$\square$

Next, we will give lemmas on the upper bounds of each term in Equation (71).

**Lemma E.4.** *Under the event $\mathcal{E}_8$ in Lemma D.5, it holds that:*

$$\sum_{h=1}^{H}\sum_{k=1}^{K}(1+\frac{2}{H})^{h-1}\psi_{h+1}^{k} \leq O\left(H\sqrt{T_1}\iota\log(T_1) + H^2SN_0\log(T_1)\right).$$

*Proof.* For any $(s,h,k) \in \mathcal{S} \times [H] \times [K]$, if $N_h^k(s) = \sum_{a\in\mathcal{A}} N_h^k(s,a) \geq N_0$, the reference function $V_h^{\text{ref},k}(s)$ is updated to its final value with $V_h^{\text{ref},k}(s) = V_h^{\text{REF}}(s)$. If $N_h^k(s) < N_0$, since the reference function is non-increasing and $V_h^{\text{ref},1}(s) = H$, we have $0 \leq V_h^{\text{ref},k}(s) - V_h^{\text{REF}}(s) \leq H$. Combining two cases, for any $(s,h,k) \in \mathcal{S} \times [H] \times [K]$, it holds that $0 \leq V_h^{\text{ref},k}(s) - V_h^{\text{REF}}(s) \leq H\lambda_h^k(s)$, where $\lambda_h^k(s) = \mathbb{I}[N_h^k(s) < N_0]$ is defined in the event $\mathcal{E}_8$ in Lemma D.5. The conclusion also holds for $h = H+1$ because $V_{H+1}^{\text{ref},k}(s) = V_{H+1}^{\text{REF}}(s) = 0$. Then for any $(s,a,h,k) \in \mathcal{S} \times \mathcal{A} \times [H] \times [K]$ we have:

$$0 \leq \mathbb{P}_{s,a,h}\left(V_{h+1}^{\text{ref},k} - V_{h+1}^{\text{REF}}\right) \leq H\mathbb{P}_{s,a,h}\lambda_{h+1}^k. \tag{72}$$

Applying Equation (72) to the definition of $\psi_{h+1}^k$ Equation (61), we have:

$$\sum_{k=1}^{K}\psi_{h+1}^{k}$$

$$= \sum_{k,m,j}\mathbb{I}\left[t_h^{k,m,j} > 1\right]\frac{\sum_{i=1}^{N_h^k}\mathbb{P}_{s_h^{k,m,j},a_h^{k,m,j},h}(V_{h+1}^{\text{ref},k_{L_i}} - V_{h+1}^{\text{REF}})}{N_h^k(s_h^{k,m,j},a_h^{k,m,j})}$$

$$\leq H\sum_{k,m,j}\mathbb{I}\left[t_h^{k,m,j} > 1\right]\frac{\sum_{i=1}^{N_h^k}\mathbb{P}_{s_h^{k,m,j},a_h^{k,m,j},h}\lambda_{h+1}^{k_{L_i}}}{N_h^k(s_h^{k,m,j},a_h^{k,m,j})}$$

$$= H\sum_{k,m,j}\mathbb{I}\left[t_h^{k,m,j} > 1\right]\frac{\sum_{i=1}^{N_h^k}\mathbb{P}_{s_h^{k,m,j},a_h^{k,m,j},h}\lambda_{h+1}^{k_{L_i}}\left(\sum_{k',m',j'}\mathbb{I}[(k,m,j)_{L_i} = (k',m',j')]\right)}{N_h^k(s_h^{k,m,j},a_h^{k,m,j})}$$

$$= H\sum_{k,m,j}\sum_{k',m',j'}\frac{\sum_{i=1}^{N_h^k}\mathbb{I}[(k,m,j)_{L_i} = (k',m',j'),t_h^{k,m,j} > 1]\mathbb{P}_{s_h^{k,m,j},a_h^{k,m,j},h}\lambda_{h+1}^{k'}}{N_h^k(s_h^{k',m',j'},a_h^{k',m',j'})}.$$

According to the definition of $L_i(s_h^{k,m,j},a_h^{k,m,j},h)$, for a given triple $(k',m',j')$, $\sum_{i=1}^{N_h^k}\mathbb{I}[(k,m,j)_{L_i} = (k',m',j'),t_h^{k,m,j} > 1] = 1$ if and only if $(s_h^{k,m,j},a_h^{k,m,j}) = (s_h^{k',m',j'},a_h^{k',m',j'})$ and $1 \leq t_h^{k'}(s_h^{k,m,j},a_h^{k,m,j}) < t_h^k(s_h^{k,m,j},a_h^{k,m,j})$. Then we have $t_h^{k,m,j} = t_h^k(s_h^{k',m',j'},a_h^{k',m',j'})$. For $t > t_h^{k'}(s_h^{k',m',j'},a_h^{k',m',j'}) \geq 1$, we also have:

$$\sum_{k,m,j}\sum_{i=1}^{N_h^k}\mathbb{I}\left[(k,m,j)_{L_i} = (k',m',j'),t_h^k(s_h^{k',m',j'},a_h^{k',m',j'}) = t\right]$$

$$= \sum_{k,m,j}\mathbb{I}\left[(s_h^{k,m,j},a_h^{k,m,j}) = (s_h^{k',m',j'},a_h^{k',m',j'}),t_h^k(s_h^{k',m',j'},a_h^{k',m',j'}) = t\right]$$

$$= y_h^t(s_h^{k',m',j'},a_h^{k',m',j'}). \tag{73}$$

Let:

$$W_{k',m',j'} = \sum_{k,m,j}\frac{\sum_{i=1}^{N_h^k}\mathbb{I}[(k,m,j)_{L_i} = (k',m',j'),t_h^{k,m,j} > 1]}{N_h^k(s_h^{k',m',j'},a_h^{k',m',j'})}.$$

Then, since $(1+\frac{2}{H})^{h-1} \leq (1+\frac{2}{H})^H < e^2 < 9$, we have:

$$\sum_{h=1}^{H}\sum_{k=1}^{K}(1+\frac{2}{H})^{h-1}\psi_{h+1}^k \leq 9H\sum_{h=1}^{H}\sum_{k',m',j'}W_{k',m',j'}\mathbb{P}_{s_h^{k',m',j'},a_h^{k',m',j'},h}\lambda_{h+1}^{k'}. \tag{74}$$

Applying Equation (73), we have:

$$
\begin{aligned}
W_{k',m',j'} &= \sum_{k,m,j} \frac{\sum_{i=1}^{N_h^k} \mathbb{I}[(k,m,j)_{L_i} = (k',m',j')]}{N_h^k(s_h^{k',m',j'}, a_h^{k',m',j'})} \left( \sum_{t=t_h^{k'}+1}^{T_h} \mathbb{I}\left[ t_h^k(s_h^{k',m',j'}, a_h^{k',m',j'}) = t \right] \right) \\
&= \sum_{k,m,j} \sum_{t=t_h^{k'}+1}^{T_h} \frac{\sum_{i=1}^{N_h^k} \mathbb{I}[(k,m,j)_{L_i} = (k',m',j')]}{Y_h^{t-1}(s_h^{k',m',j'}, a_h^{k',m',j'})} \mathbb{I}\left[ t_h^k(s_h^{k',m',j'}, a_h^{k',m',j'}) = t \right] \\
&= \sum_{t=t_h^{k'}+1}^{T_h} \frac{\sum_{k,m,j} \sum_{i=1}^{N_h^k} \mathbb{I}\left[(k,m,j)_{L_i} = (k',m',j'), t_h^k(s_h^{k',m',j'}, a_h^{k',m',j'}) = t\right]}{Y_h^{t-1}(s_h^{k',m',j'}, a_h^{k',m',j'})} \\
&= \sum_{t=t_h^{k'}+1}^{T_h} \frac{y_h^t(s_h^{k',m',j'}, a_h^{k',m',j'})}{Y_h^{t-1}(s_h^{k',m',j'}, a_h^{k',m',j'})}.
\end{aligned}
$$

According to (f) in Lemma D.1, for any $(s,a,h) \in \mathcal{S} \times \mathcal{A} \times [H]$, $t \in [2, T_h(s,a)]$ and $1 \le p \le y_h^t(s,a)$, we have:

$$
Y_h^{t-1}(s,a) + p \le Y_h^t(s,a) \le (2 + \frac{2}{H}) Y_h^{t-1}(s,a).
$$

Then it holds that:

$$
\begin{aligned}
W_{k',m',j'} &= \sum_{t=t_h^{k'}+1}^{T_h} \frac{y_h^t(s_h^{k',m',j'}, a_h^{k',m',j'})}{Y_h^{t-1}(s_h^{k',m',j'}, a_h^{k',m',j'})} \\
&= \sum_{t=t_h^{k'}+1}^{T_h} \sum_{p=1}^{y_h^t} \frac{1}{Y_h^{t-1}(s_h^{k',m',j'}, a_h^{k',m',j'})} \\
&\le (2 + \frac{2}{H}) \sum_{t=t_h^{k'}+1}^{T_h} \sum_{p=1}^{y_h^t} \frac{1}{Y_h^{t-1}(s_h^{k',m',j'}, a_h^{k',m',j'}) + p} \\
&\le (2 + \frac{2}{H}) \sum_{q=1}^{T_1} \frac{1}{q} \le 4(\log(T_1) + 1).
\end{aligned}
$$

Applying the inequality of the coefficient $W_{k',m',j'}$ to Equation (74), we have:

$$
\begin{aligned}
&\sum_{h=1}^{H} \sum_{k=1}^{K} (1 + \frac{2}{H})^{h-1} \psi_{h+1}^k \\
&= 9H \sum_{h=1}^{H} \sum_{k',m',j'} W_{k',m',j'} \mathbb{P}_{s_h^{k',m',j'}, a_h^{k',m',j'}, h} \lambda_{h+1}^{k'}(s_{h+1}^{k',m',j'}) \\
&\le 9H \cdot 4(\log(T_1) + 1) \sum_{h=1}^{H} \sum_{k',m',j'} \mathbb{P}_{s_h^{k',m',j'}, a_h^{k',m',j'}, h} \lambda_{h+1}^{k'}(s_{h+1}^{k',m',j'}) \\
&= 36H(\log(T_1) + 1) \sum_{h=1}^{H} \sum_{k,m,j} \left( \lambda_{h+1}^k(s_{h+1}^{k,m,j}) + \left( \mathbb{P}_{s_h^{k,m,j}, a_h^{k,m,j}, h} - \mathbb{1}_{s_{h+1}^{k,m,j}} \right) \lambda_{h+1}^k(s_{h+1}^{k,m,j}) \right) \\
&\le 36H(\log(T_1) + 1) \left( \sum_{h=1}^{H} \sum_{k,m,j} \lambda_{h+1}^k(s_{h+1}^{k,m,j}) + \sqrt{2T_1 \iota} \right).
\end{aligned}
\tag{75}
$$

The last inequality is because of the event $\mathcal{E}_8$ in Lemma D.5. Next, we will bound the term $\sum_{h=1}^{H} \sum_{k,m,j} \lambda_{h+1}^k(s_{h+1}^{k,m,j})$ in Equation (75). We have:

$$
\begin{aligned}
\sum_{h=1}^{H} \sum_{k,m,j} \lambda_{h+1}^k(s_{h+1}^{k,m,j}) &= \sum_{h=1}^{H} \sum_{k,m,j} \lambda_{h+1}^k(s_{h+1}^{k,m,j}) \left( \sum_{s\in\mathcal{S}} \mathbb{I}\left[s_{h+1}^{k,m,j} = s\right] \right) \\
&= \sum_{h=1}^{H} \sum_{s\in\mathcal{S}} \sum_{k,m,j} \lambda_{h+1}^k(s) \mathbb{I}\left[s_{h+1}^{k,m,j} = s\right] \\
&= \sum_{h=2}^{H+1} \sum_{s\in\mathcal{S}} \sum_{k,m,j} \mathbb{I}\left[N_h^k(s) < N_0, s_h^{k,m,j} = s\right].
\end{aligned}
\tag{76}
$$

For any state $(s,h) \in \mathcal{S} \times [H]$ and $k \in [K]$, there exists the largest positive integer $k_0$ such that $N_h^{k_0}(s) < N_0$. Then for any $h \in [H+1]$, it holds that

$$
\begin{aligned}
\sum_{k,m,j} \mathbb{I}\left[N_h^k(s) < N_0, s_h^{k,m,j} = s\right] &= \sum_{k,m,j} \mathbb{I}\left[k \le k_0, s_h^{k,m,j} = s\right] \\
&= \sum_{k=1}^{k_0} \sum_{m,j} \mathbb{I}\left[s_h^{k,m,j} = s\right] \\
&= \sum_{k=1}^{k_0} \sum_{m,j} \sum_{a\in\mathcal{A}} \mathbb{I}\left[s_h^{k,m,j} = s, a_h^{k,m,j} = a\right] \\
&\le \sum_{a\in\mathcal{A}} Y_h^{t_h^{k_0}}(s,a).
\end{aligned}
\tag{77}
$$

However, according to the definition of $N_h^{k_0}(s)$, we have:

$$
\begin{aligned}
N_0 > N_h^{k_0}(s) = \sum_{a\in\mathcal{A}} N_h^{k_0}(s,a) &\ge \sum_{a\in\mathcal{A}} Y_h^{t_h^{k_0}-1}(s,a) \mathbb{I}[t_h^{k_0}(s,a) > 1] \\
&\ge \frac{1}{4} \sum_{a\in\mathcal{A}} Y_h^{t_h^{k_0}}(s,a) \mathbb{I}[t_h^{k_0}(s,a) > 1].
\end{aligned}
$$

The last inequality is because of (f) in Lemma D.1. Combined with Equation (77), we have:

$$
\begin{aligned}
&\sum_{k,m,j} \mathbb{I}\left[N_h^k(s) < N_0, s_h^{k,m,j} = s\right] \\
&= \sum_{a\in\mathcal{A}} Y_h^{t_h^{k_0}}(s,a) \mathbb{I}[t_h^{k_0}(s,a) > 1] + \sum_{a\in\mathcal{A}} Y_h^{t_h^{k_0}}(s,a) \mathbb{I}[t_h^{k_0}(s,a) = 1] \\
&\le 4N_0 + MA(H+1) \le 5N_0.
\end{aligned}
\tag{78}
$$

Here, the last inequality holds because $\beta \le H$. Applying the inequality Equation (78) to Equation (76), we have:

$$
\sum_{h=1}^{H} \sum_{k,m,j} \lambda_{h+1}^k(s_{h+1}^{k,m,j}) \le \sum_{h=2}^{H+1} \sum_{s\in\mathcal{S}} 5N_0 \le 5SHN_0.
$$

Therefore, we bound the term $\sum_{h=1}^{H} \sum_{k,m,j} \lambda_{h+1}^k(s_{h+1}^{k,m,j})$. Back to Equation (75), we have that

$$
\begin{aligned}
\sum_{h=1}^{H} \sum_{k=1}^{K} (1+\frac{2}{H})^{h-1} \psi_{h+1}^k &\le 36H \left(\log(T_1)+1\right) \left(5SHN_0 + \sqrt{2T_1\iota}\right) \\
&= O\left(H\sqrt{T_1\iota}\log(T_1) + SH^2N_0\log(T_1)\right).
\end{aligned}
$$

$\square$

**Lemma E.5.** *Under the event* $\mathcal{E}_9 \cap \mathcal{E}_{10}$*, it holds:*

$$\sum_{h=1}^{H}\sum_{k=1}^{K}(1+\frac{2}{H})^{h-1}\epsilon_{h+1}^{k} \leq O\left(\sqrt{SAH^2 T_1 \iota} + SAH^{\frac{5}{2}}(M\iota)^{\frac{1}{2}}\right).$$

*Proof.* According to the definition of $\epsilon_{h+1}^{k}$ Equation (62), we have:

$$\begin{aligned}
\sum_{k=1}^{K}\epsilon_{h+1}^{k} &= \sum_{k,m,j}\mathbb{I}\left[t_h^{k,m,j} > 1\right]\frac{\sum_{i=1}^{n_h^k}\left(\mathbb{P}_{s_h^{k,m,j},a_h^{k,m,j},h} - \mathbb{1}_{s_{h+1}^{(k,m,j)_{l_i}}}\right)\left(V_{h+1}^{k_{l_i}} - V_{h+1}^{\star}\right)}{n_h^k(s_h^{k,m,j},a_h^{k,m,j})} \\
&= \sum_{k,m,j}\sum_{k',m',j'}\frac{\sum_{i=1}^{n_h^k}\left(\mathbb{P}_{s_h^{k,m,j},a_h^{k,m,j},h} - \mathbb{1}_{s_{h+1}^{(k,m,j)_{l_i}}}\right)\left(V_{h+1}^{k_{l_i}} - V_{h+1}^{\star}\right)}{n_h^k(s_h^{k,m,j},a_h^{k,m,j})} \times \\
&\quad \mathbb{I}\left[(k,m,j)_{l_i} = (k',m',j'),t_h^{k,m,j} > 1\right] \\
&= \sum_{k',m',j'}\sum_{k,m,j}\frac{\sum_{i=1}^{n_h^k}\mathbb{I}[(k,m,j)_{l_i} = (k',m',j'),t_h^{k,m,j} > 1]}{n_h^k(s_h^{k,m,j},a_h^{k,m,j})} \times \\
&\quad \left(\mathbb{P}_{s_h^{k,m,j},a_h^{k,m,j},h} - \mathbb{1}_{s_{h+1}^{k',m',j'}}\right)\left(V_{h+1}^{k'} - V_{h+1}^{\star}\right).
\end{aligned} \tag{79}$$

For a given triple $(k',m',j')$, according to the definition of $l_i(s_h^{k,m,j},a_h^{k,m,j},h,k)$, $\sum_{i=1}^{n_h^k}\mathbb{I}[(k,m,j)_{l_i} = (k',m',j')]) = 1$ if and only if $(s_h^{k,m,j},a_h^{k,m,j}) = (s_h^{k',m',j'},a_h^{k',m',j'})$ and $t_h^k(s_h^{k,m,j},a_h^{k,m,j}) = t_h^{k'}(s_h^{k,m,j},a_h^{k,m,j}) + 1$. In this case, we have $n_h^k(s_h^{k,m,j},a_h^{k,m,j}) = y_h^{t_h^{k'}}(s_h^{k',m',j'},a_h^{k',m',j'})$ and:

$$\begin{aligned}
&\sum_{k,m,j}\sum_{i=1}^{n_h^k}\mathbb{I}[(k,m,j)_{l_i} = (k',m',j'),t_h^{k,m,j} > 1] \\
&= \sum_{k,m,j}\mathbb{I}\left[(s_h^{k,m,j},a_h^{k,m,j}) = (s_h^{k',m',j'},a_h^{k',m',j'}),1 < t_h^{k,m,j} = t_h^{k'}(s_h^{k,m,j},a_h^{k,m,j}) + 1\right] \\
&= y_h^{t_h^{k'}+1}(s_h^{k',m',j'},a_h^{k',m',j'}).
\end{aligned}$$

Let:

$$y(s,a,h,t) = (1+\frac{2}{H})^{h-1}\left(\frac{y_h^{t+1}(s,a)}{y_h^t(s,a)} - 1 - \frac{1}{H}\right).$$

Applying the equation to Equation (79), it holds that:

$$\begin{aligned}
&\sum_{h=1}^{H}\sum_{k=1}^{K}(1+\frac{2}{H})^{h-1}\epsilon_{h+1}^{k} \\
&= \sum_{h=1}^{H}\sum_{k',m',j'}(1+\frac{2}{H})^{h-1}\frac{y_h^{t_h^{k'}+1}(s_h^{k',m',j'},a_h^{k',m',j'})}{y_h^{t_h^{k'}}(s_h^{k',m',j'},a_h^{k',m',j'})} \times \\
&\quad \left(\mathbb{P}_{s_h^{k',m',j'},a_h^{k',m',j'},h} - \mathbb{1}_{s_{h+1}^{k',m',j'}}\right)\left(V_{h+1}^{k'} - V_{h+1}^{\star}\right) \\
&= \sum_{h=1}^{H}\sum_{k,m,j}y(s_h^{k,m,j},a_h^{k,m,j},h,t_h^k)\left(\mathbb{P}_{s_h^{k,m,j},a_h^{k,m,j},h} - \mathbb{1}_{s_{h+1}^{k,m,j}}\right)\left(V_{h+1}^{k} - V_{h+1}^{\star}\right) \\
&\quad + \sum_{h=1}^{H}(1+\frac{2}{H})^{h-1}(1+\frac{1}{H})\sum_{k,m,j}\left(\mathbb{P}_{s,a,h} - \mathbb{1}_{s_{h+1}^{k,m,j}}\right)\left(V_{h+1}^{k} - V_{h+1}^{\star}\right).
\end{aligned}$$

$$\tag{80}$$

$$\tag{81}$$

The term in Equation (80) is a summation of non-martingale difference, and we cannot directly use Azuma-Hoeffding inequality to bound it. Therefore, we split the term with a constant coefficient $1 + \frac{1}{H}$, which can be bounded directly by Azuma-Hoeffding inequality. According to the event $\mathcal{E}_9$ in Lemma D.5, we can bound the second term in Equation (81) with $18H\sqrt{2T_1\iota}$.

We claim that for any $t \in [T_h(s,a)]$, it holds that $|y(s,a,h,t)| \leq 9$ with the proof as follows. According to (e) in Lemma D.1, since $(1 + \frac{2}{H})^H \leq 9$, for any $h \in [H]$ and $1 \leq t \leq T_h(s,a) - 2$, we have $0 \leq y(s,a,h,t) \leq (1 + \frac{2}{H})^{h-1}\frac{1}{H} \leq \frac{9}{H}$. For $t = T_h(s,a) - 1$, we have $-9 \leq -(1 + \frac{2}{H})^{h-1}(1 + \frac{1}{H}) \leq y(s,a,h,T_h - 1) \leq (1 + \frac{2}{H})^{h-1}\frac{1}{H} \leq \frac{9}{H}$. For $t = T_h(s,a)$, since $y_h^{T_h+1}(s,a) = 0$, we have $y(s,a,h,T_h) = -(1 + \frac{2}{H})^{h-1}(1 + \frac{1}{H}) \in [-9,0]$.

Now we will deal with the first term in Equation (81):

$$\sum_{h=1}^{H}\sum_{k,m,j} y(s_h^{k,m,j}, a_h^{k,m,j}, h, t_h^k)\left(\mathbb{P}_{s_h^{k,m,j},a_h^{k,m,j},h} - \mathbb{1}_{s_{h+1}^{k,m,j}}\right)\left(V_{h+1}^k - V_{h+1}^\star\right)$$

$$= \sum_{s,a,h}\sum_{k,m,j} y(s,a,h,t_h^k)\left(\mathbb{P}_{s,a,h} - \mathbb{1}_{s_{h+1}^{k,m,j}}\right)\left(V_{h+1}^k - V_{h+1}^\star\right)\mathbb{I}\left[(s_h^{k,m,j},a_h^{k,m,j}) = (s,a)\right]$$

$$= \sum_{s,a,h}\sum_{t=1}^{T_h(s,a)} y(s,a,h,t)\sum_{k,m,j}\left(\mathbb{P}_{s,a,h} - \mathbb{1}_{s_{h+1}^{k,m,j}}\right)\left(V_{h+1}^k - V_{h+1}^\star\right) \times$$

$$\mathbb{I}\left[(s_h^{k,m,j},a_h^{k,m,j}) = (s,a), t_h^k(s,a) = t\right]$$

$$= C_1 + C_2 + C_3,$$

where

$$C_1 = \sum_{s,a,h}\sum_{t=1}^{T_h-2} y(s,a,h,t)V(s,a,h,t),$$

$$C_2 = \sum_{s,a,h} y(s,a,h,T_h-1)V(s,a,h,T_h-1),$$

$$C_3 = \sum_{s,a,h} y(s,a,h,T_h)V(s,a,h,T_h).$$

Here, $V(s,a,h,t) = \sum_{k,m,j}(\mathbb{P}_{s,a,h} - \mathbb{1}_{s_{h+1}^{k,m,j}})(V_{h+1}^k - V_{h+1}^\star)\mathbb{I}[(s_h^{k,m,j},a_h^{k,m,j}) = (s,a), t_h^k(s,a) = t]$, which is defined in the event $\mathcal{E}_{10}$ in Lemma D.5. Then based on the event $\mathcal{E}_{10}$, we have:

$$|V(s,a,h,t)| \leq 2H\sqrt{2y_h^t(s,a)\iota}.$$

Since $|y(s,a,h,t)| \leq 9$, it holds that:

$$C_1 \leq \sum_{s,a,h}\sum_{t=1}^{T_h(s,a)-2}|y(s,a,h,t)|\cdot|V(s,a,h,t)| \leq 18\sum_{s,a,h}\sum_{t=1}^{T_h(s,a)-2}\sqrt{2y_h^t(s,a)\iota}.$$

Because of Equation (16) in Lemma D.1, we have:

$$C_1 \leq 18\sum_{s,a,h}\sum_{t=1}^{T_h(s,a)-2}\sqrt{2\iota}\cdot 3\sqrt{H}\left(\sqrt{Y_h^t(s,a)} - \sqrt{Y_h^{t-1}(s,a)}\right)$$

$$= 54\sqrt{2H\iota}\sum_{s,a,h}\sqrt{Y_h^{T_h}(s,a)}$$

$$\leq 254\sqrt{2H\iota}\sqrt{SAH\sum_{s,a,h}Y_h^{T_h}(s,a)}$$

$$\leq O\left(\sqrt{SAH^2T_1\iota}\right). \tag{82}$$

The second inequality uses Cauchy-Schwarz Inequality. Similarly, it also holds:

$$\begin{aligned}
C_2 &\le \sum_{s,a,h} |y(s,a,t,T_h-1)| \cdot |V(s,a,h,T_h(s,a)-1)| \\
&\le \sum_{s,a,h} 9 \cdot 2H\sqrt{2y_h^{T_h(s,a)-1}(s,a)\iota} \\
&\le 18H\sqrt{2SAH\sum_{s,a,h} y_h^{T_h(s,a)-1}(s,a)\iota} \\
&\le 26H\left(\sqrt{SAH \cdot 9MSAH(H+1)\iota} + \sqrt{4SAT_1\iota}\right) \qquad (83) \\
&\le O\left(SAH^{\frac{5}{2}}(M\iota)^{\frac{1}{2}} + \sqrt{SAH^2T_1\iota}\right). \qquad (84)
\end{aligned}$$

Here, the third inequality uses Cauchy-Schwarz Inequality. Inequality Equation (83) is because of (h) in Lemma D.1. Similarly, since $y(s,a,t,T_h) = -(1+\frac{2}{H})^{h-1}(1+\frac{1}{H})$, we have:

$$\begin{aligned}
C_3 &\le \sum_{s,a,h}(1+\frac{2}{H})^{h-1}(1+\frac{1}{H})|V(s,a,h,T_h)| \\
&\le 18H\sum_{s,a,h}\sqrt{2y_h^{T_h(s,a)}(s,a)\iota} \\
&\le 18H\sqrt{2SAH\sum_{s,a,h} y_h^{T_h(s,a)}(s,a)\iota} \\
&\le O\left(\sqrt{SAH^2T_1\iota} + SAH^{\frac{5}{2}}(M\iota)^{\frac{1}{2}}\right). \qquad (85)
\end{aligned}$$

Here, the last inequality uses Cauchy-Schwarz Inequality. The last inequality is because of (h) in Lemma D.1.

Using the upper bound of $C_1$ Equation (82), $C_2$ Equation (84) and $C_3$ Equation (85), we can bound the first term in Equation (81) with $O(\sqrt{SAH^2T_1\iota} + SAH^{\frac{5}{2}}(M\iota)^{\frac{1}{2}})$. Then combined with the event $\mathcal{E}_9$ in Lemma D.1, it holds that:

$$\sum_{h=1}^{H}\sum_{k=1}^{K}(1+\frac{2}{H})^{h-1}\epsilon_{h+1}^k \le O\left(\sqrt{SAH^2T_1\iota} + SAH^{\frac{5}{2}}(M\iota)^{\frac{1}{2}}\right).$$

$\square$

**Lemma E.6.** *Under the event $\mathcal{E}_{11}$ in Lemma D.5, it holds that:*

$$\sum_{h=1}^{H}\sum_{k=1}^{K}(1+\frac{2}{H})^{h-1}\phi_{h+1}^k \le O(H\sqrt{T_1\iota}).$$

*Proof.* Based on the definition of $\phi_{h+1}^k$ Equation (63) and the event $\mathcal{E}_{11}$ in Lemma D.5, we have:

$$\begin{aligned}
&\sum_{h=1}^{H}\sum_{k=1}^{K}(1+\frac{2}{H})^{h-1}\phi_{h+1}^k \\
&= \sum_{h=1}^{H}\sum_{k,m,j}(1+\frac{2}{H})^{h-1}\mathbb{I}\left[t_h^{k,m,j} > 1\right]\left(\mathbb{P}_{s_h^{k,m,j},a_h^{k,m,j},h} - \mathbb{1}_{s_{h+1}^{k,m,j}}\right)\left(V_{h+1}^\star - V_{h+1}^{\pi^k}\right) \\
&\le 9H\sqrt{2T_1\iota} = O(H\sqrt{T_1\iota}).
\end{aligned}$$

$\square$

**Lemma E.7.** *Under the event $\bigcap_{i=1}^{15} \mathcal{E}_i$ in Lemma D.5, we have:*

$$\sum_{h=1}^{H} \sum_{k,m,j} \mathbb{I}\left[t_h^{k,m,j} > 1\right] \tilde{b}_h^{k,2}(s_h^{k,m,j}, a_h^{k,m,j}) \leq$$

$$O\left(\sqrt{SAH^2 T_1 \iota} + \sqrt{\beta SAH^2 T_1 \iota} + \sqrt{\beta^2 SAH^2 T_1 \iota} + SAH^{\frac{11}{4}} T_1^{\frac{1}{4}} \iota^{\frac{3}{4}} + S^{\frac{3}{2}} AH^3 \sqrt{N_0} \log(T_1) \iota\right).$$

*Proof.*

$$\sum_{h=1}^{H} \sum_{k,m,j} \mathbb{I}\left[t_h^{k,m,j} > 1\right] \tilde{b}_h^{k,2}(s_h^{k,m,j}, a_h^{k,m,j})$$

$$= 2 \sum_{h=1}^{H} \sum_{k,m,j} \mathbb{I}\left[t_h^{k,m,j} > 1\right] \sqrt{\frac{\tilde{v}_h^{\text{ref},k}(s_h^{k,m,j}, a_h^{k,m,j}) \iota}{N_h^k(s_h^{k,m,j}, a_h^{k,m,j})}} + 2 \sum_{h=1}^{H} \sum_{k,m,j} \mathbb{I}\left[t_h^{k,m,j} > 1\right] \times$$

$$\sqrt{\frac{\tilde{v}_h^{\text{adv},k}(s_h^{k,m,j}, a_h^{k,m,j}) \iota}{n_h^k(s_h^{k,m,j}, a_h^{k,m,j})}} + 10H \sum_{h=1}^{H} \sum_{k,m,j} \mathbb{I}\left[t_h^{k,m,j} > 1\right] \left(\left(\frac{\iota}{N_h^k}\right)^{\frac{3}{4}} + \left(\frac{\iota}{n_h^k}\right)^{\frac{3}{4}} + \frac{\iota}{N_h^k} + \frac{\iota}{n_h^k}\right). \tag{86}$$

Next, we will bound the first term in Equation (86). Based on Equation (40), we have:

$$\tilde{v}_h^{\text{ref},k}(s, a) = \frac{\sum_{i=1}^{N_h^k} \mathbb{V}_{s,a,h}(V_{h+1}^{\text{ref},k_{L_i}}) - (\chi_3 + \chi_4 + \chi_5)}{N_h^k(s, a)}. \tag{87}$$

According to the upper bound given in Equation (41) and Equation (42), we have:

$$\frac{1}{N_h^k(s, a)} |\chi_3| \leq 2H^2 \sqrt{\frac{2\iota}{N_h^k(s, a)}}, \quad \frac{1}{N_h^k(s, a)} |\chi_4| \leq 4H^2 \sqrt{\frac{2\iota}{N_h^k(s, a)}}. \tag{88}$$

Since $V_h^{\text{ref},k}(s) \geq V_h^{\text{REF}}(s)$, we have $\mathbb{P}_{s,a,h} V_{h+1}^{\text{ref},k} \geq \mathbb{P}_{s,a,h} V_{h+1}^{\text{REF}}$. Then according to Equation (72), using the definition of $\chi_5$ Equation (38), it holds that:

$$\begin{aligned}
|\chi_5| &= \sum_{i=1}^{N_h^k} \left(\mathbb{P}_{s,a,h} V_{h+1}^{\text{ref},k_{L_i}}\right)^2 - \frac{\left(\sum_{i=1}^{N_h^k} \mathbb{P}_{s,a,h} V_{h+1}^{\text{ref},k_{L_i}}\right)^2}{N_h^k(s, a)} \\
&\leq \sum_{i=1}^{N_h^k} \left(\mathbb{P}_{s,a,h} V_{h+1}^{\text{ref},k_{L_i}}\right)^2 - N_h^k(s, a) \left(\mathbb{P}_{s,a,h} V_{h+1}^{\text{REF}}\right)^2 \\
&= \sum_{i=1}^{N_h^k} \left[\left(\mathbb{P}_{s,a,h} V_{h+1}^{\text{ref},k_{L_i}}\right)^2 - \left(\mathbb{P}_{s,a,h} V_{h+1}^{\text{REF}}\right)^2\right] \\
&\leq 2H \sum_{i=1}^{N_h^k} \left(\mathbb{P}_{s,a,h} V_{h+1}^{\text{ref},k_{L_i}} - \mathbb{P}_{s,a,h} V_{h+1}^{\text{REF}}\right) \\
&\leq 2H^2 \sum_{i=1}^{N_h^k} \mathbb{P}_{s,a,h} \lambda_{h+1}^{k_{L_i}}. \tag{89}
\end{aligned}$$

The last inequality is because of Equation (72). According to Equation (76) and Equation (78), we have:

$$\sum_{i=1}^{N_h^k} \lambda_{h+1}^{k_{L_i}}(s_{h+1}^{(k,m,j)_{L_i}}) \leq \sum_{k,m,j} \lambda_{h+1}^k(s_{h+1}^{k,m,j}) = \sum_{s \in \mathcal{S}} \sum_{k,m,j} \mathbb{I}\left[N_h^k(s) < N_0, s_{h+1}^{k,m,j} = s\right] \leq 5SN_0. \tag{90}$$

Because of the event $\mathcal{E}_{12}$ in Lemma D.5 and Equation (90), back to Equation (89), we have:

$$\frac{1}{N_h^k(s,a)}|\chi_5| \leq 2H^2\sqrt{\frac{2\iota}{N_h^k(s,a)}} + \frac{10SH^2N_0}{N_h^k(s,a)}. \tag{91}$$

Applying inequalities Equation (88) and Equation (91) to Equation (87), we have:

$$\tilde{v}_h^{\text{ref},k}(s,a) \leq \frac{\sum_{i=1}^{N_h^k}\mathbb{V}_{s,a,h}(V_{h+1}^{\text{ref},k_{L_i}})}{N_h^k(s,a)} + \frac{10SH^2N_0}{N_h^k(s,a)} + 12H^2\sqrt{\frac{\iota}{N_h^k(s,a)}}. \tag{92}$$

For any $s \in \mathcal{S}$, $V_h^{\text{ref},k}(s) \geq V_h^\star(s)$, we have $(\mathbb{P}_{s,a,h}V_{h+1}^{\text{ref},k})^2 \geq (\mathbb{P}_{s,a,h}V_{h+1}^\star)^2$. Then:

$$\frac{\sum_{i=1}^{N_h^k}\mathbb{V}_{s,a,h}(V_{h+1}^{\text{ref},k_{L_i}})}{N_h^k(s,a)} - \mathbb{V}_{s,a,h}(V_{h+1}^\star) \leq \frac{1}{N_h^k(s,a)}\sum_{i=1}^{N_h^k}\left(\mathbb{P}_{s,a,h}(V_{h+1}^{\text{ref},k_{L_i}})^2 - \mathbb{P}_{s,a,h}(V_{h+1}^\star)^2\right)$$

$$\leq \frac{2H}{N_h^k(s,a)}\sum_{i=1}^{N_h^k}\left(\mathbb{P}_{s,a,h}(V_{h+1}^{\text{ref},k_{L_i}}) - \mathbb{P}_{s,a,h}(V_{h+1}^\star)\right). \tag{93}$$

Because for any $s \in \mathcal{S}$, $V_h^{\text{ref},k}(s) \geq V_h^{\text{REF}}(s) \geq V_h^\star(s)$, we have:

$$0 \leq V_{h+1}^{\text{ref},k}(s) - V_{h+1}^\star(s) = (V_{h+1}^{\text{ref},k}(s) - V_{h+1}^\star(s))\lambda_{h+1}^k(s) + (V_{h+1}^{\text{ref},k}(s) - V_{h+1}^\star(s))(1 - \lambda_{h+1}^k(s)).$$

If $\lambda_{h+1}^k(s) = 0$, the reference function is updated and we have $V_{h+1}^{\text{ref},k}(s) - V_{h+1}^\star(s) \leq \beta$; if $\lambda_{h+1}^k(s) = 1$, then we have $V_{h+1}^{\text{ref},k}(s) - V_{h+1}^\star(s) \leq H = H\lambda_{h+1}^k(s)$. Therefore, we have:

$$0 \leq V_{h+1}^{\text{ref},k}(s) - V_{h+1}^\star(s) \leq H\lambda_{h+1}^k(s) + \beta. \tag{94}$$

Combined with the inequality Equation (90), we have:

$$\sum_{i=1}^{N_h^k}\left(V_{h+1}^{\text{ref},k_{L_i}}(s_{h+1}^{(k,m,j)_{L_i}}) - V_{h+1}^\star(s_{h+1}^{(k,m,j)_{L_i}})\right) \leq \sum_{i=1}^{N_h^k}\left(H\lambda_{h+1}^k(s_{h+1}^{(k,m,j)_{L_i}}) + \beta\right)$$

$$\leq 5SHN_0 + \beta N_h^k(s,a).$$

Based on the event $\mathcal{E}_{15}$ in Lemma D.5, applying the inequality to Equation (93), we have:

$$\frac{\sum_{i=1}^{N_h^k}\mathbb{V}_{s,a,h}(V_{h+1}^{\text{ref},k_{L_i}})}{N_h^k(s,a)} - \mathbb{V}_{s,a,h}(V_{h+1}^\star) \leq \frac{10SH^2N_0}{N_h^k(s,a)} + 2H\beta + 4H^2\sqrt{\frac{2\iota}{N_h^k(s,a)}},$$

and then back to Equation (92), it holds:

$$\tilde{v}_h^{\text{ref},k}(s,a) \leq \mathbb{V}_{s,a,h}(V_{h+1}^\star) + \frac{20SH^2N_0}{N_h^k(s,a)} + 18H^2\sqrt{\frac{\iota}{N_h^k(s,a)}} + 2H\beta.$$

Therefore according to Lemma D.2, we have:

$$\sqrt{\frac{\tilde{v}_h^{\text{ref},k}(s_h^{k,m,j}, a_h^{k,m,j})\iota}{N_h^k(s_h^{k,m,j}, a_h^{k,m,j})}}\mathbb{I}\left[t_h^{k,m,j} > 1\right]$$

$$\leq \sum_{h=1}^{H}\sum_{k,m,j}\left(\sqrt{\frac{\mathbb{V}_{s_h^{k,m,j},a_h^{k,m,j},h}(V_{h+1}^\star)\iota}{N_h^k}} + \frac{\sqrt{20SH^2N_0\iota}}{N_h^k} + \frac{\sqrt{18H^2\iota^{\frac{3}{2}}}}{(N_h^k)^{\frac{3}{4}}} + \sqrt{\frac{2H\beta\iota}{N_h^k}}\right) \times$$

$$\mathbb{I}\left[t_h^{k,m,j} > 1\right]$$

$$\leq \sum_{s,a,h}\sqrt{\mathbb{V}_{s,a,h}(V_{h+1}^\star)Y_h^{T_h}(s,a)\iota} + 20\sqrt{SH^2N_0\iota}\cdot SAH\log(T_1) + 2^{\frac{7}{2}}\sqrt{18H^2\iota^{\frac{3}{2}}}\cdot SAHT_1^{\frac{1}{4}}$$

$$+ 4\sqrt{2H\beta\iota}\cdot\sqrt{SAHT_1}$$

$$\leq \sqrt{SAH\sum_{s,a,h}\mathbb{V}_{s,a,h}(V_{h+1}^\star)Y_h^{T_h}(s,a)\iota} + 20S^{\frac{3}{2}}AH^2\sqrt{N_0\iota}\log(T_1) + 64SAH^2T_1^{\frac{1}{4}}\iota^{\frac{3}{2}}$$

$$+ 6\sqrt{\beta SAH^2T_1\iota}. \tag{95}$$

In the last inequality, we use Cauchy-Schwarz Inequality.

Next we will bound $\sum_{s,a,h} \mathbb{V}_{s,a,h}(V_{h+1}^\star) Y_h^{T_h}(s,a)$. Because $V_{H+1}^\star(s) = 0$, removing the term $\sum_{k,m,j} V_1^\star(s_1^{k,m,j})^2$, we have the following inequality:

$$\sum_{h=1}^{H} \sum_{k,m,j} \left( \mathbb{P}_{s_h^{k,m,j}, a_h^{k,m,j}, h}(V_{h+1}^\star)^2 - V_h^\star(s_h^{k,m,j})^2 \right)$$

$$\leq \sum_{h=1}^{H} \sum_{k,m,j} \left( \mathbb{P}_{s_h^{k,m,j}, a_h^{k,m,j}, h}(V_{h+1}^\star)^2 - V_{h+1}^\star(s_{h+1}^{k,m,j})^2 \right).$$

Because of the event $\mathcal{E}_{14}$ in Lemma D.5, then we have:

$$\sum_{h=1}^{H} \sum_{k,m,j} \left( \mathbb{P}_{s_h^{k,m,j}, a_h^{k,m,j}, h}(V_{h+1}^\star)^2 - V_h^\star(s_h^{k,m,j})^2 \right) \leq H^2 \sqrt{2T_1 \iota}. \tag{96}$$

According to Equation (1), for any $a \in \mathcal{A}$, we have:

$$V_h^\star(s) \geq Q_h^\star(s,a) = r_h(s,a) + \mathbb{P}_{s,a,h} V_{h+1}^\star \geq \mathbb{P}_{s,a,h} V_{h+1}^\star.$$

Therefore, we have:

$$\sum_{h=1}^{H} \sum_{k,m,j} \left( V_h^\star(s_h^{k,m,j})^2 - \left( \mathbb{P}_{s_h^{k,m,j}, a_h^{k,m,j}, h}(V_{h+1}^\star) \right)^2 \right)$$

$$= \sum_{h=1}^{H} \sum_{k,m,j} \left( V_h^\star(s_h^{k,m,j}) + \mathbb{P}_{s_h^{k,m,j}, a_h^{k,m,j}, h}(V_{h+1}^\star) \right) \left( V_h^\star(s_h^{k,m,j}) - \mathbb{P}_{s_h^{k,m,j}, a_h^{k,m,j}, h}(V_{h+1}^\star) \right)$$

$$\leq 2H \sum_{h=1}^{H} \sum_{k,m,j} \left( V_h^\star(s_h^{k,m,j}) - \mathbb{P}_{s_h^{k,m,j}, a_h^{k,m,j}, h}(V_{h+1}^\star) \right)$$

$$= 2H \sum_{k,m,j} V_1^\star(s_1^{k,m,j}) + 2H \sum_{h=1}^{H} \sum_{k,m,j} \left( V_{h+1}^\star(s_h^{k,m,j}) - \mathbb{P}_{s_h^{k,m,j}, a_h^{k,m,j}, h}(V_{h+1}^\star) \right)$$

$$\leq 2HT_1 + 2H^2 \sqrt{2T_1 \iota}. \tag{97}$$

Here, the first inequality is because $V_h^\star(s_h^{k,m,j})$, $\mathbb{P}_{s_h^{k,m,j}, a_h^{k,m,j}, h}(V_{h+1}^\star) \leq H$. The last step is because of the event $\mathcal{E}_{13}$ in Lemma D.5. Summing Equation (96) and Equation (97) up, we have:

$$\sum_{s,a,h} \mathbb{V}_{s,a,h}(V_{h+1}^\star) Y_h^{T_h}(s,a) = \sum_{h=1}^{H} \sum_{k,m,j} \mathbb{V}_{s_h^{k,m,j}, a_h^{k,m,j}, h}(V_{h+1}^\star)$$

$$= \sum_{h=1}^{H} \sum_{k,m,j} \left( \mathbb{P}_{s_h^{k,m,j}, a_h^{k,m,j}, h}(V_{h+1}^\star)^2 - \left( \mathbb{P}_{s_h^{k,m,j}, a_h^{k,m,j}, h}(V_{h+1}^\star) \right)^2 \right)$$

$$\leq 2HT_1 + 3H^2 \sqrt{2T_1 \iota}.$$

Applying the inequality to Equation (95), we have:

$$\sqrt{\frac{\tilde{v}_h^{\text{ref},k}(s_h^{k,m,j}, a_h^{k,m,j}) \iota}{N_h^k(s_h^{k,m,j}, a_h^{k,m,j})}} \mathbb{I}\left[ t_h^{k,m,j} > 1 \right]$$

$$\leq \sqrt{3SAH^3 \iota \sqrt{2T_1 \iota}} + \sqrt{2SAH^2 T_1 \iota} + 20S^{\frac{3}{2}} AH^2 \sqrt{N_0 \iota} \log(T_1) + 64SAH^2 T_1^{\frac{1}{4}} \iota^{\frac{3}{2}}$$

$$+ 6\sqrt{\beta SAH^2 T_1 \iota}$$

$$= O\left( \sqrt{SAH^2 T_1 \iota} + \sqrt{\beta SAH^2 T_1 \iota} + SAH^2 T_1^{\frac{1}{4}} \iota^{\frac{3}{2}} + S^{\frac{3}{2}} AH^2 \sqrt{N_0 \iota} \log(T_1) \right). \tag{98}$$

Now we successfully bound the first term in Equation (86). For the second term, according to the definition of $\tilde{v}_h^{\text{adv},k}(s,a)$, we have:

$$n_h^k(s_h^{k,m,j}, a_h^{k,m,j})\tilde{v}_h^{\text{adv},k}(s_h^{k,m,j}, a_h^{k,m,j}) \leq \sum_{i=1}^{n_h^k}\left(V_{h+1}^{\text{ref},k_{l_i}}(s_{h+1}^{(k,m,j)_{l_i}}) - V_{h+1}^{k_{l_i}}(s_{h+1}^{(k,m,j)_{l_i}})\right)^2$$

$$\leq \sum_{i=1}^{n_h^k}\left(V_{h+1}^{\text{ref},k_{l_i}}(s_{h+1}^{(k,m,j)_{l_i}}) - V_{h+1}^{\star}(s_{h+1}^{(k,m,j)_{l_i}})\right)^2.$$

The last inequality is because for any $s \in \mathcal{S}$, $V_h^{\text{ref},k}(s) \geq V_h^k(s) \geq V_h^{\star}(s)$. Using Equation (94) and Cauchy-Schwarz inequality, we have:

$$n_h^k\tilde{v}_h^{\text{adv},k} \leq \sum_{i=1}^{n_h^k}\left(H\lambda_{h+1}^{k_{l_i}}(s_{h+1}^{(k,m,j)_{l_i}}) + \beta\right)^2 \leq 2\sum_{i=1}^{n_h^k}\left(H^2\lambda_{h+1}^{k_{l_i}}(s_{h+1}^{(k,m,j)_{l_i}}) + \beta^2\right). \quad (99)$$

Similar to Equation (90), we have:

$$\sum_{i=1}^{n_h^k}\lambda_{h+1}^{k_{l_i}}(s_{h+1}^{(k,m,j)_{l_i}}) \leq \sum_{k,m,j}\lambda_{h+1}^k(s_{h+1}^{k,m,j}) \leq 5SN_0.$$

Back to Equation (99), we have:

$$\tilde{v}_h^{\text{adv},k}(s_h^{k,m,j}, a_h^{k,m,j}) \leq \frac{10SH^2N_0}{n_h^k(s_h^{k,m,j}, a_h^{k,m,j})} + 2\beta^2.$$

Then using Lemma D.2, we have:

$$2\sum_{h=1}^{H}\sum_{k,m,j}\sqrt{\frac{\tilde{v}_h^{\text{adv},k}(s_h^{k,m,j}, a_h^{k,m,j})\iota}{n_h^k(s_h^{k,m,j}, a_h^{k,m,j})}}\mathbb{I}\left[t_h^{k,m,j} > 1\right]$$

$$\leq 2\sum_{h=1}^{H}\sum_{k,m,j}\mathbb{I}\left[t_h^{k,m,j} > 1\right]\left(\frac{\sqrt{10SH^2N_0\iota}}{n_h^k(s_h^{k,m,j}, a_h^{k,m,j})} + \sqrt{\frac{2\beta^2\iota}{n_h^k(s_h^{k,m,j}, a_h^{k,m,j})}}\right)$$

$$\leq 4\sqrt{2SH^2N_0\iota}\cdot 10SAH^2\log(T_1) + 2\sqrt{2\beta^2\iota}\cdot 4\sqrt{2H}\sum_{s,a,h}\sqrt{Y_h^{T_h}(s,a)}$$

$$\leq 40\sqrt{2}S^{\frac{3}{2}}AH^3\sqrt{N_0\iota}\log(T_1) + 16\sqrt{\beta^2H\iota}\cdot\sqrt{SAHT_1}$$

$$= O\left(\sqrt{\beta^2SAH^2T_1\iota} + S^{\frac{3}{2}}AH^3\sqrt{N_0\iota}\log(T_1)\right). \quad (100)$$

For the third term in Equation (87), according to Lemma D.2, we have:

$$\sum_{h=1}^{H}\sum_{k,m,j}\mathbb{I}\left[t_h^{k,m,j} > 1\right]\frac{\iota^{\frac{3}{4}}}{N_h^k(s_h^{k,m,j}, a_h^{k,m,j})^{\frac{3}{4}}} \leq 4^{\frac{7}{4}}\iota^{\frac{3}{4}}\sum_{s,a,h}Y_h^{T_h}(s_h^{k,m,j}, a_h^{k,m,j})^{\frac{1}{4}}$$

$$\leq 16SAHT_1^{\frac{1}{4}}\iota^{\frac{3}{4}},$$

$$\sum_{h=1}^{H}\sum_{k,m,j}\mathbb{I}\left[t_h^{k,m,j} > 1\right]\frac{\iota^{\frac{3}{4}}}{n_h^k(s_h^{k,m,j}, a_h^{k,m,j})^{\frac{3}{4}}} \leq 2^{\frac{17}{4}}H^{\frac{3}{4}}\iota^{\frac{3}{4}}\sum_{s,a,h}Y_h^{T_h}(s_h^{k,m,j}, a_h^{k,m,j})^{\frac{1}{4}}$$

$$\leq 32SAH^{\frac{7}{4}}T_1^{\frac{1}{4}}\iota^{\frac{3}{4}},$$

$$\sum_{h=1}^{H}\sum_{k,m,j}\mathbb{I}\left[t_h^{k,m,j} > 1\right]\frac{\iota}{N_h^k(s_h^{k,m,j}, a_h^{k,m,j})} \leq 4\iota\sum_{s,a,h}\log(Y_h^{T_h}(s_h^{k,m,j}, a_h^{k,m,j}))$$

$$\leq 4SAH\log(T_1)\iota,$$

and

$$\sum_{h=1}^{H} \sum_{k,m,j} \mathbb{I}\left[t_h^{k,m,j} > 1\right] \frac{\iota}{n_h^k(s_h^{k,m,j}, a_h^{k,m,j})} \leq 8H\iota \sum_{s,a,h} \log(Y_h^{T_h}(s_h^{k,m,j}, a_h^{k,m,j}))$$

$$\leq 8SAH^2 \log(T_1)\iota.$$

Summing the four inequalities, we can bound the third term in Equation (87) with:

$$O\left(SAH^{\frac{11}{4}} T_1^{\frac{1}{4}} \iota^{\frac{3}{4}} + SAH^3 \log(T_1)\iota\right). \tag{101}$$

Applying the upper bound Equation (98), Equation (100) and Equation (101) to Equation (86), we have:

$$\sum_{h=1}^{H} \sum_{k,m,j} \mathbb{I}\left[t_h^{k,m,j} > 1\right] \tilde{b}_h^{k,2}(s_h^{k,m,j}, a_h^{k,m,j}) \leq$$

$$O\left(\sqrt{SAH^2 T_1 \iota} + \sqrt{\beta SAH^2 T_1 \iota} + \sqrt{\beta^2 SAH^2 T_1 \iota} + SAH^{\frac{11}{4}} T_1^{\frac{1}{4}} \iota^{\frac{3}{4}} + S^{\frac{3}{2}} AH^3 \sqrt{N_0} \log(T_1)\iota\right).$$

$\square$

## F    PROOF OF THEOREM 4.2

*Proof.* Because of (e) in Lemma D.1, we have:

$$\hat{T} \geq \sum_{s,a,h} \sum_{t=1}^{T_h-1} y_h^t(s,a) \geq \sum_{s,a,h} y_h^1(s,a) \sum_{t=1}^{T_h-1} (1+\frac{1}{H})^{t-1} \geq \sum_{s,a,h} MH^2 \left[(1+\frac{1}{H})^{T_h-1} - 1\right].$$

The last inequality is because $y_h^1(s,a) \geq MH$ according to (c) in Lemma D.1. Using Jensen's inequality, we have:

$$\sum_{s,a,h} (1+\frac{1}{H})^{T_h-1} \geq SAH(1+\frac{1}{H})^{\frac{\sum_{s,a,h}(T_h-1)}{SAH}}.$$

Therefore, it holds:

$$\hat{T} \geq MSAH^3 (1+\frac{1}{H})^{\frac{\sum_{s,a,h}(T_h-1)}{SAH}} - MSAH^3.$$

This indicates that

$$\sum_{s,a,h} T_h(s,a) \leq SAH + SAH \frac{\log(\frac{\hat{T}}{MSAH^3} + 1)}{\log(1+\frac{1}{H})}. \tag{102}$$

Because $u_{syn} = \text{TRUE}$, in each round, there exists at least one triple $(s,a,h)$ such that the triggering condition is met on it, the total number of rounds is at most the total times of triggering conditions met for $\forall(s,a,h) \in (\mathcal{S}, \mathcal{A}, H)$. Next, we will discuss the times of triggering conditions met for each triple $(s,a,h)$. If the triggering condition for $(s,a,h)$ is met at round $k$, the increase of visits to $(s,a,h)$ is between $c_h^k(s,a)$ and $Mc_h^k(s,a)$. We will discuss how many times the triggering condition can be met at most in one stage for each $(s,a,h) \in (\mathcal{S}, \mathcal{A}, H)$.

1. In the first stage of $(s,a,h)$, $c_h^k(s,a) = 1$. Then FedQ-Advantage will meet at most $MH$ times the triggering condition for $(s,a,h)$.

2. In the stage $t$ $(2 \leq t \leq T_h(s,a))$ of $(s,a,h)$, when $\tilde{n}_h^k(s,a) \leq (1-\frac{1}{H})y_h^{t-1}(s,a)$ for round $k$, we have $c_h^k(s,a) = \lceil \frac{y_h^{t-1}(s,a) - \tilde{n}_h^k(s,a)}{M} \rceil$.

   Assume in this case, it meets $p$ times the corresponding triggering condition at the round $k_1 < k_2 < ... < k_p$. For any $i \in [p]$, since $k_i \geq k_{i-1} + 1$, and $k_i$ and $k_{i-1}$ are in the same stage, we have $\hat{n}_h^{k_{i-1}+1} \leq \tilde{n}_h^{k_i}$. Especially, we know $\hat{n}_h^{k_p-1+1} \leq \tilde{n}_h^{k_p} \leq (1-\frac{1}{H})y_h^{t-1}(s,a)$. For any $i \in [p]$, since the triggering condition is met at the round $k_i$, the increase of the

visits to $(s, a, h)$ in round $k_i$ is at least $c_h^{k_i}(s, a)$. Therefore, according to Equation (6), we have:

$$\hat{n}_h^{k_i+1} - \tilde{n}_h^{k_i} \geq c_h^{k_i}(s, a) \geq \frac{y_h^{t-1}(s, a) - \tilde{n}_h^{k_i}(s, a)}{M}.$$

Let $S_0 = 0$ and $S_i = \hat{n}_h^{k_i+1}$, then for $i \in [p]$ we have:

$$S_i \geq \frac{M-1}{M} \tilde{n}_h^{k_i} + \frac{y_h^{t-1}(s, a)}{M} \geq \frac{M-1}{M} \hat{n}_h^{k_{i-1}+1} + \frac{y_h^{t-1}(s, a)}{M}$$
$$= \frac{M-1}{M} S_{i-1} + \frac{y_h^{t-1}(s, a)}{M}.$$

From the inequality, with mathematical induction, we can derive that:

$$S_i \geq \left(1 - (\frac{M-1}{M})^i\right) y_h^{t-1}(s, a).$$

According to $S_{p-1} \leq (1 - \frac{1}{H}) y_h^{t-1}(s, a)$, we know $p \leq \frac{\log(H)}{\log(\frac{M}{M-1})} + 1$. The last inequality also holds for $M = 1$.

For $M = 1$, we have $p = 0$.

3. In the stage $t$ $(2 \leq t \leq T_h(s, a))$ of $(s, a, h)$, when $\tilde{n}_h^k(s, a) > (1 - \frac{1}{H}) y_h^{t-1}(s, a)$ for round $k$, we have $c_h^k(s, a) = \lfloor \frac{1}{MH} n_h^k(s, a) \rfloor = \lfloor \frac{1}{MH} y_h^{t-1}(s, a) \rfloor$.

Assume it meets $q$ times the triggering condition for $(s, a, h)$ in this case. For $t \geq 2$, there exists a positive integer $r$ such that $rMH \leq y_h^{t-1}(s, a) < (r+1)MH$ and then $c_h^k(s, a) = r$. When the triggering condition of $(s, a, h)$ is met for one time, the increase in the visits is at least $r$. After it is met for $q - 1$ times, we have $r(q-1) \leq \frac{2}{H} y_h^{t-1}(s, a) < 2(r+1)M$. Here, the first inequality is because $y_h^t / y_h^{t-1} \leq 1 + 2/H$, and the second one is because $y_h^{t-1}(s, a) < (r+1)MH$. Therefore, we know $q \leq 4M + 1$.

Combining the three cases, the total times of triggering conditions met for given triple $(s, a, h)$ is at most:

$$MH + \left(\frac{\log(H)}{\log(\frac{M}{M-1})} + 4M + 2\right)(T_h(s, a) - 1).$$

Therefore, combined with the inequality Equation (102), we have:

$$K \leq \sum_{s,a,h} \left(MH + \left(\frac{\log(H)}{\log(\frac{M}{M-1})} + 4M + 2\right)(T_h(s, a) - 1)\right)$$
$$\leq MSAH^2 + SAH \left(\frac{\log(H)}{\log(\frac{M}{M-1})} + 4M + 2\right) \frac{\log(\frac{\hat{T}}{MSAH^3} + 1)}{\log(1 + \frac{1}{H})}$$
$$= MSAH^2 + SAH \left(\frac{\log(H)}{\log(\frac{M}{M-1})} + 4M + 2\right) \frac{\log(\frac{T}{SAH^3} + 1)}{\log(1 + \frac{1}{H})}$$
$$\leq MSAH^2 + 4MSAH^2(\log(H) + 3) \log\left(\frac{T}{SAH^3} + 1\right).$$

The last equality is because $\hat{T} = MT$ and $\log(1 + x) \geq x/2$ for any $0 < x \leq 1$. The last inequality also holds for $M = 1$.

For $u_{syn} = $ FALSE, in each round, all $M$ agents meet the trigger condition, then the round number is at least:

$$K \leq SAH^2 + 4SAH^2(\log(H) + 3) \log\left(\frac{T}{SAH^3} + 1\right).$$

$\square$

