# OpenReview forum: "Federated $Q$-Learning with Reference-Advantage Decomposition: Almost Optimal Regret and Logarithmic Communication Cost"
_ICLR.cc/2025/Conference — ICLR 2025 Poster_

### Official Review · Reviewer_7yMZ · 2024-11-02

**Soundness:** 3
**Presentation:** 2
**Contribution:** 3
**Rating:** 6
**Confidence:** 3

**Summary:**

The paper presents FedQ-Advantage, a novel model-free federated reinforcement learning (FRL) algorithm designed for tabular episodic Markov decision processes (MDPs). This algorithm addresses the challenges of achieving optimal learning performance with minimal communication costs in a distributed setting where multiple agents collaborate under the coordination of a central server.
The proposed FedQ-Advantage enhances existing federated Q-learning methods by introducing mechanisms for variance reduction through reference-advantage decomposition and active policy updates coordinated by the central server.
The algorithm employs event-triggered communication, aligning communication rounds with specific conditions while allowing unaligned updates across state-action-step tuples. This approach reduces communication overhead significantly.
The introduction of heterogeneous triggering conditions allows agents to optimize exploration early in the learning process, further enhancing efficiency by limiting unnecessary communication later on.
An optional forced synchronization mechanism improves robustness and reduces waiting times for agents, streamlining the overall learning process.
The paper demonstrates that FedQ-Advantage achieves an almost optimal regret that approaches the information theoretical bound, scaling logarithmically with the number of communication rounds, and provides near-linear regret speedup compared to single-agent algorithms.
The authors propose heterogeneous triggering conditions that decrease the synchronization frequency, ultimately improving communication costs.
Overall, the paper makes significant strides in federated reinforcement learning by successfully developing an algorithm that balances optimal learning performance with low communication costs. Statistical theory demonstrates that FedQ-Advantage is the first model-free federated RL algorithm to achieve both almost optimal regret and logarithmic communication efficiency. These contributions position this paper as a valuable contribution to the field.

**Strengths:**

This paper is theoretically solid and well structured.

Novelty: This paper develops the first near-optimal model-free RL algorithm under the setting of a tabular MDP and a communication graph with a central server.

Significance: The communication cost matches with the best policy switching cost in Q-learning.

Quality: The paper has good quality. The theoretical results look solid. It is good to see the synthetic experimental results match the theory. .

Overall, I think this is a good theoretical paper.

**Weaknesses:**

There is no major flaw in this paper. However, the following can be addressed.

1. The paper is not easy to read and can be improved. For examples:

a) At the end of page 5, the paper mentioned that the visitation number $y_{t+1}$ is similar to the exponential rate in related works due to the first case and ‘Equation 4’. However, equation 4 is in page 7 and thus it is very hard to really understand what it means when reading the sentence Page 5. To improve, I think the authors could (i) either not to add such kind of details in section 3.1. Just presenting some intuition would be enough, or (ii) provide a brief preview of the key idea from Equation 4 when it's first mentioned on page 5. These might help readers follow the logic more easily without having to jump ahead in the paper.

b) Page 7, Item 4 can be improved. It is very hard to read. I understand that it is inevitable for this paper to be notationally heavy. But in the main text one might still want to avoid in-line equations of such lengths. If indeed one wish to insert these in the main text, then I think an explanation after each chunk of equations is necessary. The explanation can be as short as one sentence like “the central server finds the tuples whose visitation number exceeds… ”.

2. The claimed technical novelty of “non-martingale concentration” is a bit misleading. If my understanding is correct, the paper is simply approximating a non-martingale sequence with a martingale. In order to achieve this, the algorithm is specifically designed with a heterogeneous trigger condition that forces communication with the server when the number of visitations exceeds a threshold. This will ensure the ‘distance’ between the non-martingale sequence and the implicit martingale sequence to be small enough. After that, the entire error bound is a standard martingale convergence argument. If my understanding is not wrong, then this seems to me a quite straightforward approach to handle concentration under an asynchronous communication setting. I wouldn't expect the paper to give a real non-martingale concentration inequality. After all, if the communication is truly arbitrary and random, then I imagine it is impossible to get any valid concentration result.
To resolve this issue, I would suggest not to call this a ‘non-martingale’ concentration to avoid misunderstanding from people doing literature search for real non-martingale concentration.

Overall, I do not detect major technical issues in this theoretical paper. The above issues should be easily fixable.

**Questions:**

Please see the **Weaknesses** section.

---

> ### Author Response · Authors · 2024-11-22
> **Response to Reviewer 7yMZ**
>
> We thank the reviewer for the careful reading and thoughtful comments. We have addressed the reviewer's questions in detail below and revised the paper accordingly. In particular, we have made significant efforts to improve the presentation, readability, and accessibility of the manuscript. The changes are marked in blue in the revised manuscript. We hope that the responses provided and the updates made to the paper satisfactorily address the reviewer’s concerns.
>
> **Weakness 1:** The presentation can be improved.
>
>
> Thank you for your constructive feedback regarding the presentation of our paper. Following your suggestion, we have made significant revisions to improve the presentation, readability, and accessibility of the manuscript. Below, we summarize the key updates incorporated into our revised manuscript:
>
> (a) **Revised the Structure of Section 3.** In the revised manuscript, following your suggestions, Section 3.1 now introduces the basic structure of our algorithm design (namely, aligned rounds and unaligned stages), Section 3.2 presents the algorithm details, and Section 3.3 provides the intuition behind the algorithm’s design.
>
> (b) **Significantly Reduced Notational Complexity.** In Section 3.2, we have removed the large block summarizing all the mathematical details of notations. Instead, notations are defined and explained contextually when each component of the algorithm is introduced. A comprehensive summary of all notations is now included in Appendix A for reference.
>
>
> (c) **Enhanced Explanations.** We have added detailed explanations for each step of the algorithm in Section 3.2. For instance, following your suggestions to address the comment on the readability of Step 4 in **Weakness 1.(b)**, we have avoided in-line equations, added more descriptions, and explained the reference-advantage decomposition structure after introducing the decomposition-type update (Equation (13)).
>
>
> (d) **Improved Readability of Equations.** To enhance the readability of equations, we have provided a brief preview of the key idea or added an explanation after the equations. Also, we have avoided in-line equations that may make them less readable. For example,  the updates of the estimated $Q$ functions in Step 4 are presented as Equations (11) to (13) instead of within-line equations, and more explanations have been added after presenting Equations (11) to (13).
>
>
> (e) **Enhanced Algorithm Presentation.** The presentation of Algorithm 1 has been enhanced with clearly labeled steps and comments to match each stage of the algorithm design, improving overall clarity.
>
> (f) **New Section 3.3 on Intuition.** We have added a dedicated section to provide intuition behind the key features of our algorithm design, including exponential increasing stages, reference-advantage decompositions, heterogeneous event-triggered communication, and optional forced synchronization. Here, following your suggestions to address the comment to improve the readability of exponentially increasing stage sizes in **Weakness 1.(a)**, we have moved such kind of details to the new Section 3.3 and added more intuition and explanations after showing our communication condition and stage renewal condition (Equations (2) and (7)) in Section 3.2.
>
> We believe these changes address the concerns raised and significantly improve the presentation and accessibility of our manuscript. Thank you for your valuable suggestions.
>
>
>
> **Weakness 2:** Clarification of "non-martingale concentration''
>
> Thanks for this helpful suggestion. We appreciate your understanding of our approach. Our technical novelty lies in bounding the difference between the martingale sum and the non-martingale sum using stage-wise approximation, rather than relying on a direct element-wise approximation.
>
> Following your suggestion, we have updated our revised manuscript to summarize this technical novelty in Section 1 under "Stage-wise approximations in non-martingale analysis" to ensure clarity and avoid potential misunderstandings.

---

> > ### Comment · Reviewer_7yMZ · 2024-11-23
> > **Response**
> >
> > I thank the authors for their detailed rebuttal.
> > I am satisfied with the promised change to be made to the paper for better readability (notations, better explanation of the setting, etc.).
> > I also recommend making necessary change according to other reviewers' suggestions. For example, the following seems particularly important:
> >
> > 1. condition and process of our communication
> > 2. discussion on communication conditions in heterogeneous settings
> > 3. A brief discussion on comparison with model-based approaches
> > 4. additional related works
> >
> > Overall, I think this is a cool paper and I maintain my score at 6 for acceptance.

---

> > > ### Author Response · Authors · 2024-11-24
> > > **Following up on the rebuttal**
> > >
> > > Thank you once again for your insightful comments and valuable advice! We have revised the paper in accordance with suggestions from other reviewers and summarized the changes below:
> > >
> > > **Condition and process of our communication:**
> > >
> > > Our algorithm uses **aligned rounds** and **unaligned stages**. Communication occurs at the end of each round, and policy switching takes place at the end of each stage. Both communication and policy-switching are triggered by events related to visiting numbers. This process is provided in **Step 2 and Step 3 of Section 3.2** in our revised draft. Additionally, we have expanded the discussion in **Section 3.3** to elaborate on the relationship between these communication conditions and their impact on the algorithm's performance.
> > >
> > > **Discussion on communication conditions in heterogeneous settings:**
> > >
> > > The communication conditions, regardless of the choice of forced synchronization, do not impact the algorithm's ability to achieve almost optimal regret or its adaptability to heterogeneous exploration speeds among agents. As shown in **Theorem 4.1** of our paper, our algorithm achieves almost optimal regret and linear speedup, even with variations in the local exploration speeds of agents.
> > >
> > > With respect to communication costs, the communication conditions introduce a trade-off between communication efficiency and waiting time. This trade-off is discussed in detail in **Section 3.3** of our revised manuscript.
> > >
> > > **A brief discussion on comparison with model-based approaches:**
> > >
> > > Prior to our work, only a few studies have addressed the regret and communication cost of on-policy model-based multi-agent RL methods for tabular episodic MDPs with heterogeneous transition kernels. **Table 1** in our manuscript provides a detailed comparison of the regret bounds and communication costs between our proposed FedQ-Advantage and three model-based approaches, including [1] and two multi-batch RL methods.
> > >
> > > To summarize, our approach achieves a superior regret bound and lower communication cost compared to these existing works.
> > >
> > > **Additional related works:**
> > >
> > > In **Appendix B** of the revised manuscript, we have separately listed federated value-based methods and federated policy gradient-based methods, along with additional related work. Furthermore, we have reviewed additional papers that address linear function approximation in the same appendix.
> > >
> > > [1] Yiding Chen, Xuezhou Zhang, Kaiqing Zhang, Mengdi Wang, and Xiaojin Zhu. Byzantine-robust online and offline distributed reinforcement learning. AISTATS, 2023.

---

### Official Review · Reviewer_wG6t · 2024-11-03

**Soundness:** 3
**Presentation:** 2
**Contribution:** 3
**Rating:** 6
**Confidence:** 3

**Summary:**

This paper proposes the first model-free federated tabular RL algorithm that achieves almost optimal regret with logarithmic communication cost. The authors adopt a reference-advantage decomposition when updating the $Q$ function via UCB for variance reduction. In addition, several designs surrounding communication and synchronization ensure a lower communication cost. Numerical experiments also support the benefit of FedQ-Advantage.

**Strengths:**

* The authors effectively highlight the contributions, including technical novelty.
* FedQ-Advantage is the first design for an almost optimal federated model-free RL algorithm that enjoys a logarithmic communication cost in a federated tabular setting.
* Theoretical analysis is provided and it can match the performance of the state-of-the-art results in tabular RL.

**Weaknesses:**

* The related work is not comprehensive. The authors mentioned *most existing model-free federated algorithms do not actively update the exploration policies for local agents and fail to provide low regret*, which is not precise. Please also consider including [1, 2, 3] for full comparison. Although they are more based on functional approximation instead of pure tabular setting, they definitely incorporate active exploration strategies in federated learning.
* The writing can be improved, especially the algorithm design section. Please refer to the questions below.


#### [1] Dubey, Abhimanyu, and Alex Pentland. "Provably efficient cooperative multi-agent reinforcement learning with function approximation." arXiv preprint arXiv:2103.04972 (2021).

#### [2] Min, Yifei, et al. "Cooperative multi-agent reinforcement learning: Asynchronous communication and linear function approximation." ICML, 2023

#### [3] Hsu, Hao-Lun, et al. "Randomized Exploration in Cooperative Multi-Agent Reinforcement Learning." NeurIPS, 2024

**Questions:**

* The setting requires more description. I am not sure if I misunderstood the algorithm design, but it seems that the notation and Figure 1 is not clear to me. Specifically, $k^t_h(s,a)$ is denoted as the index of the first round that belongs to stage t, so I thought "t" refers to the index of stage while the first row in Figure indicates $k_2^1$ belongs to stage 2. In addition, the second row in Figure 1 shows $k_{12}$ and $k_{22}$ also confuses me. Please also explain more about the logic of stage $t$ how it is composed of rounds $k_h^t, k_h^t +1, ..., k_h^{t+1}-1$.

* Can you mainly compare your threshold/ event-triggered communication strategy with [1, 2]? Again, I know that your work is in a tabular setting, but since I am not fully clear with Figure 1. A comparison can make your design more readable.

* I was wondering how your method can be extended to function approximation out of a tabular setting with the curse of dimensionality, especially for those main components in your algorithm, such as separate event-triggered communication and policy switching, heterogeneous communication triggering conditions, optional forced synchronization, and reference-advantage decomposition. Any challenge or limitation for your designs to convert to linear functional approximation setting?

* In Figure 3, it shows that the empirical regret from Concurrent UCB-Advantage is larger than FedQ-Advantage while Concurrent UCB-Advantage has the same regret as FedQ-Advantage from a theoretical standpoint. Could you please elaborate more on where the gap comes from? I am also interested in the rounds of communication as the comparison for Figure 3 (FedQ-Advantage vs. Concurrent UCB-Advantage), which you can ignore UCB-A: $10^6$.

#### [1] Dubey, Abhimanyu, and Alex Pentland. "Provably efficient cooperative multi-agent reinforcement learning with function approximation." arXiv preprint arXiv:2103.04972 (2021).

#### [2] Min, Yifei, et al. "Cooperative multi-agent reinforcement learning: Asynchronous communication and linear function approximation." ICML, 2023

---

> ### Author Response · Authors · 2024-11-22
> **Response to Reviewer wG6t (part one)**
>
> We thank the reviewer for the careful reading and thoughtful comments. We have addressed the reviewer's questions in detail below and revised the paper accordingly. In particular, we have made significant efforts to improve the presentation, readability, and accessibility of the manuscript. The changes are marked in blue in the revised manuscript. We hope that the responses provided and the updates made to the paper satisfactorily address the reviewer’s concerns.
>
> **Weakness 1:** Related work and including [1, 2, 3] for full comparison.
>
> Thanks for pointing out this issue and highlighting the related works [1, 2, 3] that use on-policy methods based on the function approximation. We have refined the statement to be more precise (see Lines 41-42 in Section 1) and included the comparison with [1, 2, 3] in Appendix B of the revised manuscript. Below, we provide a detailed response to your comment.
>
> [1] introduces Coop-LSVI for cooperative multi-agent reinforcement learning (MARL), which achieves a regret upper bound of $\tilde{O} \left(d^{\frac{3}{2}}\sqrt{MH^3T}\right)$ and a communication complexity of $\tilde{O}(dHM^3)$. [2] propose an asynchronous version of LSVI-UCB (originally from [4]), which matches the same regret bound but with improved communication complexity of $\tilde{O}(dHM^2)$ compared to [1]. [3] develops two algorithms that incorporate randomized exploration, achieving the same regret and communication complexity as [2]. [1, 2, 3] assume the structure of linear function approximation and are more general than our tabular MDP assumption.
>
> **Methodological Comparison.** All three works in MARL, [1, 2, 3], employ event-triggered communication conditions based on determinants of certain quantities. Similarly, FedQ-Advantage adopts event-triggered communication conditions; however, due to the tabular setting, we use counts of visits to control communication more directly. During synchronization, in [1, 2, 3], local agents share original rewards or trajectories with the server, whereas, in FedQ-Advantage, local agents only share summary statistics (i.e., counts and estimated value functions), preserving data privacy and reducing communication overhead.
>
> **Performance Comparison.** When comparing performance under tabular settings, FedQ-Advantage demonstrates better regret and communication efficiency due to its tailored design for discrete state-action spaces. On the one hand, in Example 1 of [4], when recovering tabular MDPs using linear function approximation, we have $d = SA$. Therefore, the regret bounds given in [1, 2, 3] are at least $\tilde{O} \left(\sqrt{H^3S^3A^3MT}\right)$, while the regret bound of FedQ-Advantage is $\tilde{O} \left(\sqrt{H^2SAMT}\right)$. On the other hand, the communication complexity of [1, 2, 3] is defined as the total number of communication rounds. When $d = SA$, this translates to at least $\tilde{O}(M^2HSA)$. Additionally, in [1, 2, 3], each communication round involves agents sharing original data (e.g., rewards in [1], trajectories in [2, 3]) with the server, leading to a total communication cost scaling that scales as $\Theta(MT)$. In contrast, FedQ-Advantage achieves a logarithmic communication cost.

---

> ### Author Response · Authors · 2024-11-22
> **Response to Reviewer wG6t (part two)**
>
> **Weakness 2:** Improving the writing, especially the algorithm design section.
>
> Thank you for your constructive feedback regarding the writing of our paper. Following your suggestion, we have made significant revisions to improve the presentation, readability, and accessibility of the manuscript, especially the algorithm design section. Below, we summarize the key updates incorporated into our revised manuscript:
>
> (a) **Revised the Structure of Section 3.** In the revised manuscript, Section 3.1 now introduces the basic structure of our algorithm design (namely, aligned rounds and unaligned stages), Section 3.2 presents the algorithm details, and Section 3.3 provides the intuition behind the algorithm’s design.
>
> (b) **Significantly Reduced Notational Complexity.** In Section 3.2, we have removed the large block summarizing all the mathematical details of notations. Instead, notations are defined and explained contextually when each component of the algorithm is introduced. A comprehensive summary of all notations is now included in Appendix A for reference.
>
>
> (c) **Enhanced Explanations.** We have added detailed explanations for each step of the algorithm in Section 3.2. For instance, in Step 4, we explain the reference-advantage decomposition structure after introducing the decomposition-type update (Equation (13)).
>
>
> (d) **Improved Readability of Equations.** To enhance the readability of equations, we have provided a brief preview of the key idea or added an explanation after the equations. Also, we have avoided in-line equations that may make them less readable. For example,  the updates of the estimated $Q$ functions in Step 4 are presented as Equations (11) to (13) instead of within-line equations, and more explanations have been added after presenting Equations (11) to (13).
>
>
> (e) **Enhanced Algorithm Presentation.** The presentation of Algorithm 1 has been enhanced with clearly labeled steps and comments to match each stage of the algorithm design, improving overall clarity.
>
> (f) **New Section 3.3 on Intuition.** We have added a dedicated section to provide intuition behind the key features of our algorithm design, including exponential increasing stages, reference-advantage decompositions, heterogeneous event-triggered communication, and optional forced synchronization.
>
> We believe these changes address the concerns raised and significantly improve the presentation and accessibility of our manuscript. Thank you for your valuable suggestions.
>
> **Question 1:** The notations in Figure 1 and why a stage contains consecutive episodes.
>
> Thanks for your careful review. The notation $k_2^1$ was a typo in the previous manuscript, and we have revised the notations and added definitions in the updated Figure 1 to enhance clarity. In the updated description of Figure 1, $k_{1t}$ is the last round index for stage $t$ of $(s^1,a^1,h^1)$ and $k_{2t}$ is the last round index for stage $t$ of $(s^2,a^2,h^2)$. These notations ($k_{1t}$ and $k_{2t}$) are only used in Figure 1 to illustrate the relationship between aligned rounds and unaligned stages for two different triples $(s^1,a^1,h^1)$ and $(s^2,a^2,h^2)$.
>
> Additionally, we agree that it is helpful to further explain the logic behind the design of aligned rounds and unaligned stages, which form the basic structure of our proposed FedQ-Advantage. In the revised manuscript, Section 3.1 introduces the basic structure, Section 3.2 presents the algorithm details, and Section 3.3 provides the intuition behind the algorithm’s design. Below, we provide a detailed response to address your comment.
>
> We define stages for each $(s,a,h)$, and $t_h^k(s,a,h)$ represents the stage index for round $k$ at $(s,a,h)$. Section 3.1 also introduces the notation $k_h^t(s,a)$ to denote the first round of stage $t$ at $(s,a,h)$. At the end of each round, Step 3 of Section 3.2 determines whether to renew the stage for each $(s,a,h)$ based on the number of visits in the current stage (see Equation (8)). Since the stage renewal is judged independently for each $(s,a,h)$, the stages for different triples are inherently **unaligned**. This design ensures that consecutive rounds for each $(s,a,h)$ are grouped into stages. This structure enables FedQ-Advantage to effectively balance exploration and communication, leveraging the flexibility provided by unaligned stages to adapt to the dynamics of each $(s,a,h)$.

---

> ### Author Response · Authors · 2024-11-22
> **Response to Reviewer wG6t (part three)**
>
> **Question 2:** Comparison of threshold or event-triggered communication strategy with [1, 2].
>
> Thanks for this helpful comment. Our event-triggered communication strategy shares similarities with those in [1, 2], but there are notable differences in how the trigger conditions are applied.
>
> In [1, 2], if any agent meets the trigger condition at any step $h$, the exploration terminates and communication occurs.
>
> In our work, with forced synchronization, if any agent meets the trigger condition for any state-action-step triple $(s,a,h)$, exploration terminates and communication occurs; without forced synchronization, communication occurs only after every agent meets the trigger condition for at least one state-action-step triple $(s,a,h)$.
>
> These differences reflect our design considerations for the tabular MDP setting, where balancing exploration, communication efficiency, and synchronization across agents is critical.
>
> **Question 3:** Extension to function approximation
>
> Extending our algorithm to non-tabular settings, such as those involving linear function approximation, is an interesting direction for future research. In this context, [5] proposed a federated offline RL algorithm that utilizes confidence-bound methods for non-tabular settings, and [6, 7] applied the reference-advantage decomposition technique in offline RL settings with linear function approximation. These works demonstrate the feasibility of employing confidence bounds and reference-advantage decomposition techniques in non-tabular settings.
>
> However, generalizing our event-trigger condition and policy-switching strategy to non-tabular settings remains a significant challenge. For non-tabular settings with continuous state spaces, it becomes difficult to quantify visit numbers. This necessitates the development of new mechanisms to terminate agent exploration effectively and update agent policies through event-trigger conditions, all while adhering to communication cost constraints.
>
> **Question 4:** The empirical regret gap and communication rounds difference.
>
> Thanks for this insightful comment. We would like to clarify that Concurrent UCB-Advantage is not included in Figure 3 because it is not a federated algorithm. Unlike FedQ-Advantage, which shares only summary statistics with the central server, Concurrent UCB-Advantage requires agents to share all local trajectories, making it unsuitable for federated settings.
>
> Figure 3 illustrates the regret results of FedQ-Advantage with 10 agents and $10^5$ episodes for each agent and single-agent UCB-Advantage with $10^5$ and $10^6$ episodes, demonstrating the speedup effect of FedQ-Advantage. The larger regret observed for UCB-Advantage with $10^5$ episodes arises from the reduced sample size for policy training compared to the other two cases. This highlights the efficiency of FedQ-Advantage in leveraging multiple agents to achieve lower regret under the same total number of episodes.
>
> The comparison of communication rounds between Concurrent UCB-Advantage and FedQ-Advantage is presented in Table 1 (see Lines 108-120 on Page 3). Specifically: Concurrent UCB-Advantage incurs a communication cost of $O(MT)$, as it involves sharing all original trajectories; FedQ-Advantage, by contrast, achieves a logarithmic communication cost, scaling logarithmically with $T$. If plotted, the communication cost of Concurrent UCB-Advantage would appear as a half-line starting from the origin, whereas FedQ-Advantage would grow logarithmically with $T$. FedQ-Advantage becomes much more communication-efficient when $T$ is large.
>
> We hope this explanation clarifies the source of the observed gap and the communication cost comparison between these methods.
>
> [1] Abhimanyu Dubey, and Alex Pentland. Provably efficient cooperative multi-agent reinforcement learning with function approximation. arXiv preprint arXiv:2103.04972 (2021).
>
> [2] Yifei Min, Jiafan He, Tianhao Wang, and Quanquan Gu. Cooperative multi-agent reinforcement learning: Asynchronous communication and linear function approximation. ICML, 2023.
>
> [3] Hao-Lun Hsu, Weixin Wang, Miroslav Pajic, and Pan Xu. Randomized Exploration in Cooperative Multi-Agent Reinforcement Learning. NeurIPS, 2024.
>
> [4] Chi Jin, Zhuoran Yang, Zhaoran Wang, and Michael I. Jordan. Provably efficient reinforcement learning with linear function approximation. Mathematics of Operations Research 48, no. 3 (2023): 1496-1521.
>
> [5] Doudou Zhang, Yufeng Zhang, Aaron Sonabend-W, Zhaoran Wang, Junwei Lu, and Tianxi Cai. Federated offline reinforcement learning. Journal of the American Statistical Association, 2024.
>
> [6] Wei Xiong, Han Zhong, Chengshuai Shi, Cong Shen, Liwei Wang, and Tong Zhang. Nearly Minimax Optimal Offline Reinforcement Learning with Linear Function Approximation: Single-Agent MDP and Markov Game. ICLR, 2023.
>
> [7] Qiwei Di, Heyang Zhao, Jiafan He, and Quanquan Gu. Pessimistic Nonlinear Least-Squares Value Iteration for Offline Reinforcement Learning. ICLR, 2024.

---

> ### Author Response · Authors · 2024-11-25
> **Following up on the rebuttal**
>
> Thanks again for your insightful comments and valuable advice! We have uploaded the revised draft and replied to your suggested weaknesses and questions. If you have further questions or comments, we are happy to reply in the author-reviewer discussion period, which ends on Nov 26th at 11:59 pm, AoE. If our response resolves your concerns, we kindly ask you to consider raising the rating of our work. Thank you very much for your time and efforts!

---

> > ### Comment · Reviewer_wG6t · 2024-11-26
> > **Reviewer response**
> >
> > Thanks to the authors for their comprehensive response and the corresponding manuscript modification. I can see the improvement in writing makes the algorithm design much clearer. I mostly agree with the comparisons of those related work the authors add while [3] proposed a version of federating setting in algorithm 4 sharing the weight of the collected estimated $Q$ functions, which can also reduce communication cost. In addition, I can see the success of the proposed framework in the tabular setting, but as the authors mentioned, generalizing their event-trigger condition and policy-switching strategy to non-tabular settings remains a significant challenge, which is actually the main contribution of this work. Overall, I am satisfied with the authors' response of the questions I have within tabular scope while I still have concerns about its scalability in a non-tabular setting.

---

> > > ### Author Response · Authors · 2024-11-26
> > > **Response to the new comment of Reviewer wG6t**
> > >
> > > Thank you for your thoughtful reply and additional comments on the non-tabular setting.
> > >
> > > **Discussion of Algorithm 4 in [3]**
> > >
> > > [3] proposed a randomized exploration framework in MARL. Algorithm 4 in Appendix I.2 of [3] is a modified version of their main algorithm (Algorithm 1). While Algorithm 4 introduces mean aggregation of local estimated $Q-$functions, synchronization in Algorithm 4 is based on constant round sizes, rather than event-triggered conditions (the determinant synchronization condition, as outlined in Equation (4.3) of [3]) analyzed in Algorithm 1 of [3]. Although the constant local iteration used in Algorithm 4 is a feasible synchronization condition mentioned in Section 4.1, the regret bound and communication round bound for Algorithm 1 are derived under the determinant synchronization condition. Due to these differences, **the regret and communication round analyses for Algorithm 1 in [3] do not apply to their Algorithm 4**, making it unclear whether Algorithm 4 could theoretically ``reduce communication cost". Thus, whether Algorithm 4 achieves a provably low communication cost remains an open question.
> > >
> > > **Discussion of Algorithm 1 in [3]**
> > >
> > > Within the scope of provably efficient methods with low theoretical regret and communication rounds, [3] demonstrated that, when using linear function approximation in their Algorithm 1, event-triggered conditions can be employed for synchronization. If the squared loss is used in each local $Q-$update (e.g., Equation (4.5) in [3]), agents no longer need to share original trajectories. Instead, agents only share matrices of size $d\times d$ and vectors of size $d$ (see Equation (4.8) in [3]) at each step. This reduces the communication cost per round to $O(MHd^2)$, leading to a total communication cost of $\tilde{O}(M^3H^2d^3)$. This is a significant theoretical contribution of [3] over earlier works like [1, 2]. We have added this comparison to Related Works in our revised manuscript.
> > >
> > >
> > > **FedQ-Advantage in the Tabular Setting**
> > >
> > > The tabular setting offers a natural separation of each state-action-step triple, enabling a key tool: **counts of visits** for each triple. This separation is crucial for on-policy FedRL, where communication should occur when local agents have sufficiently explored. FedQ-Advantage defines sufficient exploration **separately** for each triple (see Step 2 in Section 3.2), using visit counts to trigger communication. In contrast, the event-triggered condition in [3] with linear function approximation is defined globally, without leveraging per-triple statistics.
> > >
> > > When recovering tabular MDPs using linear function approximation ($d = SA$), compared to the communication costs of $\tilde{O}(M^3H^2d^3)$ in [3], FedQ-Advantage achieves lower communication costs in terms of $M, S, A$, with $\tilde{O}(M^2H^3S^2A)$ for forced synchronization and $\tilde{O}(MH^3S^2A)$ without forced synchronization. This is because FedQ-Advantage uses visit counts, which enables finer control over exploration and synchronization.
> > >
> > > **Extending FedQ-Advantage to Non-Tabular Settings**
> > >
> > > Extensions of FedQ-Advantage to non-tabular settings are feasible if the MDP has a separable structure. In such cases, event-triggered conditions similar to those in [3] could be defined for groups of triples, allowing synchronization when any condition within a group is satisfied. This approach could improve communication costs by refining the universal condition. However, when such separability is absent, generalizing the count-based conditions for communication and policy-switching becomes significantly more challenging. This limitation underscores the difficulty of extending our approach to non-tabular settings without additional assumptions or novel mechanisms.
> > >
> > >
> > > Last but not least, we hope that the updated comparisons and discussions of the related works have addressed the remaining concerns.
> > >
> > > [1] Abhimanyu Dubey, and Alex Pentland. Provably efficient cooperative multi-agent reinforcement learning with function approximation. arXiv preprint arXiv:2103.04972 (2021).
> > >
> > > [2] Yifei Min, Jiafan He, Tianhao Wang, and Quanquan Gu. Cooperative multi-agent reinforcement learning: Asynchronous communication and linear function approximation. ICML, 2023.
> > >
> > > [3] Hao-Lun Hsu, Weixin Wang, Miroslav Pajic, and Pan Xu. Randomized Exploration in Cooperative Multi-Agent Reinforcement Learning. NeurIPS, 2024.

---

> ### Comment · Reviewer_wG6t · 2024-12-01
> **Reviewer response**
>
> Thanks to the authors for their further discussion of my questions. I think I have a better understanding about the value, limitation, and potential future work of this manuscript. Thus, I have further increased my rating accordingly.

---

### Official Review · Reviewer_RUvj · 2024-11-04

**Soundness:** 3
**Presentation:** 1
**Contribution:** 2
**Rating:** 3
**Confidence:** 4

**Summary:**

This paper studies online federated Q-learning, which seeks to develop an optimal Q-function by utilizing collaborative exploration of multiple agents in a federated manner. The authors introduce a federated Q-learning algorithm that incorporates reference-advantage decomposition to reduce variance, which leads to the improved regret bound. Additionally, they propose event-triggered communication and policy switching, which save communication costs.

**Strengths:**

1. The paper incorporated the variance reduction technique to federated Q-learning and demonstrated improved sample efficiency through regret analysis.
2. They empirically demonstrated the performance of the proposed algorithm in a synthetic environment, comparing it with baseline algorithms.

**Weaknesses:**

1. Considering the existing literature on federated Q-learning [1,2,3], Q-learning with variance reduction (reference-advantage decomposition) [4,5], and Q-learning with unaligned stage design [5], the integration of these existing techniques into federated Q-learning seems incremental.
2. The communication cost provided in this paper (O(SAH^2) or O(MSAH^2)) does not seem particularly low or noteworthy, given that other federated Q-learning algorithms [2,3] have demonstrated that O(H) communication rounds are sufficient.
3. The presentation of the paper seems to need further refinement. Due to excessive notation and the use of vaguely defined terms, the descriptions of the algorithm and the equations are somewhat confusing, making them difficult to follow.
4. In federated settings, privacy is very crucial; however, the algorithm seems to require the agents to send too many estimates to the server alongside value estimates, which could potentially expose too much information about the datasets.

[1] Zhong Zheng, Fengyu Gao, Lingzhou Xue, and Jing Yang. Federated q-learning: Linear regret speedup with low communication cost. In The Twelfth International Conference on Learning Representations, 2024.\
[2] Jiin Woo, Laixi Shi, Gauri Joshi, and Yuejie Chi. Federated offline reinforcement learning: Collaborative single-policy coverage suffices. In International Conference on Machine Learning, 2024.\
[3] Sudeep Salgia and Yuejie Chi. The sample-communication complexity trade-off in federated Q-learning. arXiv preprint arXiv:2408.16981, 2024.\
[4] Gen Li, Laixi Shi, Yuxin Chen, Yuantao Gu, and Yuejie Chi. Breaking the sample complexity barrier to regret-optimal model-free reinforcement learning. Advances in Neural Information Processing Systems, 34:17762–17776, 2021.\
[5] Zihan Zhang, Yuan Zhou, and Xiangyang Ji. Almost optimal model-free reinforcement learning via reference-advantage decomposition. Advances in Neural Information Processing Systems, 33: 15198–15207, 2020.

**Questions:**

See weaknesses.

---

> ### Author Response · Authors · 2024-11-22
> **Response to Reviewer RUvj (part one)**
>
> We thank the reviewer for the careful reading and thoughtful comments. We have addressed the reviewer's questions in detail below and revised the paper accordingly. In particular, we have made significant efforts to improve the presentation, readability, and accessibility of the manuscript. The changes are marked in blue in the revised manuscript. We hope that the responses provided and the updates made to the paper satisfactorily address the reviewer’s concerns.
>
> **Weakness 1:** Contribution and difficulties other than integration.
>
> We respectfully disagree with the claim that "the integration of these existing techniques into federated Q-learning seems incremental." This perspective overlooks the fundamental challenges addressed in our work.
>
> In the RL literature, achieving information-theoretic bounds is a critical benchmark, as demonstrated by [4, 5] for single-agent RL. Establishing almost optimal regret in the **on-policy FedRL setting**, where agents actively update their implemented policies, is inherently challenging and represents the core contribution of our paper. The simple integration of techniques from [1, 2, 3, 4, 5] cannot resolve the non-trivial difficulties inherent in the on-policy FedRL setting.
>
> On the one hand, [2, 3] focus on different RL settings that are not directly applicable to on-policy FedRL. Specifically: [2] studies offline and off-policy RL, where agents have access to an offline dataset and do not need to update their policies during the learning process; and [3] operates in a generative model (simulator) setting, where agents can generate next states and rewards based on a known transition probability function for any state-action pair. Techniques from [2, 3] are incompatible with on-policy settings, as they do not optimize policies actively and therefore cannot ensure low regret.
>
> On the other hand, fundamental challenges remain in adapting the techniques of [1, 4, 5] to achieve almost optimal regret in the on-policy FedRL setting. While [1] addresses on-policy FRL, it does not achieve almost optimal regret due to limitations in its round design, which cannot accommodate the stage design proposed in [4]. [4, 5] introduce techniques for achieving almost optimal regret in single-agent settings. However, extending these methods to the federated setting with multiple agents introduces non-trivial complexities, particularly in designing efficient communication protocols and ensuring logarithmic communication costs.
>
> The key to our almost optimal results lies in our novel mechanisms: **optional forced synchronization** and **heterogeneous event-triggered communication**. The optional forced synchronization offers flexibility to balance waiting time and communication rounds, addressing the challenges of heterogeneous agent speeds. The heterogeneous event-triggered communication mechanism allows for more visits in the early rounds of each stage compared to [1], significantly improving the communication cost. These mechanisms, {adapted to the stage design}, play a critical role in achieving the nearly optimal regret and logarithmic communication cost demonstrated in our work.
>
> In summary, our work goes beyond merely integrating existing techniques. It addresses unique challenges in the on-policy FedRL setting and provides a novel framework that achieves almost optimal regret and logarithmic communication cost, which existing approaches cannot accomplish.

---

> ### Author Response · Authors · 2024-11-22
> **Response to Reviewer RUvj (part two)**
>
> **Weakness 2:** Communication cost compared with [2] and [3].
>
> We respectfully disagree with the direct comparison to [2, 3] on the communication cost. As highlighted in our response to **Weakness 1**, we have pointed out that [2, 3] focus on RL settings that differ significantly from the on-policy FedRL setting studied in our work: [2] studies offline and off-policy RL, while [3] operates in a generative model (simulator) setting. Consequently, directly comparing the communication costs of our approach to those of [2, 3] is not justified due to these inherent differences.
>
> Below, we explain the additional $HSA$ term in the communication rounds of our method compared to [2, 3], highlighting the key distinctions.
>
> **Assumptions under Different RL Settings in [2, 3]:** [2] assumes the existence of a stationary visiting distribution for offline and off-policy RL, and [3] operates in a generative model (simulator) setting, where agents can simulate transitions for any state-action pair. These assumptions allow [2, 3] to avoid active exploration since their algorithms do not dynamically implement learned policies. This reduces their exploration requirements and contributes to their lower communication costs. However, these assumptions are specific to the offline, off-policy, and simulator-based RL settings and do not directly apply to the on-policy FedRL problem addressed in our work. Also, it is important to note that, despite achieving $O(H)$ communication rounds, [2] did not achieve the information-theoretic regret bound.
>
> **Key Challenges in On-Policy FedRL:** In contrast, our work focuses on the on-policy setting, which requires adaptively updating policies to balance exploration and exploitation while covering the entire state-action space. For on-policy FedRL, maintaining almost optimal regret while minimizing communication costs was an open question prior to our work. Using the novel algorithm design, our proposed FedQ-Advantage achieves an almost optimal regret with a logarithmic communication cost. We use an event-triggered condition to control visitation numbers: When one agent triggers the termination condition, we can only guarantee a sufficient increase in visits for a single $(s,a,h)$ triple, but without direct knowledge of other triples. Since there are $HSA$ state-action-horizon triples in finite-horizon episodic MDPs, it leads to the additional $HSA$ term in our communication cost.
>
>
> In summary, while [2, 3] achieve lower communication costs by operating in different RL settings that bypass the need for active exploration or dynamic policy updates, our method tackles the more complex on-policy FedRL problem. The additional $HSA$ term reflects the intrinsic challenges of balancing exploration and exploitation in this setting, making direct comparisons to [2, 3] inappropriate.

---

> ### Author Response · Authors · 2024-11-22
> **Response to Reviewer RUvj (part three)**
>
> **Weakness 3:** Presentation.
>
> Thank you for your feedback regarding the presentation in our paper. Following your suggestion, we have made significant revisions to improve the presentation, readability, and accessibility of the manuscript. Below, we summarize the key updates incorporated into our revised manuscript:
>
> (a) **Revised the Structure of Section 3.** In the revised manuscript, Section 3.1 now introduces the basic structure of our algorithm design (namely, aligned rounds and unaligned stages), Section 3.2 presents the algorithm details, and Section 3.3 provides the intuition behind the algorithm’s design.
>
> (b) **Significantly Reduced Notational Complexity.** In Section 3.2, we have removed the large block summarizing all the mathematical details of notations. Instead, notations are defined and explained contextually when each component of the algorithm is introduced. A comprehensive summary of all notations is now included in Appendix A for reference.
>
>
> (c) **Enhanced Explanations.** We have added detailed explanations for each step of the algorithm in Section 3.2. For instance, in Step 4, we explain the reference-advantage decomposition structure after introducing the decomposition-type update (Equation (13)).
>
>
> (d) **Improved Readability of Equations.** To enhance the readability of equations, we have provided a brief preview of the key idea or added an explanation after the equations. Also, we have avoided in-line equations that may make them less readable. For example,  the updates of the estimated $Q$ functions in Step 4 are presented as Equations (11) to (13) instead of within-line equations, and more explanations have been added after presenting Equations (11) to (13).
>
>
> (e) **Enhanced Algorithm Presentation.** The presentation of Algorithm 1 has been enhanced with clearly labeled steps and comments to match each stage of the algorithm design, improving overall clarity.
>
> (f) **New Section 3.3 on Intuition.** We have added a dedicated section to provide intuition behind the key features of our algorithm design, including exponential increasing stages, reference-advantage decompositions, heterogeneous event-triggered communication, and optional forced synchronization.
>
> We believe these changes address all the concerns raised in **Weakness 3** and significantly improve the presentation and accessibility of our manuscript. Thank you for your valuable suggestions.

---

> ### Author Response · Authors · 2024-11-22
> **Response to Reviewer RUvj (part four)**
>
> **Weakness 4:** Privacy concerns.
>
> Thanks for this question. We would like to clarify the incorrect perception that "the algorithm seems to require the agents to send too many estimates to the server alongside value estimates".
>
> In each round of FedQ-Advantage, local agents share two types of **summary statistics** with the server (see Line 11 of Algorithm 2): counts of visits ($n_h^{m,k}(s,a)$), and local sums of five estimated value functions on the next states of the visits. These value functions include reference value functions and advantage value functions, all of which are used in the reference-advantage type update. Importantly, these quantities are **summary statistics** rather than original trajectories, preserving the privacy of the underlying datasets.
>
> The communication cost of each round is only $O(HS)$, independent of the number of generated episodes. Sharing summary statistics, such as counts and values, is a standard practice in federated Q-Learning algorithms on tabular MDPs. For example, [1] employs similar sharing for on-policy FedRL (see Line 11 in Algorithm 1 of [1]), and [6] adopts a similar approach for off-policy FedRL (see Line 7 of Algorithm 2 and Equation (27) of [6]).
>
> These practices effectively balance privacy preservation with efficient communication. While privacy is indeed critical in federated settings, the use of summary statistics ensures that original trajectory data remains private, aligning with the privacy considerations of existing FedRL methods.
>
> [1] Zhong Zheng, Fengyu Gao, Lingzhou Xue, and Jing Yang. Federated Q-learning: Linear regret speedup with low communication cost. ICLR, 2024.
>
> [2] Jiin Woo, Laixi Shi, Gauri Joshi, and Yuejie Chi. Federated offline reinforcement learning: Collaborative single-policy coverage suffices. ICML, 2024.
>
> [3] Sudeep Salgia and Yuejie Chi. The sample-communication complexity trade-off in federated Q-learning. arXiv preprint arXiv:2408.16981, 2024.
>
> [4] Gen Li, Laixi Shi, Yuxin Chen, Yuantao Gu, and Yuejie Chi. Breaking the sample complexity barrier to regret-optimal model-free reinforcement learning. NeurIPS, 2021.
>
> [5] Zihan Zhang, Yuan Zhou, and Xiangyang Ji. Almost optimal model-free reinforcement learning via reference-advantage decomposition. NeurIPS, 2020.
>
> [6] Jiin Woo, Gauri Joshi, and Yuejie Chi. The blessing of heterogeneity in federated Q-learning: Linear speedup and beyond. ICML, 2023.

---

> ### Author Response · Authors · 2024-11-25
> **Following up on the rebuttal**
>
> Thanks again for your insightful comments and valuable advice! We have uploaded the revised draft and replied to your suggested weaknesses and questions. If you have further questions or comments, we are happy to reply in the author-reviewer discussion period, which ends on Nov 26th at 11:59 pm, AoE. If our response resolves your concerns, we kindly ask you to consider raising the rating of our work. Thank you very much for your time and efforts!

---

### Official Review · Reviewer_bZxN · 2024-11-04

**Soundness:** 3
**Presentation:** 3
**Contribution:** 2
**Rating:** 8
**Confidence:** 4

**Summary:**

This paper proposes a model-free federated reinforcement learning method called FedQ-Advantage. It focuses on  improving regret bounds and communication efficiency in multi-agent learning scenarios. This method leverages reference-advantage decomposition to reduce variance, designs an event-triggered communication method, and allows heterogeneous conditions. FedQ-Advantage achieves almost optimal regret, low communication cost, and near-linear regret speedup compared to the single-agent setting. It is especially efficient for distributed reinforcement learning tasks.

**Strengths:**

1. SOTA Results. The proposed model-free FedRL method achieves a logarithmic communication cost.
2. Novel techniques. The heterogeneous event-triggered communication in FedRL is new and effective.
3. The guarantees are given with solid theoretical proof.

**Weaknesses:**

1. The tabular setting is studied without function approximation. The action and state space are discrete and finite.
2. Literature Review. Is it possible to list federated value-based, and federated policy gradient-based methods separately and clearly? It would be better to have some pros and cons of federated Q-learning (value-based) methods compared to policy gradient-based methods.

**Questions:**

1. Comparison with model-based approaches. How does FedQ-Advantage compare to model-based FedRL methods in terms of convergence speed and communication complexity, given that model-based methods often have lower complexity?
2. Is there a trade-off between regret and communication complexity?

---

> ### Author Response · Authors · 2024-11-22
> **Response to Reviewer bZxN (part one)**
>
> We thank the reviewer for the careful reading and thoughtful comments. We have addressed the reviewer's questions in detail below and revised the paper accordingly. In particular, we have made significant efforts to improve the presentation, readability, and accessibility of the manuscript. The changes are marked in blue in the revised manuscript. We hope that the responses provided and the updates made to the paper satisfactorily address the reviewer’s concerns.
>
> **Weakness 1:** Performance with function approximation.
>
> Extending our algorithm to non-tabular settings, such as those involving linear function approximation, is an interesting direction for future research. In this context, [1] proposed a federated offline RL algorithm that utilizes confidence-bound methods for non-tabular settings, and [2, 3] applied the reference-advantage decomposition technique in offline RL settings with linear function approximation. These works demonstrate the feasibility of employing confidence bounds and reference-advantage decomposition techniques in non-tabular settings.
>
> However, generalizing our event-trigger condition and policy-switching strategy to non-tabular settings remains a significant challenge. For non-tabular settings with continuous state spaces, it becomes difficult to quantify visiting numbers. This necessitates the development of new mechanisms to terminate agent exploration effectively and update agent policies through event-trigger conditions, all while adhering to communication cost constraints. In addition, achieving logarithmic communication cost in non-tabular settings thus remains an open problem, as existing approaches such as [4, 5, 6] often require transmitting raw data (e.g., rewards or trajectories) to the central server, resulting in communication costs that scale as $O(MT)$. Addressing these challenges is a critical direction for future work.
>
> **Weakness 2:** Comparison between federated value-based methods and federated policy gradient-based methods.
>
> Thanks for this constructive comment. We have followed your suggestion to list the federated value-based methods and federated policy gradient-based methods separately and clearly in Appendix B of the revised manuscript.
>
> In general, federated policy gradient-based methods offer advantages over federated value-based methods in high-dimensional problems and benefit from the flexibility of stochastic policies. However, they are often associated with high variance in the gradients used for policy updates, which can lead to unstable training. Additionally, policy gradient-based methods typically require larger sample sizes to achieve convergence to an optimal solution, making them more resource-intensive compared to value-based methods. We hope this comparison is helpful in understanding the trade-offs between these approaches in federated RL settings.

---

> ### Author Response · Authors · 2024-11-22
> **Response to Reviewer bZxN (part two)**
>
> **Question 1:** Comparison with model-based FRL methods.
>
> Thanks for this helpful comment. Prior to our work, only a few studies have addressed the regret and communication cost of on-policy model-based multi-agent RL methods for tabular episodic MDPs with heterogeneous transition kernels. Table 1 in our manuscript provides a detailed comparison of the regret and communication costs between our proposed FedQ-Advantage and three model-based approaches, including [7] and two multi-batch RL methods.
>
> More specifically, [7] proposed a model-based FedRL algorithm with a regret bound of $\tilde{O}(\sqrt{MH^3S^2AT})$ and a communication cost of $O(M^2H^2S^2A^2\log(T))$. Compared to  FedQ-Advantage, the dependency on $M$, $H$, and $S$ in [7] is worse, and this increased complexity arises from its design to ensure robustness against adversarial agents.
>
> In contrast, FedQ-Advantage achieves an almost optimal regret with a logarithmic communication cost, maintaining a more favorable dependency on these parameters. While model-based approaches could benefit from faster convergence due to their explicit modeling of the environment, they may incur higher computational and communication costs, particularly when scaling to larger state-action spaces or handling heterogeneous agents.
>
> **Question 2:** Trade-off between regret and communication complexity.
>
> Thanks for this insightful comment. FedQ-Advantage reaches an almost optimal regret with a logarithmic communication cost. While our paper does not explicitly provide an option to control the trade-off between regret and communication complexity, such a mechanism can be incorporated by introducing two additional hyperparameters, say
>  $c_1$ and $c_2$, to adjust the exponential growth rate of stage sizes. Specifically, the growth rate can be set as $(y_{t+1} - y_t)/y_t\in [c_1/H,c_2/H]$ where $c_1,c_2>1$.
>
> Since our current growth rate $c_1=1,c_2=2$ induces an almost optimal regret, we discuss the trade-off for a better communication cost. By increasing $c_1$ and $c_2$, policy switching and communication become less frequent, reducing communication complexity. However, this comes at the cost of a potentially higher regret. This trade-off could be explored further in future work to provide more flexibility for different application needs.
>
> [1] Doudou Zhang, Yufeng Zhang, Aaron Sonabend-W, Zhaoran Wang, Junwei Lu, and Tianxi Cai. Federated offline reinforcement learning. Journal of the American Statistical Association, 2024.
>
> [2] Wei Xiong, Han Zhong, Chengshuai Shi, Cong Shen, Liwei Wang, and Tong Zhang. Nearly Minimax Optimal Offline Reinforcement Learning with Linear Function Approximation: Single-Agent MDP and Markov Game. ICLR, 2023.
>
> [3] Qiwei Di, Heyang Zhao, Jiafan He, and Quanquan Gu. Pessimistic Nonlinear Least-Squares Value Iteration for Offline Reinforcement Learning. ICLR, 2024
>
> [4] Dubey, Abhimanyu, and Alex Pentland. Provably efficient cooperative multi-agent reinforcement learning with function approximation. arXiv preprint arXiv:2103.04972 (2021).
>
> [5] Yifei Min, Jiafan He, Tianhao Wang, and Quanquan Gu. Cooperative multi-agent reinforcement learning: Asynchronous communication and linear function approximation. ICML, 2023.
>
> [6] Hao-Lun Hsu, Weixin Wang, Miroslav Pajic, and Pan Xu. Randomized Exploration in Cooperative Multi-Agent Reinforcement Learning. NeurIPS, 2024.
>
> [7] Yiding Chen, Xuezhou Zhang, Kaiqing Zhang, Mengdi Wang, and Xiaojin Zhu. Byzantine-robust online and offline distributed reinforcement learning. AISTATS, 2023.

---

> > ### Comment · Reviewer_bZxN · 2024-11-26
> > **Response**
> >
> > Thank you for your reply and clarifications. I will keep my score.

---

> ### Author Response · Authors · 2024-11-25
> **Following up on the rebuttal**
>
> Thanks again for your insightful comments and valuable advice! We have uploaded the revised draft and replied to your suggested weaknesses and questions. If you have further questions or comments, we are happy to reply in the author-reviewer discussion period, which ends on Nov 26th at 11:59 pm, AoE. If our response resolves your concerns, we kindly ask you to consider raising the rating of our work. Thank you very much for your time and efforts!

---

### Official Review · Reviewer_d5vs · 2024-11-06

**Soundness:** 3
**Presentation:** 3
**Contribution:** 3
**Rating:** 6
**Confidence:** 3

**Summary:**

This paper introduces FedQ-Advantage, a federated Q-learning algorithm for tabular episodic Markov Decision Processes. The algorithm utilizes reference-advantage decomposition to reduce variance and incorporates innovations like event-triggered communication, policy switching, and heterogeneous communication conditions. The paper proves that FedQ-Advantage achieves near-optimal regret with logarithmic communication cost.

**Strengths:**

1. The paper presents a well-structured and coherent methodology, detailing the innovations introduced in FedQ-Advantage, such as reference-advantage decomposition and communication mechanisms.

2. The proof of the algorithm’s near-optimal regret and logarithmic communication cost is rigorous and well-supported. The mathematical treatment is thorough, providing strong theoretical guarantees for the algorithm’s performance.

3. The algorithm's design, especially the heterogeneous communication triggering conditions and forced synchronization, shows creativity in addressing the challenges of federated learning.

**Weaknesses:**

1. The paper assumes a specific structure for communication conditions (e.g., event-triggered communication), which may not be universally applicable. The impact of these assumptions on algorithm performance in heterogeneous environments could be better analyzed.

2. The description of forced synchronization and its impact on performance, while novel, lacks a detailed explanation of how it interacts with other parts of the algorithm. A clearer explanation would help in understanding its necessity and effectiveness in the overall approach.

**Questions:**

1. How does the event-triggered communication strategy affect the algorithm’s performance when applied to environments with highly variable agent performance?

2. What are the trade-offs between forced synchronization and the other two mechanisms (communication triggering and policy switching)?

3. How would the FedQ-Advantage algorithm perform when applied to non-tabular environments or environments with larger state-action spaces, where Q-learning is typically more challenging?

4. In figure 1, what do $k_{12}$ and $k_{22}$ mean? Are these typos? Moreover, can you explain again about the condition and process when communication occurs?

---

> ### Author Response · Authors · 2024-11-22
> **Response to Reviewer d5vs (part one)**
>
> We thank the reviewer for the careful reading and thoughtful comments. We have addressed the reviewer's questions in detail below and revised the paper accordingly. In particular, we have made significant efforts to improve the presentation, readability, and accessibility of the manuscript. The changes are marked in blue in the revised manuscript. We hope that the responses provided and the updates made to the paper satisfactorily address the reviewer’s concerns.
>
> **Weaknesses 1 \& 2 and Questions 1 \& 2:** Communication conditions in heterogeneous settings such as highly variable agent performance.
>
> Thank you for your insightful comments and questions. Following your suggestions, we have expanded the discussion on the impact of our communication conditions in Section 3.3 of the revised manuscript. Below, we provide a detailed response to address the key points raised.
>
> Our communication conditions incorporate two key features: heterogeneous event-triggered communication and optional forced synchronization (see Step 2 of the algorithm design in Section 3.2 of the revised manuscript). As these features often work together to influence performance (both regret and communication cost), we discuss them collectively to address your comments on Weaknesses 1 \& 2 and Questions 1 \& 2. Specifically: **Weakness 1** and **Question 1** focus on (a) the adaptation and regret analysis under heterogeneous agents’ exploration speeds. The first part of **Weakness 2** pertains to (b) the impact of optional forced synchronization on performance. **Question 2** and the remaining part of **Weakness 2** address (c) the impact of optional forced synchronization on communication triggering and policy switching.
>
> Our response to **(a) Adaptation and Regret Analysis Under Heterogeneous Exploration Speeds:**
>
> The communication conditions, regardless of the choice of forced synchronization, do not affect the algorithm's almost optimal regret and adapt to heterogeneous exploration speeds among agents. As demonstrated in Theorem 4.1 of our paper, our algorithm achieves an almost optimal regret and linear speedup, which remains unaffected by variations in local agents' exploration speeds. This demonstrates the adaptability of our method to heterogeneous agents.
>
> Our response to **(b) Impact of Optional Forced Synchronization on Performance.**
>
> The optional forced synchronization induces a trade-off between waiting time and communication rounds when agents explore at different speeds. The following equations highlight the conditions under which communication occurs, depending on whether synchronization is enforced:
> $$\forall (s,a,h,m,k),\ n^{m,k}\_h(s,a)\leq c_h^k(s,a),$$
> $$\forall k,\ \exists (s,a,h,m),\ s.t.\ n^{m,k}\_h(s,a) = c_h^k(s,a),\quad\mbox{ if }u_{syn}=\textnormal{TRUE}\quad (1)$$
> $$\forall m,k,\ \exists (s,a,h),\ s.t.\ n^{m,k}\_h(s,a) = c_h^k(s,a),\quad\mbox{ if }u_{syn}=\textnormal{FALSE}.\quad (2)$$
> When optional forced synchronization is enabled (i.e., $u_{syn} = \textnormal{TRUE}$), exploration and communication occur as soon as one agent reaches the threshold $c_h^k(s,a)$. This allows faster agents to avoid waiting for slower ones, minimizing waiting time. However, Equation (1) guarantees sufficient exploration by only one agent, resulting in varied episode counts across agents. This configuration is suitable for tasks sensitive to waiting time.
>
> When optional forced synchronization is disabled (i.e., $u_{syn} = \textnormal{FALSE}$), communication occurs only after all agents meet the threshold $c_h^k(s,a)$. Equation (2) ensures sufficient exploration by all agents, with episode counts being roughly balanced. This allows for more extensive exploration within a round, reducing communication costs but potentially increasing waiting time for faster agents.
>
> Our response to **(c) Impact of Optional Forced Synchronization on Communication and Policy Switching:**
>
> The optional forced synchronization affects communication triggering but not policy switching.
>
> On the one hand, under our communication conditions and the stage renewal condition (as explained in Step 3 in Section 3.2 of the revised manuscript), stage sizes grow exponentially: $y_{t+1} = (1+\Theta(1/H))y_t$ (see the first intuition in Section 3.3 of our revised manuscript). Since policy switching only occurs at the end of each stage, it remains unaffected by the choice of synchronization. On the other hand, as discussed in (b), enabling forced synchronization (i.e., $u_{syn} = \textnormal{TRUE}$) guarantees sufficient exploration by only one agent, resulting in more frequent communication triggers. In contrast, disabling synchronization (i.e., $u_{syn} = \textnormal{FALSE}$) ensures all agents sufficiently explore, reducing the number of communication triggers at the cost of potential waiting time.
>
> We believe these clarifications address your concerns and provide a deeper understanding of how our communication conditions impact regret and communication costs.

---

> ### Author Response · Authors · 2024-11-22
> **Response to Reviewer d5vs (part two)**
>
> **Question 3:** Performance in non-tabular environments or environments with larger state-action spaces
>
> Thank you for this constructive comment. In the sequel, we first address the performance of FedQ-Advantage in environments with larger state-action spaces and then discuss its potential extension to non-tabular settings.
>
> On the one hand, our current results already cover the MDPs with finite but potentially larger state-action spaces, where both the regret bound and communication cost retain the same forms, involving $HSA$. Figure 4 in Appendix C presents numerical experiments under two settings: (1) a smaller setting with $H=5, S=3, A=2$, replicating the setup in [1]; and (2) a larger setting with $H=20, S=20, A=5$, where learning the optimal policy for a single replication takes approximately 24 hours. These results highlight the superior regret and communication cost performance of FedQ-Advantage. Notably, FedQ-Hoeffding fails to converge in the larger-scale setting, while FedQ-Advantage successfully converges.
>
> On the other hand, generalizing FedQ-Advantage to non-tabular settings, such as those involving linear function approximation, is an interesting direction for future work.  In this context, [2] proposed a federated offline reinforcement learning algorithm using confidence-bound methods, and [3, 4] applied reference-advantage decomposition techniques to offline RL with linear function approximation. These studies demonstrate the feasibility of using confidence bounds and reference-advantage decomposition techniques in non-tabular environments. However, adapting our event-trigger condition and policy-switching strategy to non-tabular settings presents significant challenges. For environments with continuous state spaces, quantifying visitation counts, terminating exploration, and updating policies based on event-trigger conditions become complex, especially under communication cost constraints. In addition, achieving logarithmic communication cost in non-tabular settings thus remains an open problem, as existing approaches such as [5, 6, 7] often require transmitting raw data (e.g., rewards or trajectories) to the central server, resulting in communication costs that scale as $O(MT)$. Addressing these challenges is a critical direction for future work.
>
> **Question 4.1:** The definition of notations in Figure 1.
>
> Thanks for pointing out this issue. We have revised the notations and added definitions in Figure 1 to enhance clarity. In the updated description of Figure 1, $k_{1t}$ is the last round index for stage $t$ of $(s^1,a^1,h^1)$ and $k_{2t}$ is the last round index for stage $t$ of $(s^2,a^2,h^2)$.
>
> **Question 4.2:** Condition and process of our communication.
>
> We have provided a detailed explanation of the communication condition and process in Step 2 of Section 3.2 to enhance readability and clarity. Below, we summarize the key points:
>
> At the start of round $k$, agents start collecting trajectories based on the total visiting number of the previous stage, $n_h^k(s,a)$, and the existing visiting number in the current stage, $\tilde{n}\_h^k(s,a)$, for any $(s,a,h)$, received from the central server. If $u_{syn} = \textnormal{TRUE}$, for any agent $m$, at the end of each episode, if any $(s,a,h)$ has been visited by $c_h^k(s,a)$ times, the agent will stop exploration and send a signal to the server that requests all agents to abort the exploration. If $u_{syn} = \textnormal{FALSE}$, the central server will wait until, for each agent, there exists a triple $(s,a,h)$ that has been visited by $c_h^k(s,a)$ times. Once these conditions are satisfied, agents send their local sums to the central server, and round $k$ concludes.
>
>
> [1] Zhong Zheng, Fengyu Gao, Lingzhou Xue, and Jing Yang. Federated Q-learning: Linear regret speedup with low communication cost. ICLR, 2024.
>
> [2] Doudou Zhang, Yufeng Zhang, Aaron Sonabend-W, Zhaoran Wang, Junwei Lu, and Tianxi Cai. Federated offline reinforcement learning. Journal of the American Statistical Association, 2024.
>
> [3] Wei Xiong, Han Zhong, Chengshuai Shi, Cong Shen, Liwei Wang, and Tong Zhang. Nearly Minimax Optimal Offline Reinforcement Learning with Linear Function Approximation: Single-Agent MDP and Markov Game. ICLR, 2023.
>
> [4] Qiwei Di, Heyang Zhao, Jiafan He, and Quanquan Gu. Pessimistic Nonlinear Least-Squares Value Iteration for Offline Reinforcement Learning. ICLR, 2024
>
> [5] Dubey, Abhimanyu, and Alex Pentland. Provably efficient cooperative multi-agent reinforcement learning with function approximation. arXiv preprint arXiv:2103.04972 (2021).
>
> [6] Yifei Min, Jiafan He, Tianhao Wang, and Quanquan Gu. Cooperative multi-agent reinforcement learning: Asynchronous communication and linear function approximation. ICML, 2023.
>
> [7] Hao-Lun Hsu, Weixin Wang, Miroslav Pajic, and Pan Xu. Randomized Exploration in Cooperative Multi-Agent Reinforcement Learning. NeurIPS, 2024.

---

> > ### Comment · Reviewer_d5vs · 2024-11-26
> >
> > Thank you for the detailed response, I will keep my positive score.

---

> ### Author Response · Authors · 2024-11-25
> **Following up on the rebuttal**
>
> Thanks again for your insightful comments and valuable advice! We have uploaded the revised draft and replied to your suggested weaknesses and questions. If you have further questions or comments, we are happy to reply in the author-reviewer discussion period, which ends on Nov 26th at 11:59 pm, AoE. If our response resolves your concerns, we kindly ask you to consider raising the rating of our work. Thank you very much for your time and efforts!

---

### Official Review · Reviewer_CkE8 · 2024-11-07

**Soundness:** 3
**Presentation:** 3
**Contribution:** 3
**Rating:** 8
**Confidence:** 3

**Summary:**

The paper addresses the problem of federated reinforcement learning (RL) in a model-free, tabular episodic Markov Decision Process setting.  The FedQ-Advantage algorithm  proposed allows multiple agents to collaboratively learn an optimal policy through a central server without sharing raw data. Variance reduction is achieved through a reference adaptive decomposition. Novel contributions include event triggered communication and policy switching. The result is an an algorithm with almost optimal cost and communication that scales logarithmically with $T$ and a constant that scales better than previous algorithms reported in the literature.

**Strengths:**

The paper seems technically sound, I only skimmed most of the material in the appendix but the structure of the proofs and flow makes sense. I believe the paper will be of interest to the ICLR community. I believe the combination of the reduced communication burden and the near optimal regret together make this a significant result.

**Weaknesses:**

The main weakness of the paper is that it is extremely notationally heavy and very little intuition is provided in any of the steps - a representative example of this is the section "Updates of estimated value functions and policies". It's 90% equation. I appreciate that it is difficult to keep this under control, but I believe this needs to be improved in order for it to have any value.

**Questions:**

Can the authors find a way to describe their work and provide some intuition behind the various steps. While I believe the validity of the results, I don't understand where the insights and intuition come from.

---

> ### Author Response · Authors · 2024-11-22
> **Response to Reviewer CkE8**
>
> We thank the reviewer for the careful reading and thoughtful comments. We have addressed the reviewer's questions in detail below and revised the paper accordingly. In particular, we have made significant efforts to improve the presentation, readability, and accessibility of the manuscript. The changes are marked in blue in the revised manuscript. We hope that the responses provided and the updates made to the paper satisfactorily address the reviewer’s concerns.
>
> **Weakness and Question:** Description of the work and intuition behind key steps.
>
> Thank you for your feedback regarding the notational complexity and lack of intuition in our paper. Following your suggestion, we have made significant revisions to improve the presentation, readability, and accessibility of the manuscript. Below, we summarize the key updates incorporated into our revised manuscript:
>
> (a) **Revised the Structure of Section 3.** In the revised manuscript, Section 3.1 now introduces the basic structure of our algorithm design (namely, aligned rounds and unaligned stages), Section 3.2 presents the algorithm details, and Section 3.3 provides the intuition behind the algorithm’s design.
>
> (b) **Significantly Reduced Notational Complexity.** In Section 3.2, we have removed the large block summarizing all the mathematical details of notations. Instead, notations are defined and explained contextually when each component of the algorithm is introduced. A comprehensive summary of all notations is now included in Appendix A for reference.
>
>
> (c) **Enhanced Explanations.** We have added detailed explanations for each step of the algorithm in Section 3.2. For instance, in Step 4 ("Updates of estimated value functions and policies"), we explain the reference-advantage decomposition structure after introducing the decomposition-type update (Equation (13)).
>
>
> (d) **Improved Readability of Equations.** To enhance the readability of equations, we have provided a brief preview of the key idea or added an explanation after the equations. Also, we have avoided in-line equations that may make them less readable. For example,  the updates of the estimated $Q$ functions in Step 4 (``Updates of estimated value functions and policies") are presented as Equations (11) to (13) instead of within-line equations, and more explanations have been added after presenting Equations (11) to (13).
>
>
> (e) **Enhanced Algorithm Presentation.** The presentation of Algorithm 1 has been enhanced with clearly labeled steps and comments to match each stage of the algorithm design, improving overall clarity.
>
> (f) **New Section 3.3 on Intuition.** We have added a dedicated section to provide intuition behind the key features of our algorithm design, including exponential increasing stages, reference-advantage decompositions, heterogeneous event-triggered communication, and optional forced synchronization.
>
> We believe these changes address the concerns raised and significantly improve the presentation and accessibility of our manuscript. Thank you for your valuable suggestions.

---

> ### Author Response · Authors · 2024-11-25
> **Following up on the rebuttal**
>
> Thanks again for your insightful comments and valuable advice! We have uploaded the revised draft and replied to your suggested weaknesses and questions. If you have further questions or comments, we are happy to reply in the author-reviewer discussion period, which ends on Nov 26th at 11:59 pm, AoE. If our response resolves your concerns, we kindly ask you to consider raising the rating of our work. Thank you very much for your time and efforts!

---

> > ### Comment · Reviewer_CkE8 · 2024-11-26
> > **Response**
> >
> > I appreciate the effort the authors have put into revising the manuscript. It provides a big improvement in clarity. I will adjusted my score accordingly.

---

### Meta-Review · Area_Chair_GXmf · 2024-12-21

**Metareview:**

The authors proposed the FedQ-Advantage Algorithm for federated reinforcement learning, enabling multiple agents to learn collaboratively through a central server. The algorithm improves upon existing methods by: (1) reducing communication between agents and the server by sharing updates only when necessary, (2) introducing a novel reference-advantage decomposition to enhance learning efficiency, and (3) allowing agents to learn independently while benefiting from collective knowledge. The results demonstrate that FedQ-Advantage achieves near-optimal performance with significantly reduced communication compared to previous methods.

This work has received predominantly positive evaluations. Minor concerns were raised regarding the lack of comparison with other online exploration-based cooperative learning algorithms, the heavy use of notations impacting readability, and the need for clearer explanations of the synchronization condition’s novelty and intuition. Following the authors’ detailed responses and discussions, these issues appear to have been adequately addressed.

I recommend accepting this paper and encourage the authors to further improve the clarity of their contributions and intuitions in the final version.

**Additional Comments On Reviewer Discussion:**

Some reviewers raised concerns about the lack of comparison with other online exploration-based cooperative learning algorithms, such as Coop-LSVI, Asyn-LSVI, and CoopTS. Additionally, they noted that the heavy notation in the presentation reduced readability and sought greater clarity on the novelty and intuition of the synchronization condition. After the authors provided detailed responses and engaged in discussions, these issues appear to have been adequately addressed.

One reviewer rated the paper relatively low, arguing that the techniques used are derived from other settings, such as single-agent RL, and that similar algorithms have been studied in offline RL and the generative model setting. However, I do not believe that comparisons with offline RL algorithms are necessary, as this paper addresses the online exploration setting, which is significantly more challenging. Moreover, communication costs in offline and generative model settings are not appropriate benchmarks for refuting the results of this work.

While I agree to some extent that the technical contributions are not entirely unique, integrating these techniques into a novel setting and achieving both low regret and reduced communication costs is a nontrivial accomplishment. Given that the reviewer in question did not engage in further discussion and the paper’s overall quality has been acknowledged by most reviewers, I recommend acceptance.

---

### Decision · Program_Chairs · 2025-01-22

Accept (Poster)